# Context Learning for Multi-Agent Discussion

**Xingyuan Hua**[1], **Sheng Yue**[2]*, **Xinyi Li**[3], **Yizhe Zhao**[1], **Jinrui Zhang**[1], **Ju Ren**[1,4]

[1]Department of Computer Science and Technology, Tsinghua University,
[2]School of Cyber Science and Technology, Sun Yat-sen University,
[3]College of Computer Science, Northwest University,
[4]State Key Laboratory of Internet Architecture, Tsinghua University

## Abstract

Multi-Agent Discussion (MAD) has garnered increasing attention very recently, where multiple LLM instances collaboratively solve problems via structured discussion. However, we find that current MAD methods easily suffer from discussion inconsistency—LLMs fail to reach a coherent solution—due to the misalignment between their individual contexts. In this paper, we introduce a multi-LLM context learning method (`M2CL`) that learns a context generator for each agent, capable of dynamically generating context instructions per discussion round via automatic information organization and refinement. Specifically, inspired by our theoretical insights on the context instruction, `M2CL` trains the generators to control context coherence and output discrepancies via a carefully crafted self-adaptive mechanism. It enables LLMs to avoid premature convergence on "majority noise" and progressively reach the correct consensus. We evaluate `M2CL` on challenging tasks, including academic reasoning, embodied tasks, and mobile control. The results show that the performance of `M2CL` significantly surpasses existing methods by 20%–50%, while enjoying favorable transferability and computational efficiency.[1]

## 1 Introduction

Large Language Models (LLMs) have demonstrated transformative impact across a large number of real-world domains, including education, healthcare, and scientific research, where an LLM instance is employed to automate the process of content generation, reasoning, or decision-making (Zheng et al., 2023; Imani et al., 2023; Zhang et al., 2024; Goyal et al., 2024; Wu et al., 2025; Yu et al., 2025). Yet, it has been recognized that single-LLM-built systems often struggle in the problems requiring complex multi-step reasoning or multi-tool using, such as complicated proof (Cobbe et al., 2021), large-scale code generation (Wang et al., 2025), and embodied agentic tasks (Ahn et al., 2022; Shen et al., 2024), because its single viewpoint of a problem easily limits the ability to explore multiple reasoning paths, leverage external tools effectively, or adapt to dynamic task requirements (Li et al., 2023; Smit et al., 2024).

Very recently, research has shifted towards *Multi-Agent Discussion* (MAD) (Smit et al., 2024), where multiple LLM instances collaboratively solve problems via structured discussions (Du et al., 2023; Liu et al., 2024). In a typical MAD framework, each LLM instance is pre-assigned a set of crafted *contexts* that represent diverse solution perspectives of the problem to be solved. Equipped with these distinct context instructions, LLMs continue to discuss with each other for a solution consensus (Park et al., 2023; Shanahan et al., 2023; Wei et al., 2023; Lu et al., 2024; Liu et al., 2024). Such 'society-of-mind' paradigms are expected to improve reasoning accuracy by enhancing creativity and expanding the search space for possible solutions, and have shown great potential across various complex tasks, including software engineering (Gu, 2023) and scientific discovery (Sprueill et al., 2024).

Albeit achieving improved performance over single-LLM settings, we find that current multi-agent collaboration approaches typically suffer from discussion inconsistency, that is, the majority of LLM instances fail to reach an agreement on a coherent solution (as showcased in Fig. 2), easily making the collaborative decision dominated by noise rather than principled reasoning. The underlying reason

---

*Correspondence to: `yuesh5@mail.sysu.edu.cn`
[1]Code is available at `https://github.com/HansenHua/M2CL-ICLR26`.

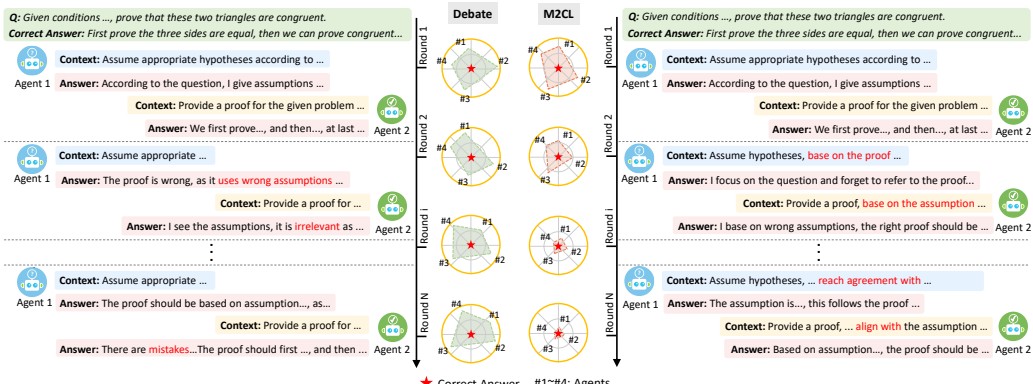

Figure 1: An illustration of context misalignment of an existing method (`Debate` Du et al. (2023)) on a multi-step proof task. Pre-assigned context instructions (in the blue and yellow boxes of the left part) provide insufficient guidance on information fusion, leading to conflict in reasoning.

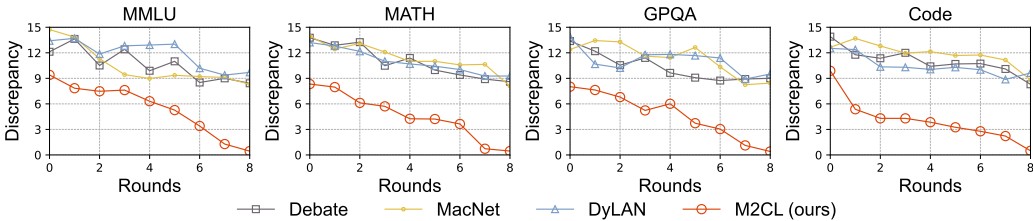

Figure 2: The discrepancy between the answers of participating LLM instances. The discrepancy is characterized by the maximum distance between participating LLMs' output embeddings.

primarily lies in *context misalignment* between LLMs. On one hand, the pre-assigned contexts or role-based instructions lack a nuanced understanding of the task; they are often rigid, incomplete or biased, which would misguide the reasoning of individual LLMs (Jang et al., 2025). On the other hand, these contexts often fall short in effective fusion of information exchanged among LLMs, and thus hardly steer the discussion towards coherent solutions. As illustrated in Fig. 1, for a multi-step mathematical proof task, one LLM agent may correctly derive an intermediate result, yet another LLM—despite receiving this result in the extended context—does not effectively incorporate it into its own reasoning chain. Since the context instruction does not explicitly enforce leveraging conclusions from other LLMs, other LLMs may redundantly re-derive the step or even give an inconsistent argument (see Appendix H) for full results).

This paper aims to answer: *"how can we obtain the contexts for MAD that can continually guide multi-LLM discussion towards a correct consensus?"* A straightforward solution is to manually adapt instructions in contexts as the discussion progresses. It, however, is labor-intensive and requires extra expert knowledge, rendering it impractical for complex tasks or large-scale collaboration. Instead, a more reasonable approach is to develop context learning mechanisms that enable evolving the context instructions based on the intermediate discussion results. Although promising, it is highly challenging to evaluate the contribution of LLMs' contexts to the final solution and control the coherence among inter- and intra-LLM outputs.

To tackle these challenges, we propose a *multi-LLM context learning* (`M2CL`) method for efficient MAD, which learns a context generator for each agent, capable of dynamically generating context instructions per discussion round via automatic information organization and refinement. First, we characterize the impact of initial and evolving contexts on the discussion performance. Building upon the analytical insights, we train the generators to control context coherence and output discrepancies. To strike the right tradeoff therein, we devise a self-adaptive balancing mechanism, enabling LLMs to progressively align on correct consensus while avoiding premature convergence on "majority noise". Further, we develop a lightweight context initialization approach, which tends to assign LLMs with diverse initial instructions that are approximately orthogonal in the latent space, enabling sufficient coverage of complementary solution perspectives.

We systematically evaluate the proposed method across 9 challenging benchmarks, including LLM reasoning, embodied agentic tasks, and mobile GUI control. The results demonstrate that M2CL significantly enhances the MAD performance under various numbers of participating LLMs, consistently outperforming existing methods by **20%–50%**, particularly in complex GUI control tasks. M2CL enjoys a more favorable MAD "scaling law" and exhibits great efficiency, where a runtime overhead of at most $10\%$ suffices to achieve more than $20\%$ performance gains. Further, we find that the learned context generators can be migrated to different LLM architectures with consistent performance improvement.

Our main contributions are summarized as follows:

- We propose M2CL, a principled multi-LLM context learning method that learns a context generator for each agent, capable of dynamically generating context instructions per discussion round.
- We devise a lightweight context initialization approach that can assign LLMs with diverse initial instructions, thereby enabling sufficient coverage of complementary solution perspectives.
- We systematically evaluate the proposed method across a range of challenging benchmarks and corroborate the efficacy and efficiency of the proposed method.

## 2 RELATED WORK

**Multi-agent framework.** Multi-agent frameworks have been proposed to enhance the reasoning capabilities of single-LLM prompting methods. Several works utilize two LLMs to iteratively generate and evaluate to refine the final answer (Raman et al., 2022; Yao et al., 2023; Paul et al., 2023; Madaan et al., 2024). However, scaling them to a larger number of LLMs to make full use of the wisdom of crowd is challenging due to the pairwise dependency of generation and evaluation. To incorporate more LLMs, several studies have explored another multi-agent framework in which each LLM has access to the history of all preceding LLMs' responses (Chan et al., 2023; Du et al., 2023; Liang et al., 2023; Smit et al., 2024; Liu et al., 2024; Zhuge et al., 2024). Most existing methods remain constrained by manually defined inter-LLM topologies or workflows. More recently, researches have attempted to overcome this limitation by automatically optimizing workflows (Zhuge et al., 2024; Liu et al., 2024; Zhang et al., 2025b). However, such evolution is often one-shot. For instance, Zhuge et al. (2024) describe language agent systems as optimizable computational graphs and enables automatic improvements of LLM prompts and inter-LLM orchestration. Zhang et al. (2025b) employs Monte Carlo Tree Search to construct complex Multi-LLM systems tailored to a specific task domain. Liu et al. (2024) propose a multi-LLM collaboration method which constructs dynamic communication structure by scoring other LLMs. While these approaches allow for scaling the number of LLMs, designing appropriate contexts for them to collaboratively solve problems remains difficult, as single-viewpoint context instructions fall short in guidance on how to organize and refine information across agents.

**Context learning.** Context learning has recently attracted significant attention, as it allows LLMs to adjust their behavior at inference time by modifying the input context, without requiring any gradient-based training Von Oswald et al. (2023); Todd et al. (2024); Li et al. (2024a). One line of work focuses on context selection, which identifies the most relevant examples or information to include in the context Zhang et al. (2022); Lu et al. (2023); Xiong et al. (2024); Purohit et al. (2025). To eliminate manual selection, Lu et al. (2023) leverage reinforcement learning to learn a context selection policy. Xiong et al. (2024) query LLM to obtain knowledge and then query a retriever to obtain the final context. Beyond selection, some recent methods have proposed to generate or evolve context (Zhuge et al., 2024; Li et al., 2024b; Zhang et al., 2025a). Madaan et al. (2024) leverage feedback generated by LLMs as extra prompt and iteratively incorporate it into revised drafts, aiming to enhance the coherence of the generated text. Pandita et al. (2025) dynamically refines prompts during inference using textual feedback from prior outputs, aiming to improve contextual alignment and generate more consistent responses. Albeit with promising results, these methods are grounded in single-LLM formulations and struggle with inter-LLM inconsistency during multi-agent dscussion, where it is crucial to guide LLMs to make full use of others' intermediate results.

## 3 PRELIMINARIES

In this section, we provide necessary backgrounds and definitions of our investigated problem.

**Context learning.** To formally define context learning, we begin with the standard probabilistic model of an autoregressive LLM. An autoregressive model, parameterized by $\phi$, generates an output sequence $X = (x_1, x_2, \ldots, x_L)$ given an input context $C$ by maximizing the conditional probability:

$$P_\phi(X|C) = \prod_{l=1}^{L} P_\phi(x_l|x_{<l}, C). \tag{1}$$

Historically, in the paradigm of prompt engineering, the context $C$ was treated as a static string of text. This view is insufficient for complex agentic tasks which leverage dynamic, structured, and multifaceted information stream. To address this, context learning is redefined as a dynamically organized collection of information, denoted as $\{c_1, c_2, \ldots, c_n\}$. Each component $c_k$ can be instructions, external knowledge (Lewis et al., 2020), available external tools (Qin et al., 2023), and memory (Zhang et al., 2025a). These components are sourced, filtered, and formatted by a set of functions $f_k$, and orchestrated into a coherent representation by a high-level assembly function $\mathcal{A}$:

$$C = \mathcal{A}(f_1(c_1), f_2(c_2), \ldots, f_n(c_n)). \tag{2}$$

**Multi-LLM context learning.** MAD involves a set of $N$ LLM instances collaboratively solving a task via multi-round inter-LLM discussion (debate). Each LLM $i$ is endowed with an evolving instruction context in terms of the task description, available external tools and current knowledge aggregation. At the $t$-th round, we use the concatenation function as the assembly function $\mathcal{A}$ and construct the context with three components: (*i*) the task goal $P \in \mathbb{R}^{d_{model} \times n}$, (*ii*) the concatenation of responses from all other LLMs in the previous round $\bar{X}_i^{t-1} = [X_j^{t-1}]_{j \neq i} \in \mathbb{R}^{d_{model} \times n}$, where $X_j^{t-1}$ is the response of LLM $j$ at round $t-1$, and (*iii*) the current instruction context $I_i^t \in \mathbb{R}^{d_{model} \times n}$ ($d_{model}$ refers to the dimension of embedding). The task goal represents the ultimate objective of collaboration and remains invariant across rounds. The second component serves as a dynamic memory that incorporates cross-LLM interaction history. For the third component, in constrast to existing efforts relying on static preassigned roles, we employ an instruction generator $\mathcal{G}$, parameterized by $\theta_i$, to adaptively refine it into a per-step instruction $I_i^t$, conditioned on the task goal $P$, the initial instruction $I_i^b$, and the concatenated response $\bar{X}_i^{t-1}$:

$$I_i^t = \mathcal{G}_{\theta_i}([P; I_i^b; \bar{X}_i^{t-1}]). \tag{3}$$

Given context $C_i^t = [I_i^t, \bar{X}_i^{t-1}, P]$, each LLM $i$, parameterized by $\phi_i$, generate its response by:

$$X_i^t = \arg\max_X P_{\phi_i}(X|C_i^t). \tag{4}$$

After $T$ rounds of interaction, the final result is obtained by a majority vote on the LLMs' outputs generated in the final round.

## 4 MOTIVATION

In this section, we investigate the quantified impact of the contexts on the MAD performance. First, we introduce the formulation of attention activation, denoted as $a(\cdot)$, as follows (Vaswani, 2017):

$$a(C_i^t) \doteq W_V[I_i^t; \bar{X}_i^{t-1}; P]\mathrm{softmax}\left(\frac{(W_K[I_i^t; \bar{X}_i^{t-1}; P])^T W_Q P}{\sqrt{d}}\right) \tag{5}$$

where $\sqrt{d}$ is a scaling factor. $W_Q, W_K$, and $W_V$ denote the parameter weight matrices in the attention mechanism. $a(C_i^t) \in \mathbb{R}^{d_{model} \times n}$ is a matrix with the same shape as $[P; X; I]$

From this, we have the following theorem characterizing the total distance between the activations of the correct answer $a_c$ and that induced by context $C_j^t$. Here, we utilize activation distance instead of token embedding distance as attention activation captures deep representational similarity learned through the model's internal reasoning process, making it more robust to superficial linguistic variations.

**Theorem 4.1.** *Assume that the attention activation function is $L_a$-smooth and define the weight vector as $\omega \doteq [\omega_1, \omega_2, \ldots, \omega_N]$, with the initial context $C_i^b \doteq [I_i^b; P]$. Then, the following fact holds:*

$$\sum_{i=1}^{N} \|a_c - a(C_i^t)\| \leq \sum_{i=1}^{N} \Big( \sum_{j=1}^{N} \|a(C_i^t) - a(C_j^t)\| + (N+1)L_a\|C_i^t - C_i^b\| \Big)$$

$$+ N \min_{\omega} \|a_c - \sum_{i=1}^{N} \omega_i a(C_i^b)\|. \tag{6}$$

*Proof.* For a detailed proof, please refer to Appendix B. $\square$

Theorem 4.1 corroborates the necessity of multi-LLM context learning, consisting of both initialization and evolution. The first term, $\|a(C_i^t) - a(C_j^t)\| + (N+1)L_a\|C_i^t - C_i^b\|$, captures the divergence among LLMs' activations as well as the deviation from their initial contexts. This indicates contexts must be continuously evolved to reduce inter-LLM discrepancies while keeping coherent, promoting consistency of reasoning chains. The second term, $\min_{\omega} \|a_c - \sum_{i=1}^{N} \omega_i a(C_i^b)\|$, depends solely on the initial contexts $I_i^b$. It implies that orthogonality among initial activations provides a comprehensive basis, allowing the contexts to approximate the correct activation more effectively. This highlights the importance of diverse solution perspectives at the initialization stage. Together, the result motivates a two-stage design: initializing contexts to ensure diverse solution perspectives and evolving intermediate contexts to drive a consensus across LLMs.

## 5 MULTI-LLM CONTEXT LEARNING

In this section, we introduce a multi-LLM context learning method that can select appropriate contexts for each LLM and dynamically adapt the context according to the evolving task completion status.

### 5.1 CONTEXT INITIALIZATION

The context initialization serves as the foundation of the entire multi-agent interaction process, as it frames the capability scope of LLMs and fundamentally shapes the chain of subsequent information exchange. Therefore, we propose to select initial contexts from a predefined pool $\{I_i^b\}_{i=1}^{M}$ containing prompts with diverse perspectives.

Motivated by Theorem 4.1, we provide the following context initialization mechanism:

$$\boldsymbol{I}^b = \{I_1^b, \ldots, I_N^b\} = \arg\min_{\boldsymbol{I}^b} \Big\{ \min_{\omega} \|\sum_{i=1}^{N} \omega_i a([I_i^b; P] - a_c)\| \Big\}. \tag{7}$$

Eq. (7) identifies a subset of contexts $\boldsymbol{I}^b$ whose activation best reconstruct the target activation $a_c$. As the dimension of the activation matrix $a([I_i^b; P]) \in \mathbb{R}^{d_{model} \times n}$ is far larger than the number of selected contexts $N$, a set of matrices that aims to best reconstruct the correct activation $a_c$ naturally tends toward forming a set of basis-like directions. The resulting near-orthogonal activations form a compact basis, ensuring each context contributes unique, non-overlapping information for subsequent discussion.

However, Eq. (7) is impractical since the correct activation $a_c$ is not accessible during initialization. Inspired by prior work (Yang et al., 2024b; 2025), we project the activation into a latent space with function $f$:

$$\boldsymbol{I}^b = \{I_1^b, \ldots, I_N^b\} = \arg\min_{\boldsymbol{I}^b} \Big\{ \min_{\omega} \|\sum_{i=1}^{N} \omega_i f(a([I_i^b; P])) - f(a_c)\| \Big\}, \tag{8}$$

Then, we use the problem space as the selected contexts are solely dependent on the problem. This projection preserves the original orthogonality properties of activations, ensuring that diverse perspectives remain distinguishable in other spaces. Therefore, we reformulate the initialization

mechanism:

$$\boldsymbol{I}^b = \{I_1^b, \ldots, I_N^b\} = \arg\min_{\boldsymbol{I}^b} \left\{ \min_{\omega} \| \sum_{i=1}^{N} \omega_i f(a([I_i^b; P])) - v_P \| \right\}, \tag{9}$$

where $v_P$ denotes the sentence vector of the question $P$. To obtain the projection $f(\cdot)$ (parameterized by $\phi_f$), we utilize the activation of inputting the answer $A$ with the question $P$ as the correct activation $a_c = a([A; P])$ and provide the loss function as:

$$L(\phi_f) = \| v_P - f(a([A; P])) \|. \tag{10}$$

However, directly leveraging Eq. (9) to initialize context is computational costly as it requires all the context activations in the context pool. Therefore, we distill a lightweight $\mathcal{F}(\cdot)$ (parameterized by $\phi_F$) which directly projects the initial context into the problem space through the following loss function:

$$L(\phi_F) = \| \mathcal{F}([I_i^b; P]) - f(a([I_i^b; P])) \|. \tag{11}$$

Then, we provide the final context initialization as:

$$\boldsymbol{I}^b = \{I_1^b, \ldots, I_N^b\} = \arg\min_{\boldsymbol{I}^b} \left\{ \min_{\omega} \| \sum_{i=1}^{N} \omega_i \mathcal{F}([I_i^b; P]) - v_P \| \right\}. \tag{12}$$

Eq. (12) provides a computational efficient context initialization (as illustrated in Fig. 23) that aligns with Eq. (7) to encourage orthogonality of selected context activations, thereby providing diverse reasoning perspectives and expanding the search space for solutions.

## 5.2 Context Evolution

As discussed in Section 5.1, context initialization aims to select contexts with diverse perspectives to solve the problem. Nevertheless, these individual contexts address the problem from a single viewpoint while lacking explicit instructions on how to incorporate perspectives provided by other LLMs. Directly using them often results in inconsistent reasoning across LLMs due to misaligned inter-LLM guidance. To address this, we iteratively refine the instruction to integrate collaborative instructions that guide the LLMs more effectively toward generating the desired solution.

### 5.2.1 Evaluate the Contribution of Contexts

To achieve efficient context evolution, it is crucial to quantify the contribution of different contexts (refer to a utility function). A natural solution is to assign utilities only at the final round. However, this leads to inefficient and unstable training process due to its sparsity Liu et al. (2025). On the other hand, current methods typically use correctness as the only criterion (Zelikman et al., 2022; Rafailov et al., 2023; Guo et al., 2025). Albeit with promising performance in single-LLM question answering, this simple criterion ignores the complex dependency among LLMs with different contexts in MAD. As a result, an LLM that provides crucial insights which enable others to reach the correct answer may still receive a low utility, simply because it fails to produce the correct final answer itself. To tackle these challenges, we introduce a novel round-wise criterion to evaluate the contribution of $C_i^t$ in multi-agent interaction:

$$\max_{j \in [N]} \left\{ -\alpha \| C_i^t - C_i^b \| - \| a(C_i^t) - a(C_j^t) \| \right\}, \tag{13}$$

where $\alpha \geq 0$ is a weighting parameter and $\| a(C_i^t) - a(C_j^t) \|$ represents the activation difference between contexts.

***Remarks.*** Of note, if removing the second term of Eq. (13), the context generation reduces to that of prompt engineering: $\min_{I_{1:N}^t} \sum_{i \in [N]} \alpha \| C_i^t - C_i^b \|$, which utilize fixed instructions in contexts to LLMs throughout the discussion (Liu et al., 2024; Lu et al., 2024). As mentioned in Section 1, this easily results in inconsistency across LLMs. Thus, we introduce $\| a(C_i^t) - a(C_j^t) \|$ into the objective to encourage LLMs to remain aligned with each other.

This design addresses the two aforementioned challenges. First, this round-wise criterion provides denser feedback, mitigating the inefficiency and instability caused by sparse final-round utilities.

Second, the activation-based alignment term captures inter-LLM dependencies, thereby encouraging them to offer guidance on how to integrate others' responses.

Applying Eq. (13) as the criterion for LLM $i$ requires the contexts of all other LLMs. However, since these contexts are simultaneously being optimized, a bias inevitably arises as each LLM is updated based on stale snapshots of others' contexts. Such outdated information introduces a biased utility, which in turn hinders consistency and degrades the quality of the learned contexts. To overcome this issue, we decouple the inter-LLM dependencies by designing an alternative per-LLM criterion:

$$-\alpha\|C_i^t - C_i^b\| - \|a([I_i^t, P]) - a([X_i^{t-1}, P])\|. \tag{14}$$

In Lemma C.1, we can prove that the summation of Eq. (14) over all LLMs serves as an upper bound of the summation of Eq. (13) over all LLMs. Intuitively, this criterion conveys a clear meaning: the first term, $-\alpha\|C_i^t - C_i^b\|$, preserves the fundamental problem-solving capability endowed by the initialization, while the second term, $-\|a([I_i^t, P]) - a([X_i^{t-1}, P])\|$, enforces consistency between an LLM's current instruction and its own previous response. This local consistency constraint implicitly aligns all LLMs, because when every LLM evolves its context in a temporally coherent way, the divergence among them is gradually reduced. Hence, the collective effect of all LLMs following this rule is that their contexts evolve coherently across rounds, progressively shrinking activation differences and ultimately driving the MAD toward consistency in their final answers.

To intuitively show the tightness of the bound, consider the case where $I_i^t = X_i^{t-1}$ holds, i.e., the instruction of each LLM in the next round equals its response in the previous round. That means $a(I_i^t) = a(X_i^{t-1})$. Given that all LLMs receive the same context, the second term of both Eq. (13) and Eq. (14) becomes zero. In this case, the two formulations coincide.

### 5.2.2 MULTI-ROUND CONTEXT EVOLVING

We slightly abuse notation by denoting $a(I_i^t) = a([I_i^t; P])$ and $a(X_i^{t-1}) = a([X_i^{t-1}; P])$, and provide the multi-round context evolution objective by accumulating contributions as follows:

$$\sum_{t=1}^{T} \max_{I_i^t} \left\{ -\alpha\|C_i^t - C_i^b\| - \|a(I_i^t) - a(X_i^{t-1})\| \right\}. \tag{15}$$

However, selecting the optimal value for the weight $\alpha$ is non-trivial, as it must be carefully tuned for the evolving discussion process. To alleviate the need for manual tuning, we recast Eq. (15) into a constrained optimization problem, where the context adjustment $\|I_i^t - I_i^b\|$ is treated as a constraint:

$$\min_{I_i^1}\left(\|a(I_i^1) - a(X_i^0)\| + \min_{I_i^2}\left(\|a(I_i^2) - a(X_i^1)\| + \cdots + \min_{I_i^T}\|a(I_i^T) - a(X_i^{T-1})\|\right)\right)$$
$$\text{s.t. } \|C_i^t - C_i^b\| \le \beta, \quad \forall t, i. \tag{16}$$

According to the detailed derivation in Appendix E, we prove that taking the dual of this problem recovers Eq. (15) and yields an auxiliary update for the dual variable $\alpha$. Consequently, the optimization of Problem (16) can be implemented using an approximate dual gradient descent procedure, alternating gradient updates of $L(\theta_i)$ and $L(\alpha_i)$.

$$L(\theta_i) = \|a(\mathcal{G}_{\theta_i}(P, I_i^b, \bar{X}_i^{t-1})) - a(X_i^{t-1})\| + \alpha\|C_i^t - C_i^b\|$$
$$L(\alpha_i) = \alpha_i\big(\beta - \|\mathcal{G}_{\theta_i}(P, I_i^b, \bar{X}_i^{t-1}) - C_i^b\|\big). \tag{17}$$

At the beginning of the discussion, the initial contexts are intentionally diverse to encourage multi-perspective reasoning of the problem. At this stage, when the answers differ greatly, $\alpha$ decreases rapidly, thereby weakening the constraint on the distance between the generated contexts and their initial contexts. This guides the generation of contexts towards promoting faster convergence on a unified solution. As the discussion progresses and the LLMs gradually reach agreement, $\alpha$ will be kept at a certain level. This adjustment prevents premature consensus and supports a richer, more comprehensive final solution by shifting the contexts' focus toward preserving multiple perspectives and exploring nuanced differences.

Overall, we name our proposed algorithm `M2CL`, with its pseudocode detailed in Appendix F.4.

Table 1: Accuracy (%) on different datasets. The number of LLMs is 4 for all dataset. We exhibit the performance advantage with `BoN` and highlight the  best  result.

| Model | Method | MMLU | MATH | GPQA | Code | ALFWorld | SciWorld | GAIA | PDDL |
|---|---|---|---|---|---|---|---|---|---|
| Qwen-7B | Single | 61.2$_{\downarrow13.0}$ | 12.9$_{\downarrow12.0}$ | 20.2$_{\downarrow16.2}$ | 51.2$_{\downarrow11.3}$ | 23.8$_{\downarrow7.7}$ | 25.2$_{\downarrow10.1}$ | 15.6$_{\downarrow5.5}$ | 21.0$_{\downarrow5.3}$ |
| | BoN | 74.2$_{\uparrow0.0}$ | 24.9$_{\uparrow0.0}$ | 36.4$_{\uparrow0.0}$ | 62.5$_{\uparrow0.0}$ | 31.5$_{\uparrow0.0}$ | 35.3$_{\uparrow0.0}$ | 21.1$_{\uparrow0.0}$ | 26.3$_{\uparrow0.0}$ |
| | Debate | 71.1$_{\downarrow3.1}$ | 19.9$_{\downarrow5.0}$ | 28.7$_{\downarrow7.7}$ | 60.0$_{\downarrow2.5}$ | 30.6$_{\downarrow0.9}$ | 32.2$_{\downarrow3.1}$ | 21.0$_{\downarrow0.1}$ | 24.8$_{\downarrow1.5}$ |
| | DyLAN | 74.3$_{\uparrow0.1}$ | 26.7$_{\uparrow1.8}$ | 35.4$_{\downarrow1.0}$ | 63.4$_{\uparrow0.9}$ | 29.8$_{\downarrow1.7}$ | 29.4$_{\downarrow5.9}$ | 18.4$_{\downarrow2.7}$ | 23.4$_{\downarrow2.9}$ |
| | GPTSwarm | 76.3$_{\uparrow2.1}$ | 26.2$_{\uparrow1.3}$ | 35.6$_{\downarrow0.8}$ | 62.7$_{\uparrow0.2}$ | 29.9$_{\downarrow1.6}$ | 30.8$_{\downarrow4.5}$ | 20.6$_{\downarrow0.5}$ | 24.5$_{\downarrow1.8}$ |
| | MacNet | 71.5$_{\downarrow2.7}$ | 21.4$_{\downarrow3.5}$ | 30.8$_{\downarrow5.6}$ | 59.3$_{\downarrow3.2}$ | 33.6$_{\uparrow2.1}$ | 37.0$_{\uparrow1.7}$ | 21.5$_{\uparrow0.4}$ | 29.5$_{\uparrow3.2}$ |
| | M2CL (ours) | 92.5$_{\uparrow18.3}$ | 47.8$_{\uparrow22.9}$ | 66.1$_{\uparrow29.7}$ | 80.3$_{\uparrow17.8}$ | 39.9$_{\uparrow8.4}$ | 45.3$_{\uparrow10.0}$ | 33.6$_{\uparrow12.5}$ | 34.7$_{\uparrow8.4}$ |
| Qwen-14B | Single | 67.2$_{\downarrow12.5}$ | 21.6$_{\downarrow6.2}$ | 21.2$_{\downarrow11.6}$ | 56.7$_{\downarrow12.7}$ | 30.1$_{\downarrow7.4}$ | 31.3$_{\downarrow10.3}$ | 18.7$_{\downarrow6.9}$ | 25.7$_{\downarrow2.8}$ |
| | BoN | 79.7$_{\uparrow0.0}$ | 27.8$_{\uparrow0.0}$ | 32.8$_{\uparrow0.0}$ | 69.4$_{\uparrow0.0}$ | 37.5$_{\uparrow0.0}$ | 41.6$_{\uparrow0.0}$ | 25.6$_{\uparrow0.0}$ | 28.5$_{\uparrow0.0}$ |
| | Debate | 77.2$_{\downarrow2.5}$ | 27.4$_{\downarrow0.4}$ | 30.3$_{\downarrow2.5}$ | 66.9$_{\downarrow2.5}$ | 36.8$_{\downarrow0.7}$ | 39.4$_{\downarrow2.2}$ | 26.4$_{\uparrow0.8}$ | 30.0$_{\uparrow1.5}$ |
| | DyLAN | 86.8$_{\uparrow7.1}$ | 31.2$_{\uparrow3.4}$ | 39.4$_{\uparrow6.6}$ | 76.6$_{\uparrow7.2}$ | 34.8$_{\downarrow2.7}$ | 36.0$_{\downarrow5.6}$ | 22.8$_{\downarrow2.8}$ | 28.0$_{\downarrow0.5}$ |
| | GPTSwarm | 87.0$_{\uparrow7.3}$ | 31.3$_{\uparrow3.5}$ | 38.7$_{\uparrow5.9}$ | 75.9$_{\uparrow6.5}$ | 34.9$_{\downarrow2.6}$ | 35.9$_{\downarrow5.7}$ | 25.0$_{\downarrow0.6}$ | 28.4$_{\downarrow0.1}$ |
| | MacNet | 78.9$_{\downarrow0.8}$ | 28.7$_{\uparrow0.9}$ | 31.9$_{\downarrow0.9}$ | 70.4$_{\uparrow1.0}$ | 39.2$_{\uparrow1.7}$ | 46.0$_{\uparrow4.4}$ | 32.2$_{\uparrow6.6}$ | 32.7$_{\uparrow4.2}$ |
| | M2CL (ours) | 93.7$_{\uparrow14.0}$ | 51.7$_{\uparrow23.9}$ | 66.2$_{\uparrow33.4}$ | 91.1$_{\uparrow21.7}$ | 48.2$_{\uparrow10.7}$ | 56.1$_{\uparrow14.5}$ | 42.0$_{\uparrow16.4}$ | 43.0$_{\uparrow14.5}$ |
| Qwen-72B | Single | 72.5$_{\downarrow11.7}$ | 31.6$_{\downarrow19.4}$ | 34.9$_{\downarrow11.0}$ | 59.1$_{\downarrow13.1}$ | 48.2$_{\downarrow9.3}$ | 50.4$_{\downarrow11.7}$ | 31.1$_{\downarrow10.1}$ | 41.0$_{\downarrow10.0}$ |
| | BoN | 84.2$_{\uparrow0.0}$ | 51.0$_{\uparrow0.0}$ | 45.9$_{\uparrow0.0}$ | 72.2$_{\uparrow0.0}$ | 57.5$_{\uparrow0.0}$ | 62.1$_{\uparrow0.0}$ | 41.2$_{\uparrow0.0}$ | 51.0$_{\uparrow0.0}$ |
| | Debate | 82.7$_{\downarrow1.5}$ | 48.4$_{\downarrow2.6}$ | 43.4$_{\downarrow2.5}$ | 69.1$_{\downarrow3.1}$ | 60.4$_{\uparrow2.9}$ | 65.2$_{\uparrow3.1}$ | 42.9$_{\uparrow1.7}$ | 49.1$_{\downarrow1.9}$ |
| | DyLAN | 91.5$_{\uparrow7.3}$ | 63.1$_{\uparrow12.1}$ | 51.6$_{\uparrow5.7}$ | 80.4$_{\uparrow8.2}$ | 55.1$_{\downarrow2.4}$ | 58.0$_{\downarrow4.1}$ | 40.4$_{\downarrow0.8}$ | 45.5$_{\downarrow5.5}$ |
| | GPTSwarm | 91.5$_{\uparrow7.3}$ | 64.7$_{\uparrow13.7}$ | 52.4$_{\uparrow6.5}$ | 79.6$_{\uparrow7.4}$ | 56.8$_{\downarrow0.7}$ | 60.2$_{\downarrow1.9}$ | 40.3$_{\downarrow0.9}$ | 49.7$_{\downarrow1.3}$ |
| | MacNet | 83.8$_{\downarrow0.4}$ | 52.9$_{\uparrow1.9}$ | 46.2$_{\uparrow0.3}$ | 70.5$_{\downarrow1.7}$ | 61.8$_{\uparrow4.3}$ | 68.4$_{\uparrow6.3}$ | 46.4$_{\uparrow5.2}$ | 53.7$_{\uparrow2.7}$ |
| | M2CL (ours) | 95.1$_{\uparrow10.9}$ | 72.5$_{\uparrow21.5}$ | 78.9$_{\uparrow33.0}$ | 90.7$_{\uparrow18.5}$ | 79.0$_{\uparrow21.5}$ | 88.9$_{\uparrow26.8}$ | 67.2$_{\uparrow26.0}$ | 70.5$_{\uparrow19.5}$ |

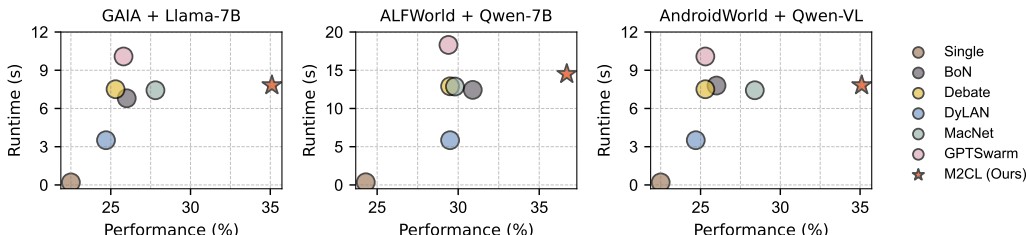

Figure 3: Performance versus runtime under different settings. Circles closer to the lower-left corner indicate higher efficiency.

# 6 EXPERIMENT

In this section, we conduct experiments to evaluate the performance of `M2CL` by answering the following research questions:

Q1. How does `M2CL` perform compared to existing methods across various benchmarks, especially in complex agentic tasks?

Q2. How does the performance of MAD scale with the number of LLMs?

Q3. How is the performance affected by factors such as context constraint and components such as context initialization and context evolution?

Q4. How do contexts promote consensus, and are they transferable to other models?

## 6.1 EXPERIMENTAL SETUP

**Dataset.** We run experiments with 3 domains including 9 datasets: **1) LLM reasoning**, including MMLU (Hendrycks et al., 2021), MATH (Hendrycks et al., 2021), and GPQA (Rein et al., 2023), HumanEval (Chen et al., 2021). **2) Embodied Agentic**, including ALFWorld Shridhar et al. (2021), SciWorld Wang et al. (2022), GAIA (Mialon et al., 2024), and PDDL Chang et al. (2024). **3) Mobile GUI** AndroidWorld Rawles et al. (2025). Details on datasets can be found in Appendix F.1.

**Baselines.** We evaluate our method against six strong baseline methods: **1)** `Single execution`, querying a single LLM to solve the task. **2)** `Best-of-N`, querying a single LLM N times and sampling the most correct answer. **3)** `Debate` (Du et al., 2023), a multi-agent framework where LLMs discuss their responses and reasoning processes over multiple rounds. **4)** `DyLAN` (Liu et al., 2024), a multi-agent framework where LLMs score each other and collaborate dynamically. **5)** `GPTSwarm` (Zhuge et al., 2024), a multi-agent framework that refines LLM prompts and improves

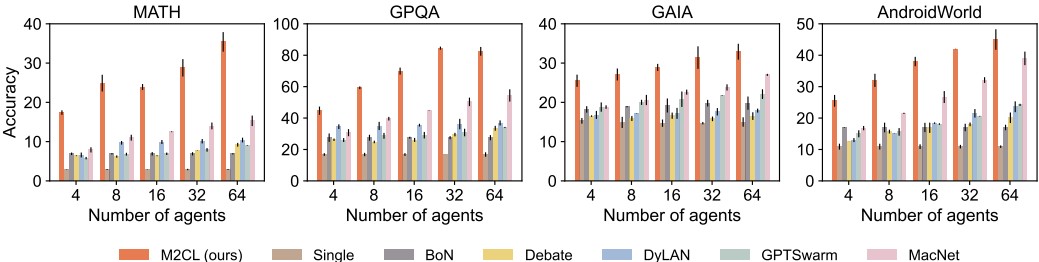

Figure 4: Performance of varying the numbers of LLMs. Uncertainty intervals depict standard deviation over three seeds.

LLM orchestration by changing their connectivity. **6)** `MacNet` (Jiang et al., 2023), a recent multi-LLM framework where LLMs are invoked between LLM interactions to provide actionable instructions to the next LLM based on the previous LLM's outputs.

**Reproducibility.** All details of our experiments are provided in the appendices in terms of the tasks, network architectures, hyperparameters, etc. We conduct experiment on two series of LLM (Llama-2 (Touvron et al., 2023) and Qwen-2.5 (Yang et al., 2024a)) and three model sizes each for LLM reasoning, and Qwen2.5-VL (Bai et al., 2025) for GUI reasoning. All the experiments are run on Ubuntu 22.04.4 LTS with 8 NVIDIA H800 GPUs.

## 6.2 EXPERIMENTAL RESULTS

**Comparative results.** To answer the first question, we evaluate `M2CL`'s performance across all datasets, with varying base models and number of LLMs (ranging from 4 to 64). We select Qwen series models and 4 LLMs participating in Table 1 and provide full results in Appendices G.2 and G.3. We find `M2CL` consistently outperforms baselines in all 9 datasets, often by a significant margin in terms of performance. Of note, `BoN` outperforms most of the baselines especially in complex multi-round agentic tasks, revealing the drawback of fixed contexts, which, despite expanding the exploration space, do not converge and thus hinder LLMs from achieving true cooperative reasoning. In contrast, `M2CL` can adapt contexts and enhance the relevance of responses and questions while ensuring creativity, indicating that `M2CL` well avoids that LLMs with different contexts easily influence each other and successfully brings LLMs into cooperation by reaching a consensus through discussion.

In addition, we evaluate the efficiency of `M2CL` by visualizing the performance versus runtime. As shown in Fig. 3, `M2CL` consistently delivers the highest performance improvement (more than $20\%$) while maintaining a modest increase in runtime (less than $10\%$). This clearly demonstrates the efficiency of `M2CL` due to its lightweight context generator.

**Multi-agent scaling law.** To answer the second question, we run experiments with varying numbers of LLMs (ranging from 4 to 64). The data and parameter setup adhere to that of Table 2. We present selected results in Fig. 4 and full results in Tables 4 to 8 and Figs. 5 to 10 of Appendix G.2. Scaling our method reveals a more efficient scaling law as the performance grows logarithmically before saturation and improves faster than baselines. We speculate this arises because collaborative instructions in our generated contexts enable genuine inter-LLM cooperation, thereby unleashing the multidimensional reasoning capabilities of the MAD.

**Context constraint.** To answer the third question, we vary the context constraint $\beta$ from 0 to 10 and run experiments across all datasets. Full results are shown in Figs. 12 to 16 of Appendix G.4. The results clearly indicate that as $\beta$ increases, there is an initial improvement in performance; once it reaches a sufficiently large value, performance tends to drop. A strict (refers to a smaller $\beta$) context constraint leads to generated contexts getting closer to the initial contexts, thereby causing discussion inconsistency. On the other hand, a loose context constraint leads to naive consistency, where LLMs tend to generate the same answers, resulting in insufficient creativity.

**Ablation studies.** We assess the effect of key components by ablating them on all datasets under the same setting. *1) Importance of context initialization. 2) Importance of tuning $\alpha$. 3) Importance of context evolution.* As illustrated in Tables 11 to 13, LLMs struggle to specialize and coordinate

efficiently, whereas context initialization enables them to acquire high-impact contexts, significantly enhancing the foundational capabilities of MAD. Without tuning $\alpha$ during discussion rounds, LLMs tend to reach an agreement in the first round, leading to responses that lack creativity and diversity, which ultimately reduces problem-solving ability. Without context evolution, LLMs lack collaborative guidance considering previous responses, which prevents them from effectively leveraging the outputs of other LLMs.

**Discrepancy intensity.** To answer the fourth question, we explored the change of Discrepancy intensity ($\max_{i,j \in [N]} \|a_i - a_j\|^2$) with rounds to verify whether the generated context can help the LLMs gradually reach agreement with the discussion. Complete results are shown in Figs. 17 to 21, of Appendix G.5. We find that discrepancy intensity from `M2CL` decreases faster than other methods, corroborating its efficacy and superiority in converging the search space of multiple LLMs.

**Transferability of contexts.** To answer the fourth question, we also conduct experiments on transferring contexts directly into stronger LLMs to verify whether the generated contexts have better interpretation ability and efficacy. Full results are depicted in Table 10 of Appendix G.6. The results show that the transferred contexts deliver consistent improvement, indicating that the trained context generator can directly adapt to a wide range of models without additional retraining.

# 7 LIMITATION AND DISCUSSION

In this paper, we propose a novel context learning method, designed to expand LLMs' horizons and reach consensus in MAD. By initializing and adjusting LLMs' contexts based on the problem and discussion states, our method significantly improves problem-solving capabilities across diverse benchmarks while maintaining computational efficiency. A limitation of `M2CL` is the MAD framework in which diversity is brought about by the number of LLMs with heterogeneous characteristics which is computationally inefficient. An avenue for future work is to enable LLMs to truly capture the sub-tasks they are interested in or excel at.

## ACKNOWLEDGMENTS

This research was supported in part by the National Natural Science Foundation of China under Grants 62572496 and 62402269, the Shenzhen Science and Technology Program under Grant JCYJ20250604175500001, and the Young Elite Scientist Sponsorship Program by CAST under Contract ZB2025-218.

## ETHICS STATEMENT

This work advances the field of large language model (LLM) research by introducing a context learning method for multi-agent discussion, demonstrating significant improvements in accuracy, diversity, and consensus across multiple benchmarks.

However, the broader implications of deploying such systems warrant careful consideration. Multi-agent discussion, while powerful, may inadvertently propagate biases introduced during context initialization or amplify errors during consensus-building. These risks are particularly critical in high-stakes domains like legal, financial, and healthcare applications, where erroneous outputs could lead to significant ethical, social, or economic consequences.

## REPRODUCIBILITY STATEMENT

The full algorithmic details of `M2CL` are presented in the main paper, with additional implementation details and hyperparameter settings included in Section 6. We provide source code at `https://github.com/HansenHua/M2CL-ICLR26`, which includes scripts for training, evaluation, and reproducing all reported experiments. We provide checkpoints at `https://huggingface.co/hansenhua/M2CL`. For theoretical results, we include complete proofs in Appendices B to D.

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

## A  USAGE OF LLMS

We use large language models (LLMs) solely as an assistive tool for polishing the writing and improving clarity of exposition. All content generated by LLMs was carefully reviewed, verified, and, where necessary, revised by the authors. The authors take full responsibility for the correctness and integrity of the final manuscript.

## B  PROOF OF THEOREM 4.1

*Proof.* Define the optimal $\omega^* = \arg\min_\omega \|a_c - \sum_{i=1}^N \omega_i a(C_i^b)\|$ and its corresponding activation $a = \sum_{i=1}^N \omega_i^* a(C_i^b)$. Then, we can derive the upper bound of the activation difference using the triangle inequality:

$$\sum_{i=1}^N \|a_c - a(C_i^t)\| \le \sum_{i=1}^N \|a(C_i^t) - a\| + N\|a - a_c\|. \tag{18}$$

Using the triangle inequality, we can bound the first term in Eq. (18):

$$\sum_{i=1}^N \|a(C_i^t) - a\| = \sum_{i=1}^N \|\sum_{j=1}^N \omega_j^*(a(C_i^t) - a(C_j^b))\|$$

$$\le \sum_{i=1}^N \sum_{j=1}^N |\omega_j^*| \cdot \|a(C_i^t) - a(C_j^b)\| \le \sum_{i=1}^N \sum_{j=1}^N \|a(C_i^t) - a(C_j^b)\|$$

$$\le \sum_{i=1}^N \sum_{j=1}^N \|a(C_i^t) - a(C_j^t)\| + \sum_{i=1}^N \sum_{j=1}^N \|a(C_j^t) - a(C_j^b)\|$$

$$\le \sum_{i=1}^N \sum_{j=1}^N \|a(C_i^t) - a(C_j^t)\| + \sum_{i=1}^N \sum_{j=1}^N L_a\|C_j^t - C_j^b\|. \tag{19}$$

Then, we bound the second term in Eq. (18):

$$\|a - a_c\| \le \|\sum_{i=1}^N \omega_i^* a(C_i^t) - a_c\| = \|\sum_{i=1}^N \omega_i^* a(C_i^b) + \sum_{i=1}^N \omega_i^*(a(C_i^t) - a(C_i^b)) - a_c\|$$

$$\le \min_\omega \|a_c - \sum_{i=1}^N \omega_i a(C_i^b)\| + \sum_{i=1}^N \|\omega_i^*(a(C_i^t) - a(C_i^b))\|$$

$$\le \min_\omega \|a_c - \sum_{i=1}^N \omega_i a(C_i^b)\| + \sum_{i=1}^N \|a(C_i^t) - a(C_i^b)\|$$

$$\le \min_\omega \|a_c - \sum_{i=1}^N \omega_i a(C_i^b)\| + L_a \sum_{i=1}^N \|C_i^t - C_i^b\|. \tag{20}$$

In the first line, we add and subtract $\sum_{i=1}^N a(C_i^b)$. In the second line, we use the triangle inequality and the definition of $\omega^*$. In the third line, we scale $\omega^*$ to 1. Finally, we utilize the smoothness property of the activation function to bound the second term Substituting into Eq. (18), we can derive:

$$\sum_{i=1}^N \|a_c - a(C_i^t)\| \le \sum_{i=1}^N \left( \sum_{j=1}^N \|a(C_i^t) - a(C_j^t)\| + (N+1)L_a\|C_i^t - C_i^b\| \right)$$

$$+ N \min_\omega \|a_c - \sum_{i=1}^N \omega_i a(C_i^b)\|, \tag{21}$$

$\square$

## C  DECOUPLED CRITERION FUCNTION

**Lemma C.1.** *Under the assumption of one-block transformer, LLMs' activation diversity can be bound by the activation difference between their instructions and responses.*

$$\|a(C_i^t) - a(C_j^t)\| \le \|a(I_i^t) - a(X_i^t)\| + \|a(I_j^t) - a(X_j^t)\| + 18 L_V n N \exp(2\rho^2). \quad (22)$$

*Proof.* We first decompose the activation of context into a combination of its components:

$$a(C_i^t) = a(C_i^t) - [a(I_i^t) + \sum_{k \ne i}^{N} a(X_k^{t-1}) - (N-1)a(P)] + [a(I_i^t) + \sum_{k \ne i}^{N} a(X_k^{t-1}) - (N-1)a(P)]$$

$$a(C_j^t) = a(C_j^t) - [a(I_j^t) + \sum_{k \ne j}^{N} a(X_k^{t-1}) - (N-1)a(P)] + [a(I_j^t) + \sum_{k \ne j}^{N} a(X_k^{t-1}) - (N-1)a(P)]. \quad (23)$$

By using Lemma D.3, we can derive the upper bound of the activation difference as:

$$\|a(C_i^t) - a(C_j^t)\|$$

$$\le \|a(C_i^t) - [a(I_i^t) + \sum_{k \ne i}^{N} a(X_k^{t-1}) - (N-1)a(P)]\| + \|a(I_i^t) - a(X_i^{t-1})\|$$

$$+ \|a(C_j^t) - [a(I_j^t) + \sum_{k \ne j}^{N} a(X_k^{t-1}) - (N-1)a(P)]\| + \|a(I_j^t) - a(X_j^{t-1})\|$$

$$\le \|a(I_i^t) - a(X_i^{t-1})\| + \|a(I_j^t) - a(X_j^{t-1})\| + 18 L_V n N \exp(2\rho^2). \quad (24)$$

$\square$

## D  USEFUL LEMMA

**Lemma D.1.** *Suppose $\|W_V X\| \le L_V$ and $\|(W_K X)^T W_Q X\| \le \rho^2$, the difference between the activation of softmax attention $a(X)$ and linear attention $a'(X)$ can be bounded by:*

$$\|a(X) - a'(X)\| \le 3 L_V n \exp(2\rho^2) \quad (25)$$

*where $a(X) = W_V X \mathrm{softmax}\big(\frac{(W_K X)^T W_Q X}{\sqrt{d}}\big)$ and $a'(X) = W_V X (W_K X)^T W_Q X$.*

*Proof.* Define $\Delta = D^{-1} \exp(S) - \exp(S)$, where $S = (W_K X)^T W_Q X$ and $D = diag(\langle \exp(S), \mathbf{1}_n \rangle)$. Then, we can derive the upper bound of each element of $\Delta$.

$$\Delta_{ij} = \frac{\exp(S_{ij})}{D_{ii}} - \exp(S_{ij}) \le \exp(S_{ij})(|\frac{1}{D_{ii}}| + 1) \le \exp(\rho^2)(1 + \exp(\rho^2)) \le 2 \exp(2\rho^2) \quad (26)$$

Summing them up, we can derive:

$$\|D^{-1} \exp(S) - \exp(S)\| \le \|D^{-1} \exp(S) - \exp(S)\|_F = \sqrt{\sum_{i=1}^{n} \sum_{j=1}^{n} \Delta_{ij}^2} \le 2n \exp(2\rho^2) \quad (27)$$

Define $\delta = \exp(S) - S$, where $S = (W_K X)^T W_Q X$. Then, we derive the upper bound of $\delta$ by finding the upper bound of each element of $\delta$.

$$\delta_{ij} = \exp(S_{ij}) - S_{ij} \le \max\{\exp(\rho^2) - \rho^2, \rho^2 + \exp(-\rho^2)\} \le \exp(\rho^2) \quad (28)$$

Summing them up, we can derive:

$$\| \exp(S) - S \| \leq \| \exp(S) - S \|_F = \sqrt{\sum_{i=1}^{n} \sum_{j=1}^{n} \exp(S_{ij}) - S_{ij}} \leq n \exp(\rho^2) \qquad (29)$$

After bounding $\| D^{-1} \exp(S) - \exp(S) \|$ and $\| \exp(S) - S \|$, we can derive the upper bound of $\| D^{-1} \exp(S) - S \|$ by basic algebra as:

$$
\begin{aligned}
&\| D^{-1} \exp(S) - S \| \\
=& \| D^{-1} \exp(S) - \exp(S) + \exp(S) - S \| \\
\leq& \| D^{-1} \exp(S) - \exp(S) \| + \| \exp(S) - S \| \\
\leq& 2n \exp(2\rho^2) + n \exp(\rho^2) \\
\leq& 3n \exp(2\rho^2)
\end{aligned}
\qquad (30)
$$

Therefore, we derive the difference between the activation of softmax attention and linear attention as:

$$
\begin{aligned}
&\| W_V X \mathrm{softmax}\Big( \frac{(W_K X)^T W_Q X}{\sqrt{d}} \Big) - W_V X (W_K X)^T W_Q X \| \\
\leq& \| W_V X \| \cdot \| \mathrm{softmax}\Big( \frac{(W_K X)^T W_Q X}{\sqrt{d}} \Big) - (W_K X)^T W_Q X \| \\
\leq& L_V \cdot \| D^{-1} \exp(S) - S \| \\
\leq& 3 L_V n \exp(2\rho^2)
\end{aligned}
\qquad (31)
$$

$\square$

**Lemma D.2.** *Define the activation of linear attention:*

$$
\begin{aligned}
a'(Y) &\doteq W_V [Y, P] (W_K [Y, P])^T W_Q Y \\
a'(P) &\doteq W_V P (W_K P)^T W_Q P,
\end{aligned}
\qquad (32)
$$

*then the activation of a long prompt $Y = [Y_1, Y_2, \ldots, Y_N]$ can be derived by the activation of its component:*

$$a'(Y) = \sum_{i=1}^{N} a'(Y_i) - (N-1)a'(P). \qquad (33)$$

*Proof.*

$$
\begin{aligned}
a'(Y) &= W_V [Y_1, Y_2, \ldots, Y_N, P] \Big( W_K [Y_1, Y_2, \ldots, Y_N, P] \Big)^T W_Q P \\
&= W_V [Y_1, Y_2, \ldots, Y_N, P][Y_1, Y_2, \ldots, Y_N, P]^T W_K^T W_Q P \\
&= W_V \Big( \sum_{i=1}^{N} Y_i Y_i^T + P P^T \Big) W_K^T W_Q P \\
&= W_V \Big[ \sum_{i=1}^{N} (Y_i Y_i^T + P P^T) - (N-1) P P^T \Big] W_K^T W_Q P \\
&= \sum_{i=1}^{N} W_V (Y_i Y_i^T + P P^T) W_K^T W_Q P - W_V (N-1) P P^T W_K^T W_Q P \\
&= \sum_{i=1}^{N} W_V [Y_i, P] (W_K [Y_i, P])^T W_Q P - (N-1) W_K P (W_K P)^T W_Q P \\
&= \sum_{i=1}^{N} a'(Y_i) - (N-1) a'(P).
\end{aligned}
\qquad (34)
$$

$\square$

**Lemma D.3.** *We derive the difference between the activation of a long prompt $Y$ and a combination of the activations of its components $Y = [Y_1, Y_2, \ldots, Y_N, P]$ as:*

$$\|a(Y) - \sum_{i=1}^{N} a(Y_i) + (N-1)a(P)\| \leq 9L_V nN \exp(\rho^2). \tag{35}$$

*Proof.* Denote the activation of linear attention as $a'$, we can complete the proof by using Lemma D.2 to simplify the relationship of linear attention and Lemma D.1 to bound the difference:

$$\|a(Y) - \sum_{i=1}^{N} a(Y_i) + (N-1)a(P)\|$$

$$= \|a(Y) - a'(Y) + a'(Y) - [\sum_{i=1}^{N} a'(Y_i) - (N-1)a'(P)]$$

$$+ [\sum_{i=1}^{N} a'(Y_i) - (N-1)a'(P)] - [\sum_{i=1}^{N} a(Y_i) - (N-1)a(P)]\|$$

$$\leq \|a(Y) - a'(Y)\| + \sum_{i=1}^{N} \|a'(Y_i) - a(Y_i)\| + (N-1)\|a'(P) - a(P)\|$$

$$\leq \exp(2\rho^2)[3L_V nN + 3L_V Nn + 3L_V(N-1)n]$$

$$\leq 9L_V nN \exp(2\rho^2) \tag{36}$$

$\square$

**Lemma D.4.** *If $\|S\| \leq \rho$, we have $\langle \exp(S), \mathbf{1}_n \rangle \geq \exp(-\rho)$*

*Proof.*

$$\langle \exp(S), \mathbf{1}_n \rangle = \sum_{i=1}^{n} \exp(S_i) \geq \min_{i \in [n]} \exp(S_i) \geq \min_{i \in [n]} \exp(-|S_i|)$$

$$= \exp(-\max_{i \in [n]} |S_i|) = \exp(-\|S\|_\infty) \geq \exp(-\|S\|_2) \geq \exp(-\rho) \tag{37}$$

$\square$

## E  MULTI-ROUND CONTEXT LEARNING

First, we define:

$$h(I_i^t) \doteq \beta - \|I_i^t - I_i^b\| \tag{38}$$

$$f(I_i^t) \doteq \begin{cases} \|a(I_i^t) - a(X_i^{t-1})\|, & \text{if } h(I_i^t) \geq 0 \\ +\infty, & \text{otherwise.} \end{cases} \tag{39}$$

To solve the minimization optimization with inequality constraint, we can construct a Lagrangian expression with a Lagrange multiplier $\alpha_T$:

$$\text{minimize } f(I_i^t) \text{ s.t. } h(I_i^t) \geq 0 \Leftrightarrow L(I_i^t, \alpha_t) = f(I_i^t) - \alpha_t h(I_i^t). \tag{40}$$

The optimization changes to:

$$\min_{I_i^t} f(I_i^t) = \max_{\alpha_t \geq 0} \min_{I_i^t} L(I_i^t, \alpha_t). \tag{41}$$

Therefore, to minimize $f(I_i^t)$, the dual problem is listed as below. Note that to make sure $\min_{I_i^t} f(I_i^t)$ is properly minimized and would not become $+\infty$, the constraint has to be satisfied.

$$
\begin{aligned}
&\min_{I_i^t} \left[ \|a(I_i^t) - a(X_i^{t-1})\| \right] \\
&= \min_{I_i^t} f(I_i^t) \\
&= \max_{\alpha_t \geq 0} \min_{I_i^t} L(I_i^t, \alpha_t) \\
&= \max_{\alpha_t \geq 0} \min_{I_i^t} f(I_i^t) - \alpha_t h(I_i^t) \\
&= \max_{\alpha_T \geq 0} \min_{I_i^t} \left[ \|a(I_i^t) - a(X_i^{t-1})\| \right] - \alpha_t \left( \beta - \|I_i^t - I_i^b\| \right) \\
&= \max_{\alpha_T \geq 0} \min_{I_i^t} \left[ \|a(I_i^t) - a(X_i^{t-1})\| - \alpha_T \beta + \alpha_t \|I_i^t - I_i^b\| \right].
\end{aligned}
\tag{42}
$$

Define the Q-function for this multi-round multi-agent collaboration as:

$$
Q(a(I_i^t), a(X_i^{t-1})) = \|a(I_i^t) - a(X_i^{t-1})\| + \min_{I_i^t} \|a(I_i^{t+1}) - a(X_i^t)\| + \alpha_{t+1}\|I_i^{t+1,*} - I_i^b\|.
\tag{43}
$$

Here, $I_i^{t+1,*}$ denotes the optimal $I_i^{t+1}$. Therefore, the expected return is as follows, when we take one round further back to the round $T-1$:

$$
\begin{aligned}
&\min_{I_i^{T-1}} \left( \|a(I_i^{T-1}) - a(X_i^{T-2})\| + \min_{I_i^t} \|a(I_i^t) - a(X_i^{T-1})\| \right) \\
&= \min_{I_i^{T-1}} \left( Q(a(I_i^{T-1}), a(X_i^{T-2})) - \alpha_T^*\|I_i^{T,*} - I_i^b\| \right) \\
&= \max_{\alpha_{T-1}>0} \min_{I_i^{T-1}} \left( Q(a(I_i^{T-1}), a(X_i^{T-2})) - \alpha_{T-1}\left( \beta - \|I_i^{T-1} - I_i^b\| \right) - \alpha_T^*\|I_i^{T,*} - I_i^b\| \right) \\
&= \max_{\alpha_{T-1}>0} \min_{I_i^{T-1}} \left( Q(a(I_i^{T-1}), a(X_i^{T-2})) - \alpha_{T-1}\beta + \alpha_{T-1}\|I_i^{T-1} - I_i^b\| \right) - \alpha_T^*\|I_i^{T,*} - I_i^b\|.
\end{aligned}
\tag{44}
$$

Similar to the previous round,

$$
\begin{aligned}
I_i^t &= \arg\min_{I_i^t} \left[ \|a(I_i^t) - a(X_i^{t-1})\| - \alpha_t \beta + \alpha_t \|I_i^t - I_i^b\| \right] \\
\alpha_t^* &= \arg\max_{\alpha_t \geq 0} \left[ \alpha_T\|I_i^t - I_i^b\| - \alpha_t\beta \right].
\end{aligned}
\tag{45}
$$

By repeating this process, we can learn the optimal temperature parameter in every round. Hence, the loss functions for $\theta$ and $\alpha$ are as follows:

$$
\begin{aligned}
L(\theta_i^t) &= \|a(I_i^t) - a(X_i^{t-1})\| + \alpha_t\|I_i^t - I_i^b\| \\
L(\alpha) &= \alpha\left( \beta - \|I_i^t - I_i^b\| \right).
\end{aligned}
\tag{46}
$$

# F  EXPERIMENT SETUP

## F.1  DATASETS

We evaluate our method on three areas with 7 datasets which are widely used in prior studies (Dubey et al., 2024; Qian et al., 2025). We elaborate on what follows.

- MMLU (Hendrycks et al., 2021), a comprehensive benchmark covering diverse subjects and difficulty levels, designed to test world knowledge and logical reasoning through multiple-choice questions.
- MATH (Hendrycks et al., 2021), a dataset of challenging competition-level math problems requiring multi-step symbolic reasoning and advanced problem-solving ability.
- GPQA (Rein et al., 2023), a benchmark of graduate-level multiple-choice science questions that assesses deep domain knowledge and reasoning under uncertainty.
- HumanEval (Chen et al., 2021), a widely recognized benchmark for function-level code generation, designed to evaluate fundamental programming skills.
- ALFWorld (Shridhar et al., 2021), a text-based embodied environment featuring household tasks, where agents navigate and interact with objects via natural language commands.
- SciWorld (Wang et al., 2022), a text-based embodied environment for interactive science tasks, requiring agents to navigate rooms, conduct experiments, and perform procedural reasoning.
- GAIA (Mialon et al., 2024), a benchmark of real-world question answering tasks that integrate knowledge retrieval, reasoning, and multi-step tool use.
- PDDL (Chang et al., 2024), an environment comprising diverse strategic games, where agents must employ PDDL expressions to plan and execute complex tasks.
- AndroidWorld (Rawles et al., 2025), an environment with 116 dynamic tasks across 20 real-world Android apps, designed to evaluate mobile agents' capabilities in app navigation and system-level control.

We use $20\%$ of the questions to construct the training dataset and the rest as testing dataset.

## F.2  BASELINES

We test our method against six baselines. We implement them based on their publicly available implementations.

- *Single Execution* (`Single`), querying a single LLM to solve the task.
- *Best-of-N sampling* (`BoN`), querying a single LLM N times and sampling the most correct answer. We set N to 32, as increasing N further does not yield additional performance benefits and aligns closely with the number of LLM calls used for multi-agent methods.
- *Discussion* (Du et al., 2023), a multi-agent framework in which LLMs are assigned predefined distinct contexts and iteratively exchange their reasoning processes as additional prompts over multiple rounds, before producing a final answer via majority voting.
- *Dynamic LLM-Powered Agent Network* (Liu et al., 2024) (`DyLAN`), a discussion-style framework which incorporates an LLM selection algorithm based on an unsupervised metric, namely the Agent Importance Score, which identifies the most contributive LLMs through a preliminary trial tailored to the specific task.
- *GPTSwarm* (Zhuge et al., 2024), formalizing a swarm of autonomous agents as computational graphs, with nodes as manually-customized functions and edges facilitating information flow, adaptively optimizing node prompts and modifying graph connectivity during collective reasoning.
- *Multi-LLM Collaboration Network* (Qian et al., 2025) (`MacNet`), a representative framework for decentralized and scalable multi-LLM systems. It introduces edge agents that mediate interactions by generating actionable instructions for the next agent based on the outputs of the previous one.

## F.3  IMPLEMENTATION DETAILS

The context pool is constructed using GPT-4o, where we prompt it to generate a large collection of high-quality initial contexts across diverse domains, including mathematics, science, coding, and

embodied reasoning, and various domain-specific sub-contexts are included in each domain. This ensures that the pool provides a broad coverage of reasoning perspectives and is shared across all tasks.

The context generators are implemented by T5-small model (Raffel et al., 2020) for all the tasks. The dimension of the generated context vectors is 512. The learning rates for the context generators and $\alpha$ are both $1e - 4$. We use $20\%$ of the questions to train the context initialization and generator.

Table 2: Hyperparameters (identical across datasets).

| Hyperparameter | Value |
|---|---|
| Sentence vector dimension | 512 |
| Optimizer | Adam |
| Batchsize | 32 |
| Learning rate of context | 1e-4 |
| Learning rate of $\alpha$ | 1e-4 |
| Maximum rounds for discussion | 8 |
| Training rounds | 100 |
| Size of context pool | 100 |

We implement our code using Pytorch 2.3.0, built upon the open-source parameters of the llama2, Qwen2.5, and Qwen2.5-VL. Models provided at `https://huggingface.co/meta-llama` and `https://huggingface.co/Qwen`. All the experiments are run on Ubuntu 22.04.4 LTS with 8 NVIDIA H800 GPUs.

### F.4 PSEUDOCODE OF M2CL

We present the pseudocode of training `M2CL` in Algorithm 1. It begins with training the context initialization, then the context generators are trained along with the weight $\alpha$ during the discussion.

---

**Algorithm 1:** Pseudocode of `M2CL`

---

1 Initialize parameters $\phi_f$, $\phi_F$, $\{\theta_i\}_{i=1}^N$, and $\{\alpha_i\}_{i=1}^N$;
2 **for** *each question* **do**
3 $\quad$ $\phi_f \leftarrow \phi_f - \eta_f \nabla L(\phi_f), \quad \phi_F \leftarrow \phi_F - \eta_F \nabla L(\phi_F)$;
4 **end**
5 **for** *each epoch* **do**
6 $\quad$ Obtain initial contexts $\{I_i^b\}_{i=1}^N$ via Eq. (12);
7 $\quad$ **for** *each round $t$* **do**
8 $\quad\quad$ **for** $i = 1$ *to* $N$ **do**
9 $\quad\quad\quad$ Generate instructions $I_i^t$ via Eq. (3) and obtain responses $X_i^t$ via Eq. (4);
10 $\quad\quad\quad$ $\theta_i \leftarrow \theta_i - \eta_\theta \nabla L(\theta_i), \quad \alpha_i \leftarrow \alpha_i - \eta_\alpha \nabla L(\alpha_i)$;
11 $\quad\quad$ **end**
12 $\quad$ **end**
13 **end**

---

## G  ADDITIONAL RESULTS

### G.1  COMPARISON WITH MORE MODELS

To further investigate `M2CL`'s performance, we compare it with `best-of-N` using stronger close-sourced LLMs, including Qwen2.5-Max and GPT-4. As shown in Table 3, llama series models perform worse when fewer LLMs participate but achieve higher accuracy as the number of LLMs increases, demonstrating that `M2CL` can collaborate weaker models to achieve comparable performance.

Table 3: Accuracy using different base models on different datasets. We exhibit the performance advantage with `Qwen-max` and highlight the best result.

|  | Model | MMLU | MATH | GPQA | Code | ALFWorld | SciWorld | GAIA | PDDL |
|---|---|---|---|---|---|---|---|---|---|
| n=1 | Qwen-max | $86.5_{\uparrow0.0}$ | $43.6_{\uparrow0.0}$ | $35.7_{\uparrow0.0}$ | $65.9_{\uparrow0.0}$ | $73.1_{\uparrow0.0}$ | $68.7_{\uparrow0.0}$ | $61.5_{\uparrow0.0}$ | $76.4_{\uparrow0.0}$ |
|  | GPT-4 | $84.3_{\downarrow2.2}$ | $41.1_{\downarrow2.5}$ | $33.3_{\downarrow2.4}$ | $62.7_{\downarrow3.2}$ | $70.2_{\downarrow2.9}$ | $65.4_{\downarrow3.3}$ | $58.7_{\downarrow2.8}$ | $73.5_{\downarrow2.9}$ |
|  | Llama-7B | $45.3_{\downarrow41.2}$ | $2.9_{\downarrow40.7}$ | $16.8_{\downarrow18.9}$ | $15.8_{\downarrow50.1}$ | $22.5_{\downarrow50.6}$ | $20.1_{\downarrow48.6}$ | $15.3_{\downarrow46.2}$ | $24.2_{\downarrow52.2}$ |
|  | Llama-13B | $54.8_{\downarrow31.7}$ | $3.9_{\downarrow39.7}$ | $25.0_{\downarrow10.7}$ | $23.7_{\downarrow42.2}$ | $28.5_{\downarrow44.6}$ | $24.5_{\downarrow44.2}$ | $18.6_{\downarrow42.9}$ | $30.4_{\downarrow46.0}$ |
|  | Llama-70B | $68.9_{\downarrow17.6}$ | $31.6_{\downarrow12.0}$ | $27.6_{\downarrow8.1}$ | $35.5_{\downarrow30.4}$ | $46.1_{\downarrow27.0}$ | $40.2_{\downarrow28.5}$ | $29.9_{\downarrow31.6}$ | $48.4_{\downarrow28.0}$ |
| n=4 | Qwen-max | $87.5_{\uparrow0.0}$ | $44.2_{\uparrow0.0}$ | $39.2_{\uparrow0.0}$ | $67.5_{\uparrow0.0}$ | $74.0_{\uparrow0.0}$ | $69.8_{\uparrow0.0}$ | $62.5_{\uparrow0.0}$ | $77.5_{\uparrow0.0}$ |
|  | GPT-4 | $86.4_{\downarrow1.1}$ | $43.9_{\downarrow0.3}$ | $38.8_{\downarrow0.4}$ | $67.0_{\downarrow0.5}$ | $72.8_{\downarrow1.2}$ | $68.0_{\downarrow1.8}$ | $61.0_{\downarrow1.5}$ | $76.0_{\downarrow1.5}$ |
|  | Llama-7B | $59.1_{\downarrow28.4}$ | $17.4_{\downarrow26.8}$ | $44.7_{\uparrow5.5}$ | $32.9_{\downarrow34.6}$ | $35.1_{\downarrow38.9}$ | $33.0_{\downarrow36.8}$ | $25.5_{\downarrow37.0}$ | $33.1_{\downarrow44.4}$ |
|  | Llama-13B | $86.9_{\downarrow0.6}$ | $23.4_{\downarrow20.8}$ | $58.9_{\uparrow19.7}$ | $47.5_{\downarrow20.0}$ | $42.9_{\downarrow31.1}$ | $39.9_{\downarrow29.9}$ | $30.5_{\downarrow32.0}$ | $40.9_{\downarrow36.6}$ |
|  | Llama-70B | $95.6_{\uparrow8.1}$ | $40.3_{\downarrow3.9}$ | $78.7_{\uparrow39.5}$ | $70.6_{\uparrow3.1}$ | $69.5_{\downarrow4.5}$ | $65.5_{\downarrow4.3}$ | $49.6_{\downarrow13.0}$ | $66.5_{\downarrow11.0}$ |
| n=8 | Qwen-max | $88.5_{\uparrow0.0}$ | $45.0_{\uparrow0.0}$ | $40.5_{\uparrow0.0}$ | $68.8_{\uparrow0.0}$ | $75.0_{\uparrow0.0}$ | $71.0_{\uparrow0.0}$ | $63.5_{\uparrow0.0}$ | $79.0_{\uparrow0.0}$ |
|  | GPT-4 | $86.5_{\downarrow2.0}$ | $44.6_{\downarrow0.4}$ | $38.9_{\downarrow1.6}$ | $67.1_{\downarrow1.7}$ | $73.2_{\downarrow1.8}$ | $69.5_{\downarrow1.5}$ | $61.0_{\downarrow2.5}$ | $76.0_{\downarrow3.0}$ |
|  | Llama-7B | $63.8_{\downarrow24.7}$ | $24.9_{\downarrow20.1}$ | $59.3_{\uparrow18.8}$ | $38.8_{\downarrow30.0}$ | $37.2_{\downarrow37.8}$ | $36.1_{\downarrow34.9}$ | $27.1_{\downarrow36.4}$ | $34.7_{\downarrow44.3}$ |
|  | Llama-13B | $92.7_{\uparrow4.2}$ | $23.0_{\downarrow22.0}$ | $70.7_{\uparrow30.2}$ | $55.4_{\downarrow13.4}$ | $46.1_{\downarrow28.9}$ | $44.7_{\downarrow26.3}$ | $33.6_{\downarrow29.9}$ | $43.8_{\downarrow35.2}$ |
|  | Llama-70B | $95.4_{\uparrow6.9}$ | $43.5_{\downarrow1.5}$ | $87.6_{\uparrow47.1}$ | $81.0_{\uparrow12.2}$ | $78.9_{\uparrow3.9}$ | $76.2_{\uparrow5.2}$ | $58.7_{\downarrow4.8}$ | $73.6_{\downarrow5.4}$ |
| n=16 | Qwen-max | $88.5_{\uparrow0.0}$ | $45.0_{\uparrow0.0}$ | $40.5_{\uparrow0.0}$ | $68.8_{\uparrow0.0}$ | $75.0_{\uparrow0.0}$ | $71.0_{\uparrow0.0}$ | $63.5_{\uparrow0.0}$ | $79.0_{\uparrow0.0}$ |
|  | GPT-4 | $87.9_{\downarrow0.6}$ | $44.8_{\downarrow0.2}$ | $40.1_{\downarrow0.4}$ | $68.5_{\downarrow0.3}$ | $74.3_{\downarrow0.7}$ | $70.5_{\downarrow0.5}$ | $63.0_{\downarrow0.5}$ | $78.2_{\downarrow0.8}$ |
|  | Llama-7B | $71.5_{\downarrow17.0}$ | $23.9_{\downarrow21.1}$ | $69.9_{\uparrow29.4}$ | $48.3_{\downarrow20.5}$ | $39.8_{\downarrow35.2}$ | $38.4_{\downarrow32.6}$ | $28.9_{\downarrow34.6}$ | $37.0_{\downarrow42.0}$ |
|  | Llama-13B | $94.5_{\uparrow6.0}$ | $32.7_{\downarrow12.3}$ | $88.5_{\uparrow48.0}$ | $66.1_{\downarrow2.7}$ | $48.7_{\downarrow26.3}$ | $47.1_{\downarrow23.9}$ | $36.2_{\downarrow27.3}$ | $45.5_{\downarrow33.5}$ |
|  | Llama-70B | $93.9_{\uparrow5.4}$ | $51.5_{\uparrow6.5}$ | $91.3_{\uparrow50.8}$ | $97.2_{\uparrow28.4}$ | $84.9_{\uparrow9.9}$ | $81.7_{\uparrow10.7}$ | $61.5_{\downarrow2.0}$ | $78.1_{\downarrow0.9}$ |
| n=32 | Qwen-max | $89.0_{\uparrow0.0}$ | $45.5_{\uparrow0.0}$ | $41.0_{\uparrow0.0}$ | $69.5_{\uparrow0.0}$ | $75.5_{\uparrow0.0}$ | $71.5_{\uparrow0.0}$ | $64.0_{\uparrow0.0}$ | $79.5_{\uparrow0.0}$ |
|  | GPT-4 | $89.1_{\uparrow0.1}$ | $45.4_{\downarrow0.1}$ | $41.5_{\uparrow0.5}$ | $69.7_{\uparrow0.2}$ | $75.0_{\downarrow0.5}$ | $71.0_{\downarrow0.5}$ | $63.5_{\downarrow0.5}$ | $78.5_{\downarrow1.0}$ |
|  | Llama-7B | $79.1_{\downarrow9.9}$ | $28.8_{\downarrow16.7}$ | $84.5_{\uparrow43.5}$ | $56.5_{\downarrow13.0}$ | $42.5_{\downarrow33.0}$ | $40.7_{\downarrow30.8}$ | $31.4_{\downarrow32.6}$ | $38.9_{\downarrow40.6}$ |
|  | Llama-13B | $95.8_{\uparrow6.8}$ | $41.4_{\downarrow4.1}$ | $94.7_{\uparrow53.7}$ | $75.6_{\uparrow6.1}$ | $52.2_{\downarrow23.3}$ | $50.7_{\downarrow20.8}$ | $38.5_{\downarrow25.5}$ | $48.0_{\downarrow31.5}$ |
|  | Llama-70B | $97.0_{\uparrow8.0}$ | $54.5_{\uparrow9.0}$ | $95.1_{\uparrow54.1}$ | $93.7_{\uparrow24.2}$ | $88.5_{\uparrow13.0}$ | $86.0_{\uparrow14.5}$ | $65.7_{\uparrow1.7}$ | $79.8_{\uparrow0.3}$ |
| n=64 | Qwen-max | $89.5_{\uparrow0.0}$ | $46.0_{\uparrow0.0}$ | $41.5_{\uparrow0.0}$ | $70.0_{\uparrow0.0}$ | $76.0_{\uparrow0.0}$ | $72.0_{\uparrow0.0}$ | $64.5_{\uparrow0.0}$ | $80.0_{\uparrow0.0}$ |
|  | GPT-4 | $88.5_{\downarrow1.0}$ | $45.4_{\downarrow0.6}$ | $41.8_{\uparrow0.3}$ | $70.5_{\uparrow0.5}$ | $75.2_{\downarrow0.8}$ | $71.5_{\downarrow0.5}$ | $64.0_{\downarrow0.5}$ | $79.2_{\downarrow0.8}$ |
|  | Llama-7B | $81.5_{\downarrow8.0}$ | $35.4_{\downarrow10.6}$ | $82.5_{\uparrow41.0}$ | $60.6_{\downarrow9.4}$ | $44.2_{\downarrow31.8}$ | $42.9_{\downarrow29.1}$ | $32.9_{\downarrow31.6}$ | $40.9_{\downarrow39.1}$ |
|  | Llama-13B | $96.8_{\uparrow7.3}$ | $40.0_{\downarrow6.0}$ | $93.1_{\uparrow51.6}$ | $82.5_{\uparrow12.5}$ | $54.8_{\downarrow21.2}$ | $52.8_{\downarrow19.2}$ | $40.3_{\downarrow24.2}$ | $49.8_{\downarrow30.2}$ |
|  | Llama-70B | $96.3_{\uparrow6.8}$ | $58.0_{\uparrow12.0}$ | $96.9_{\uparrow55.4}$ | $92.0_{\uparrow22.0}$ | $90.8_{\uparrow14.8}$ | $88.9_{\uparrow16.9}$ | $68.2_{\uparrow3.7}$ | $82.2_{\uparrow2.2}$ |

## G.2 NUMBER OF LLMs

We evaluate `M2CL`'s performance with varying number of LLMs, ranging from 4 to 64. The comparative results are shown in Tables 4 to 8 and Figs. 5 to 9.

**Summary of key findings.** The results show that `M2CL` significantly improves MAD performance, consistently surpassing existing methods by $20\% - 50\%$, particularly in complex tasks like math and tool-using. This highlights its ability to tackle intricate reasoning tasks. `M2CL` also exhibits a more effective multi-agent scaling law, where performance consistently improves as the number of LLMs increases to 64, especially in agentic domains where a larger amount of LLMs enhances problem-solving accuracy and efficiency.

Table 4: Accuracy with varying number of LLMs on different datasets. The number of LLMs is 4 for all datasets. We exhibit the performance advantage with `BoN` and highlight the best result.

| Model | Method | MMLU | MATH | GPQA | Code | ALFWorld | SciWorld | GAIA | PDDL |
|---|---|---|---|---|---|---|---|---|---|
| Llama-7B | Single | $45.3_{\downarrow 5.2}$ | $2.9_{\downarrow 4.0}$ | $16.8_{\downarrow 10.7}$ | $15.8_{\downarrow 6.0}$ | $22.5_{\downarrow 2.9}$ | $20.1_{\downarrow 4.8}$ | $15.3_{\downarrow 2.9}$ | $24.2_{\downarrow 2.0}$ |
| | BoN | $50.5_{\uparrow 0.0}$ | $6.9_{\uparrow 0.0}$ | $27.5_{\uparrow 0.0}$ | $21.8_{\uparrow 0.0}$ | $25.4_{\uparrow 0.0}$ | $24.9_{\uparrow 0.0}$ | $18.2_{\uparrow 0.0}$ | $26.2_{\uparrow 0.0}$ |
| | Debate | $47.3_{\downarrow 3.2}$ | $6.4_{\downarrow 0.5}$ | $26.2_{\downarrow 1.3}$ | $19.8_{\downarrow 2.0}$ | $25.3_{\downarrow 0.1}$ | $23.8_{\downarrow 1.1}$ | $16.5_{\downarrow 1.7}$ | $26.2_{\uparrow 0.0}$ |
| | DyLAN | $48.1_{\downarrow 2.4}$ | $6.6_{\downarrow 0.3}$ | $34.5_{\uparrow 7.0}$ | $22.2_{\uparrow 0.4}$ | $24.7_{\downarrow 0.7}$ | $21.8_{\downarrow 3.1}$ | $16.8_{\downarrow 1.4}$ | $25.5_{\downarrow 0.7}$ |
| | GPTSwarm | $48.0_{\downarrow 2.5}$ | $5.8_{\downarrow 1.1}$ | $26.0_{\downarrow 1.5}$ | $18.9_{\downarrow 2.9}$ | $25.8_{\uparrow 0.4}$ | $25.1_{\uparrow 0.2}$ | $18.8_{\uparrow 0.6}$ | $27.9_{\uparrow 1.7}$ |
| | MacNet | $50.8_{\uparrow 0.3}$ | $7.9_{\uparrow 1.0}$ | $30.7_{\uparrow 3.2}$ | $24.0_{\uparrow 2.2}$ | $27.8_{\uparrow 2.4}$ | $26.5_{\uparrow 1.6}$ | $18.8_{\uparrow 0.6}$ | $29.3_{\uparrow 3.1}$ |
| | M2CL (ours) | $59.1_{\uparrow 8.6}$ | $17.4_{\uparrow 10.5}$ | $44.7_{\uparrow 17.2}$ | $32.9_{\uparrow 11.1}$ | $35.1_{\uparrow 9.7}$ | $33.0_{\uparrow 8.1}$ | $25.5_{\uparrow 7.3}$ | $33.1_{\uparrow 6.9}$ |
| Llama-14B | Single | $54.8_{\downarrow 16.2}$ | $3.9_{\downarrow 4.9}$ | $25.0_{\downarrow 12.7}$ | $23.7_{\downarrow 8.3}$ | $28.5_{\downarrow 2.7}$ | $24.5_{\downarrow 7.2}$ | $18.6_{\downarrow 3.2}$ | $30.4_{\downarrow 2.8}$ |
| | BoN | $71.0_{\uparrow 0.0}$ | $8.8_{\uparrow 0.0}$ | $37.7_{\uparrow 0.0}$ | $32.0_{\uparrow 0.0}$ | $31.2_{\uparrow 0.0}$ | $31.7_{\uparrow 0.0}$ | $21.8_{\uparrow 0.0}$ | $33.2_{\uparrow 0.0}$ |
| | Debate | $64.5_{\downarrow 6.5}$ | $8.4_{\downarrow 0.4}$ | $36.3_{\downarrow 1.4}$ | $27.9_{\downarrow 4.1}$ | $30.8_{\downarrow 0.4}$ | $29.3_{\downarrow 2.4}$ | $19.8_{\downarrow 2.0}$ | $31.9_{\downarrow 1.3}$ |
| | DyLAN | $69.7_{\downarrow 1.3}$ | $10.3_{\uparrow 1.5}$ | $46.5_{\uparrow 8.8}$ | $32.5_{\uparrow 0.5}$ | $31.3_{\uparrow 0.1}$ | $28.7_{\downarrow 3.0}$ | $19.4_{\downarrow 2.4}$ | $30.4_{\downarrow 2.8}$ |
| | MacNet | $70.3_{\downarrow 0.7}$ | $9.2_{\uparrow 0.4}$ | $40.9_{\uparrow 3.2}$ | $35.2_{\uparrow 3.2}$ | $31.5_{\uparrow 0.3}$ | $31.8_{\uparrow 0.1}$ | $23.4_{\uparrow 1.6}$ | $35.3_{\uparrow 2.1}$ |
| | MacNet | $70.3_{\downarrow 0.7}$ | $9.2_{\uparrow 0.4}$ | $40.9_{\uparrow 3.2}$ | $35.2_{\uparrow 3.2}$ | $31.5_{\uparrow 0.3}$ | $31.8_{\uparrow 0.1}$ | $23.4_{\uparrow 1.6}$ | $35.3_{\uparrow 2.1}$ |
| | M2CL (ours) | $86.9_{\uparrow 15.9}$ | $23.4_{\uparrow 14.6}$ | $58.9_{\uparrow 21.2}$ | $47.5_{\uparrow 15.5}$ | $42.9_{\uparrow 11.7}$ | $39.9_{\uparrow 8.2}$ | $30.5_{\uparrow 8.7}$ | $40.9_{\uparrow 7.7}$ |
| Llama-70B | Single | $68.9_{\downarrow 14.1}$ | $31.6_{\downarrow 3.2}$ | $27.6_{\downarrow 32.0}$ | $35.5_{\downarrow 12.5}$ | $46.1_{\downarrow 4.8}$ | $40.2_{\downarrow 6.5}$ | $29.9_{\downarrow 2.8}$ | $48.4_{\downarrow 4.4}$ |
| | BoN | $83.0_{\uparrow 0.0}$ | $34.8_{\uparrow 0.0}$ | $59.6_{\uparrow 0.0}$ | $48.0_{\uparrow 0.0}$ | $50.9_{\uparrow 0.0}$ | $46.7_{\uparrow 0.0}$ | $32.7_{\uparrow 0.0}$ | $52.8_{\uparrow 0.0}$ |
| | Debate | $74.8_{\downarrow 8.2}$ | $29.8_{\downarrow 5.0}$ | $50.2_{\downarrow 9.4}$ | $43.2_{\downarrow 4.8}$ | $50.4_{\downarrow 0.5}$ | $47.8_{\uparrow 1.1}$ | $32.4_{\downarrow 0.3}$ | $52.9_{\uparrow 0.1}$ |
| | DyLAN | $82.1_{\downarrow 0.9}$ | $31.5_{\downarrow 3.3}$ | $53.8_{\downarrow 5.8}$ | $44.5_{\downarrow 3.5}$ | $49.3_{\downarrow 1.6}$ | $45.1_{\downarrow 1.6}$ | $31.5_{\downarrow 1.2}$ | $50.3_{\downarrow 2.5}$ |
| | GPTSwarm | $81.4_{\downarrow 1.6}$ | $31.5_{\downarrow 3.3}$ | $54.8_{\downarrow 4.8}$ | $43.7_{\downarrow 4.3}$ | $51.0_{\uparrow 0.1}$ | $45.2_{\downarrow 1.5}$ | $32.5_{\downarrow 0.2}$ | $51.5_{\downarrow 1.3}$ |
| | MacNet | $72.9_{\downarrow 10.1}$ | $32.7_{\downarrow 2.1}$ | $49.1_{\downarrow 10.5}$ | $40.8_{\downarrow 7.2}$ | $52.7_{\uparrow 1.8}$ | $55.9_{\uparrow 9.2}$ | $36.5_{\uparrow 3.8}$ | $57.0_{\uparrow 4.2}$ |
| | M2CL (ours) | $95.6_{\uparrow 12.6}$ | $40.3_{\uparrow 5.5}$ | $78.7_{\uparrow 19.1}$ | $70.6_{\uparrow 22.6}$ | $69.5_{\uparrow 18.6}$ | $65.5_{\uparrow 18.8}$ | $49.6_{\uparrow 16.9}$ | $66.5_{\uparrow 13.7}$ |
| Qwen-7B | Single | $61.2_{\downarrow 13.0}$ | $12.9_{\downarrow 12.0}$ | $20.2_{\downarrow 16.2}$ | $51.2_{\downarrow 11.3}$ | $24.3_{\downarrow 6.6}$ | $25.9_{\downarrow 6.6}$ | $15.3_{\downarrow 4.7}$ | $21.1_{\downarrow 4.1}$ |
| | BoN | $74.2_{\uparrow 0.0}$ | $24.9_{\uparrow 0.0}$ | $36.4_{\uparrow 0.0}$ | $62.5_{\uparrow 0.0}$ | $30.9_{\uparrow 0.0}$ | $32.5_{\uparrow 0.0}$ | $20.0_{\uparrow 0.0}$ | $25.2_{\uparrow 0.0}$ |
| | Debate | $69.3_{\downarrow 4.9}$ | $17.9_{\downarrow 7.0}$ | $28.7_{\downarrow 7.7}$ | $54.5_{\downarrow 8.0}$ | $29.5_{\downarrow 1.4}$ | $31.4_{\downarrow 1.1}$ | $20.6_{\uparrow 0.6}$ | $24.3_{\downarrow 0.9}$ |
| | DyLAN | $74.2_{\uparrow 0.0}$ | $23.3_{\downarrow 1.6}$ | $34.2_{\downarrow 2.2}$ | $62.5_{\uparrow 0.0}$ | $29.5_{\downarrow 1.4}$ | $28.3_{\downarrow 4.2}$ | $19.0_{\downarrow 1.0}$ | $23.0_{\downarrow 2.2}$ |
| | GPTSwarm | $74.5_{\uparrow 0.3}$ | $24.1_{\downarrow 0.8}$ | $37.2_{\uparrow 0.8}$ | $60.4_{\downarrow 2.1}$ | $29.4_{\downarrow 1.5}$ | $29.5_{\downarrow 3.0}$ | $20.0_{\uparrow 0.0}$ | $23.9_{\downarrow 1.3}$ |
| | MacNet | $68.0_{\downarrow 6.2}$ | $20.3_{\downarrow 4.6}$ | $25.0_{\downarrow 11.4}$ | $57.1_{\downarrow 5.4}$ | $29.8_{\downarrow 1.1}$ | $33.8_{\uparrow 1.3}$ | $20.5_{\uparrow 0.5}$ | $26.6_{\uparrow 1.4}$ |
| | M2CL (ours) | $88.7_{\uparrow 14.5}$ | $40.8_{\uparrow 15.9}$ | $58.4_{\uparrow 22.0}$ | $76.8_{\uparrow 14.3}$ | $36.7_{\uparrow 5.8}$ | $41.7_{\uparrow 9.2}$ | $30.2_{\uparrow 10.2}$ | $33.0_{\uparrow 7.8}$ |
| Qwen-14B | Single | $67.2_{\downarrow 12.5}$ | $21.6_{\downarrow 6.2}$ | $21.2_{\downarrow 11.6}$ | $56.7_{\downarrow 12.7}$ | $30.2_{\downarrow 6.0}$ | $31.7_{\downarrow 7.7}$ | $19.2_{\downarrow 4.6}$ | $26.4_{\downarrow 2.4}$ |
| | BoN | $79.7_{\uparrow 0.0}$ | $27.8_{\uparrow 0.0}$ | $32.8_{\uparrow 0.0}$ | $69.4_{\uparrow 0.0}$ | $36.2_{\uparrow 0.0}$ | $39.4_{\uparrow 0.0}$ | $23.8_{\uparrow 0.0}$ | $28.8_{\uparrow 0.0}$ |
| | Debate | $71.7_{\downarrow 8.0}$ | $24.7_{\downarrow 3.1}$ | $24.8_{\downarrow 8.0}$ | $62.7_{\downarrow 6.7}$ | $35.6_{\downarrow 0.6}$ | $38.5_{\downarrow 0.9}$ | $25.5_{\uparrow 1.7}$ | $29.9_{\uparrow 1.1}$ |
| | DyLAN | $77.5_{\downarrow 2.2}$ | $26.9_{\downarrow 0.9}$ | $31.9_{\downarrow 0.9}$ | $67.9_{\downarrow 1.5}$ | $33.6_{\downarrow 2.6}$ | $34.9_{\downarrow 4.5}$ | $22.7_{\downarrow 1.1}$ | $27.8_{\downarrow 1.0}$ |
| | GPTSwarm | $80.1_{\uparrow 0.4}$ | $26.9_{\downarrow 0.9}$ | $33.0_{\uparrow 0.2}$ | $68.1_{\downarrow 1.3}$ | $34.6_{\downarrow 1.6}$ | $34.8_{\downarrow 4.6}$ | $24.5_{\uparrow 0.7}$ | $28.3_{\downarrow 0.5}$ |
| | MacNet | $71.2_{\downarrow 8.5}$ | $24.6_{\downarrow 3.2}$ | $27.6_{\downarrow 5.2}$ | $62.9_{\downarrow 6.5}$ | $36.1_{\downarrow 0.1}$ | $41.4_{\uparrow 2.0}$ | $28.6_{\uparrow 4.8}$ | $29.4_{\uparrow 0.6}$ |
| | M2CL (ours) | $91.4_{\uparrow 11.7}$ | $43.6_{\uparrow 15.8}$ | $64.6_{\uparrow 31.8}$ | $86.5_{\uparrow 17.1}$ | $45.0_{\uparrow 8.8}$ | $51.3_{\uparrow 11.9}$ | $37.2_{\uparrow 13.4}$ | $39.7_{\uparrow 10.9}$ |
| Qwen-72B | Single | $72.5_{\downarrow 11.7}$ | $31.6_{\downarrow 19.4}$ | $34.9_{\downarrow 11.0}$ | $59.1_{\downarrow 13.1}$ | $49.3_{\downarrow 7.3}$ | $51.1_{\downarrow 8.7}$ | $31.4_{\downarrow 7.2}$ | $41.8_{\downarrow 8.0}$ |
| | BoN | $84.2_{\uparrow 0.0}$ | $51.0_{\uparrow 0.0}$ | $45.9_{\uparrow 0.0}$ | $72.2_{\uparrow 0.0}$ | $56.6_{\uparrow 0.0}$ | $59.8_{\uparrow 0.0}$ | $38.6_{\uparrow 0.0}$ | $49.8_{\uparrow 0.0}$ |
| | Debate | $76.6_{\downarrow 7.6}$ | $48.0_{\downarrow 3.0}$ | $39.6_{\downarrow 6.3}$ | $66.7_{\downarrow 5.5}$ | $58.3_{\uparrow 1.7}$ | $62.5_{\uparrow 2.7}$ | $40.7_{\uparrow 2.1}$ | $47.8_{\downarrow 2.0}$ |
| | DyLAN | $84.6_{\uparrow 0.4}$ | $61.5_{\uparrow 10.5}$ | $45.5_{\downarrow 0.4}$ | $71.3_{\downarrow 0.9}$ | $54.2_{\downarrow 2.4}$ | $57.8_{\downarrow 2.0}$ | $38.2_{\downarrow 0.4}$ | $45.9_{\downarrow 3.9}$ |
| | GPTSwarm | $84.9_{\uparrow 0.7}$ | $60.9_{\uparrow 9.9}$ | $46.6_{\uparrow 0.7}$ | $68.1_{\downarrow 4.1}$ | $55.2_{\downarrow 1.4}$ | $59.1_{\downarrow 0.7}$ | $39.0_{\uparrow 0.4}$ | $48.8_{\downarrow 1.0}$ |
| | MacNet | $80.3_{\downarrow 3.9}$ | $48.1_{\downarrow 2.9}$ | $41.2_{\downarrow 4.7}$ | $68.1_{\downarrow 4.1}$ | $57.2_{\downarrow 0.6}$ | $64.1_{\uparrow 4.3}$ | $42.0_{\uparrow 3.4}$ | $49.8_{\uparrow 0.0}$ |
| | M2CL (ours) | $93.5_{\uparrow 9.3}$ | $69.3_{\uparrow 18.3}$ | $73.1_{\uparrow 27.2}$ | $88.3_{\uparrow 16.1}$ | $73.9_{\uparrow 17.3}$ | $81.9_{\uparrow 22.1}$ | $60.3_{\uparrow 21.7}$ | $65.3_{\uparrow 15.5}$ |

Table 5: Accuracy with varying number of LLMs on different datasets. The number of LLMs is 8 for all datasets. We exhibit the performance advantage with `BoN` and highlight the best result.

| Model | Method | MMLU | MATH | GPQA | Code | ALFWorld | SciWorld | GAIA | PDDL |
|---|---|---|---|---|---|---|---|---|---|
| Llama-7B | Single | $45.3_{\downarrow5.2}$ | $2.9_{\downarrow4.0}$ | $16.8_{\downarrow10.7}$ | $15.8_{\downarrow6.0}$ | $22.6_{\downarrow3.6}$ | $19.7_{\downarrow5.3}$ | $14.9_{\downarrow4.0}$ | $23.6_{\downarrow3.1}$ |
| | BoN | $50.5_{\uparrow0.0}$ | $6.9_{\uparrow0.0}$ | $27.5_{\uparrow0.0}$ | $21.8_{\uparrow0.0}$ | $26.2_{\uparrow0.0}$ | $25.0_{\uparrow0.0}$ | $18.9_{\uparrow0.0}$ | $26.7_{\uparrow0.0}$ |
| | Debate | $49.5_{\downarrow1.0}$ | $6.2_{\downarrow0.7}$ | $24.8_{\downarrow2.7}$ | $19.2_{\downarrow2.6}$ | $25.9_{\downarrow0.3}$ | $24.7_{\downarrow0.3}$ | $15.9_{\downarrow3.0}$ | $26.9_{\uparrow0.2}$ |
| | DyLAN | $54.2_{\uparrow3.7}$ | $9.7_{\uparrow2.8}$ | $35_{\uparrow7.5}$ | $26_{\uparrow4.2}$ | $25.0_{\downarrow1.2}$ | $22.0_{\downarrow3.0}$ | $17.2_{\downarrow1.7}$ | $25.4_{\downarrow1.3}$ |
| | GPTSwarm | $49.2_{\downarrow1.3}$ | $6.8_{\downarrow0.1}$ | $28.7_{\uparrow1.2}$ | $20.7_{\downarrow1.1}$ | $26.2_{\uparrow0.0}$ | $26.4_{\uparrow1.4}$ | $20.0_{\uparrow1.1}$ | $29.3_{\uparrow2.6}$ |
| | MacNet | $56.5_{\uparrow6.0}$ | $11.0_{\uparrow4.1}$ | $39.6_{\uparrow12.1}$ | $30.0_{\uparrow8.2}$ | $30.4_{\uparrow4.2}$ | $30.2_{\uparrow5.2}$ | $20.6_{\uparrow1.7}$ | $31.7_{\uparrow5.0}$ |
| | M2CL (ours) | $63.8_{\uparrow13.3}$ | $24.9_{\uparrow18.0}$ | $59.3_{\uparrow31.8}$ | $38.8_{\uparrow17.0}$ | $37.2_{\uparrow11.0}$ | $36.1_{\uparrow11.1}$ | $27.1_{\uparrow8.2}$ | $34.7_{\uparrow8.0}$ |
| Llama-14B | Single | $54.8_{\downarrow16.2}$ | $3.9_{\downarrow4.9}$ | $25.0_{\downarrow12.7}$ | $23.7_{\downarrow8.3}$ | $27.8_{\downarrow4.1}$ | $24.1_{\downarrow8.2}$ | $18.6_{\downarrow3.9}$ | $29.4_{\downarrow5.4}$ |
| | BoN | $71.0_{\uparrow0.0}$ | $8.8_{\uparrow0.0}$ | $37.7_{\uparrow0.0}$ | $32.0_{\uparrow0.0}$ | $31.9_{\uparrow0.0}$ | $32.3_{\uparrow0.0}$ | $22.5_{\uparrow0.0}$ | $34.8_{\uparrow0.0}$ |
| | Debate | $66.9_{\downarrow4.1}$ | $8.0_{\downarrow0.8}$ | $34.5_{\downarrow3.2}$ | $30.8_{\downarrow1.2}$ | $31.7_{\downarrow0.2}$ | $30.3_{\downarrow2.0}$ | $20.0_{\downarrow2.5}$ | $32.0_{\downarrow2.8}$ |
| | DyLAN | $82.1_{\uparrow11.1}$ | $12.4_{\uparrow3.6}$ | $46.7_{\uparrow9.0}$ | $37.5_{\uparrow5.5}$ | $31.7_{\downarrow0.2}$ | $29.6_{\downarrow2.7}$ | $19.4_{\downarrow3.1}$ | $31.3_{\downarrow3.5}$ |
| | MacNet | $81.7_{\uparrow10.7}$ | $14.2_{\uparrow5.4}$ | $53.9_{\uparrow16.2}$ | $43.3_{\uparrow11.3}$ | $33.4_{\uparrow1.5}$ | $35.6_{\uparrow3.3}$ | $24.9_{\uparrow2.4}$ | $37.0_{\uparrow2.2}$ |
| | MacNet | $81.7_{\uparrow10.7}$ | $14.2_{\uparrow5.4}$ | $53.9_{\uparrow16.2}$ | $43.3_{\uparrow11.3}$ | $33.4_{\uparrow1.5}$ | $35.6_{\uparrow3.3}$ | $24.9_{\uparrow2.4}$ | $37.0_{\uparrow2.2}$ |
| | M2CL (ours) | $92.7_{\uparrow21.7}$ | $23.0_{\uparrow14.2}$ | $70.7_{\uparrow33.0}$ | $55.4_{\uparrow23.4}$ | $46.1_{\uparrow14.2}$ | $44.7_{\uparrow12.4}$ | $33.6_{\uparrow11.1}$ | $43.8_{\uparrow9.0}$ |
| Llama-70B | Single | $68.9_{\downarrow14.1}$ | $31.6_{\downarrow3.2}$ | $27.6_{\downarrow32.0}$ | $35.5_{\downarrow12.5}$ | $44.1_{\downarrow8.3}$ | $39.1_{\downarrow9.7}$ | $29.1_{\downarrow3.8}$ | $47.1_{\downarrow5.7}$ |
| | BoN | $83.0_{\uparrow0.0}$ | $34.8_{\uparrow0.0}$ | $59.6_{\uparrow0.0}$ | $48.0_{\uparrow0.0}$ | $52.4_{\uparrow0.0}$ | $48.8_{\uparrow0.0}$ | $32.9_{\uparrow0.0}$ | $52.8_{\uparrow0.0}$ |
| | Debate | $82.2_{\downarrow0.8}$ | $34.3_{\downarrow0.5}$ | $58.1_{\downarrow1.5}$ | $47.1_{\downarrow0.9}$ | $51.7_{\downarrow0.7}$ | $50.1_{\uparrow1.3}$ | $32.2_{\downarrow0.7}$ | $53.3_{\uparrow0.5}$ |
| | DyLAN | $93.5_{\uparrow10.5}$ | $37.1_{\uparrow2.3}$ | $72.8_{\uparrow13.2}$ | $56.8_{\uparrow8.8}$ | $50.2_{\downarrow2.2}$ | $46.4_{\downarrow2.4}$ | $31.7_{\downarrow1.2}$ | $49.7_{\downarrow3.1}$ |
| | GPTSwarm | $93.6_{\uparrow10.6}$ | $36.6_{\uparrow1.8}$ | $71.2_{\uparrow11.6}$ | $57.1_{\uparrow9.1}$ | $51.4_{\downarrow1.0}$ | $46.5_{\downarrow2.3}$ | $32.2_{\downarrow0.7}$ | $52.2_{\downarrow0.6}$ |
| | MacNet | $82.0_{\downarrow1.0}$ | $36.9_{\uparrow2.1}$ | $62.2_{\uparrow2.6}$ | $48.8_{\uparrow0.8}$ | $57.3_{\uparrow4.9}$ | $64.0_{\uparrow15.2}$ | $40.7_{\uparrow7.8}$ | $62.6_{\uparrow9.8}$ |
| | M2CL (ours) | $95.4_{\uparrow12.4}$ | $43.5_{\uparrow8.7}$ | $87.6_{\uparrow28.0}$ | $81.0_{\uparrow33.0}$ | $78.9_{\uparrow26.5}$ | $76.2_{\uparrow27.4}$ | $58.7_{\uparrow25.8}$ | $73.6_{\uparrow20.8}$ |
| Qwen-7B | Single | $61.2_{\downarrow13.0}$ | $12.9_{\downarrow12.0}$ | $20.2_{\downarrow16.2}$ | $51.2_{\downarrow11.3}$ | $23.8_{\downarrow7.7}$ | $25.2_{\downarrow10.1}$ | $15.6_{\downarrow5.5}$ | $21.0_{\downarrow5.3}$ |
| | BoN | $74.2_{\uparrow0.0}$ | $24.9_{\uparrow0.0}$ | $36.4_{\uparrow0.0}$ | $62.5_{\uparrow0.0}$ | $31.5_{\uparrow0.0}$ | $35.3_{\uparrow0.0}$ | $21.1_{\uparrow0.0}$ | $26.3_{\uparrow0.0}$ |
| | Debate | $71.1_{\downarrow3.1}$ | $19.9_{\downarrow5.0}$ | $28.7_{\downarrow7.7}$ | $60.0_{\downarrow2.5}$ | $30.6_{\downarrow0.9}$ | $32.2_{\downarrow3.1}$ | $21.0_{\downarrow0.1}$ | $24.8_{\downarrow1.5}$ |
| | DyLAN | $74.3_{\uparrow0.1}$ | $26.7_{\uparrow1.8}$ | $35.4_{\downarrow1.0}$ | $63.4_{\uparrow0.9}$ | $29.8_{\downarrow1.7}$ | $29.4_{\downarrow5.9}$ | $18.4_{\downarrow2.7}$ | $23.4_{\downarrow2.9}$ |
| | GPTSwarm | $76.3_{\uparrow2.1}$ | $26.2_{\uparrow1.3}$ | $35.6_{\downarrow0.8}$ | $62.7_{\uparrow0.2}$ | $29.9_{\downarrow1.6}$ | $30.8_{\downarrow4.5}$ | $20.6_{\downarrow0.5}$ | $24.5_{\downarrow1.8}$ |
| | MacNet | $71.5_{\downarrow2.7}$ | $21.4_{\downarrow3.5}$ | $30.8_{\downarrow5.6}$ | $59.3_{\downarrow3.2}$ | $33.6_{\uparrow2.1}$ | $37.0_{\uparrow1.7}$ | $21.5_{\uparrow0.4}$ | $29.5_{\uparrow3.2}$ |
| | M2CL (ours) | $92.5_{\uparrow18.3}$ | $47.8_{\uparrow22.9}$ | $66.1_{\uparrow29.7}$ | $80.3_{\uparrow17.8}$ | $39.9_{\uparrow8.4}$ | $45.3_{\uparrow10.0}$ | $33.6_{\uparrow12.5}$ | $34.7_{\uparrow8.4}$ |
| Qwen-14B | Single | $67.2_{\downarrow12.5}$ | $21.6_{\downarrow6.2}$ | $21.2_{\downarrow11.6}$ | $56.7_{\downarrow12.7}$ | $30.1_{\downarrow7.4}$ | $31.3_{\downarrow10.3}$ | $18.7_{\downarrow6.9}$ | $25.7_{\downarrow2.8}$ |
| | BoN | $79.7_{\uparrow0.0}$ | $27.8_{\uparrow0.0}$ | $32.8_{\uparrow0.0}$ | $69.4_{\uparrow0.0}$ | $37.5_{\uparrow0.0}$ | $41.6_{\uparrow0.0}$ | $25.6_{\uparrow0.0}$ | $28.5_{\uparrow0.0}$ |
| | Debate | $77.2_{\downarrow2.5}$ | $27.4_{\downarrow0.4}$ | $30.3_{\downarrow2.5}$ | $66.9_{\downarrow2.5}$ | $36.8_{\downarrow0.7}$ | $39.4_{\downarrow2.2}$ | $26.4_{\uparrow0.8}$ | $30.0_{\uparrow1.5}$ |
| | DyLAN | $86.8_{\uparrow7.1}$ | $31.2_{\uparrow3.4}$ | $39.4_{\uparrow6.6}$ | $76.6_{\uparrow7.2}$ | $34.8_{\downarrow2.7}$ | $36.0_{\downarrow5.6}$ | $22.8_{\downarrow2.8}$ | $28.0_{\downarrow0.5}$ |
| | GPTSwarm | $87.0_{\uparrow7.3}$ | $31.3_{\uparrow3.5}$ | $38.7_{\uparrow5.9}$ | $75.9_{\uparrow6.5}$ | $34.9_{\downarrow2.6}$ | $35.9_{\downarrow5.7}$ | $25.0_{\downarrow0.6}$ | $28.4_{\downarrow0.1}$ |
| | MacNet | $78.9_{\downarrow0.8}$ | $28.7_{\uparrow0.9}$ | $31.9_{\downarrow0.9}$ | $70.4_{\uparrow1.0}$ | $39.2_{\uparrow1.7}$ | $46.0_{\uparrow4.4}$ | $32.2_{\uparrow6.6}$ | $32.7_{\uparrow4.2}$ |
| | M2CL (ours) | $93.7_{\uparrow14.0}$ | $51.7_{\uparrow23.9}$ | $66.2_{\uparrow33.4}$ | $91.1_{\uparrow21.7}$ | $48.2_{\uparrow10.7}$ | $56.1_{\uparrow14.5}$ | $42.0_{\uparrow16.4}$ | $43.0_{\uparrow14.5}$ |
| Qwen-72B | Single | $72.5_{\downarrow11.7}$ | $31.6_{\downarrow19.4}$ | $34.9_{\downarrow11.0}$ | $59.1_{\downarrow13.1}$ | $48.2_{\downarrow9.3}$ | $50.4_{\downarrow11.7}$ | $31.1_{\downarrow10.1}$ | $41.0_{\downarrow10.0}$ |
| | BoN | $84.2_{\uparrow0.0}$ | $51.0_{\uparrow0.0}$ | $45.9_{\uparrow0.0}$ | $72.2_{\uparrow0.0}$ | $57.5_{\uparrow0.0}$ | $62.1_{\uparrow0.0}$ | $41.2_{\uparrow0.0}$ | $51.0_{\uparrow0.0}$ |
| | Debate | $82.7_{\downarrow1.5}$ | $48.4_{\downarrow2.6}$ | $43.4_{\downarrow2.5}$ | $69.1_{\downarrow3.1}$ | $60.4_{\uparrow2.9}$ | $65.2_{\uparrow3.1}$ | $42.9_{\uparrow1.7}$ | $49.1_{\downarrow1.9}$ |
| | DyLAN | $91.5_{\uparrow7.3}$ | $63.1_{\uparrow12.1}$ | $51.6_{\uparrow5.7}$ | $80.4_{\uparrow8.2}$ | $55.1_{\downarrow2.4}$ | $58.0_{\downarrow4.1}$ | $40.4_{\downarrow0.8}$ | $45.5_{\downarrow5.5}$ |
| | GPTSwarm | $91.5_{\uparrow7.3}$ | $64.7_{\uparrow13.7}$ | $52.4_{\uparrow6.5}$ | $79.6_{\uparrow7.4}$ | $56.8_{\downarrow0.7}$ | $60.2_{\downarrow1.9}$ | $40.3_{\downarrow0.9}$ | $49.7_{\downarrow1.3}$ |
| | MacNet | $83.8_{\downarrow0.4}$ | $52.9_{\uparrow1.9}$ | $46.2_{\uparrow0.3}$ | $70.5_{\downarrow1.7}$ | $61.8_{\uparrow4.3}$ | $68.4_{\uparrow6.3}$ | $46.4_{\uparrow5.2}$ | $53.7_{\uparrow2.7}$ |
| | M2CL (ours) | $95.1_{\uparrow10.9}$ | $72.5_{\uparrow21.5}$ | $78.9_{\uparrow33.0}$ | $90.7_{\uparrow18.5}$ | $79.0_{\uparrow21.5}$ | $88.9_{\uparrow26.8}$ | $67.2_{\uparrow26.0}$ | $70.5_{\uparrow19.5}$ |

Table 6: Accuracy with varying number of LLMs on different datasets. The number of LLMs is 16 for all datasets. We exhibit the performance advantage with BoN and highlight the best result.

| Model | Method | MMLU | MATH | GPQA | Code | ALFWorld | SciWorld | GAIA | PDDL |
|---|---|---|---|---|---|---|---|---|---|
| Llama-7B | Single | $45.3_{\downarrow5.2}$ | $2.9_{\downarrow4.0}$ | $16.8_{\downarrow10.7}$ | $15.8_{\downarrow6.0}$ | $22.4_{\downarrow4.3}$ | $19.6_{\downarrow6.1}$ | $14.7_{\downarrow4.5}$ | $24.2_{\downarrow2.1}$ |
| | BoN | $50.5_{\uparrow0.0}$ | $6.9_{\uparrow0.0}$ | $27.5_{\uparrow0.0}$ | $21.8_{\uparrow0.0}$ | $26.7_{\uparrow0.0}$ | $25.7_{\uparrow0.0}$ | $19.2_{\uparrow0.0}$ | $26.3_{\uparrow0.0}$ |
| | Debate | $51_{\uparrow0.5}$ | $6.4_{\downarrow0.5}$ | $26.1_{\downarrow1.4}$ | $21.3_{\downarrow0.5}$ | $25.7_{\downarrow1.0}$ | $25.3_{\downarrow0.4}$ | $16.6_{\downarrow2.6}$ | $26.4_{\uparrow0.1}$ |
| | DyLAN | $54.5_{\uparrow4.0}$ | $9.9_{\uparrow3.0}$ | $35.5_{\uparrow8.0}$ | $26.2_{\uparrow4.4}$ | $24.6_{\downarrow2.1}$ | $22.7_{\downarrow3.0}$ | $17.2_{\downarrow2.0}$ | $25.8_{\downarrow0.5}$ |
| | GPTSwarm | $51.6_{\uparrow1.1}$ | $6.9_{\uparrow0.0}$ | $29.1_{\uparrow1.6}$ | $23.0_{\uparrow1.2}$ | $27.2_{\uparrow0.5}$ | $26.8_{\uparrow1.1}$ | $20.8_{\uparrow1.6}$ | $29.7_{\uparrow3.4}$ |
| | MacNet | $59.3_{\uparrow8.8}$ | $12.5_{\uparrow5.6}$ | $45.0_{\uparrow17.5}$ | $33.7_{\uparrow11.9}$ | $32.8_{\uparrow6.1}$ | $33.4_{\uparrow7.7}$ | $22.6_{\uparrow3.4}$ | $34.2_{\uparrow7.9}$ |
| | M2CL (ours) | $71.5_{\uparrow21.0}$ | $23.9_{\uparrow17.0}$ | $69.9_{\uparrow42.4}$ | $48.3_{\uparrow26.5}$ | $39.8_{\uparrow13.1}$ | $38.4_{\uparrow12.7}$ | $28.9_{\uparrow9.7}$ | $37.0_{\uparrow10.7}$ |
| Llama-14B | Single | $54.8_{\downarrow16.2}$ | $3.9_{\downarrow4.9}$ | $25.0_{\downarrow12.7}$ | $23.7_{\downarrow8.3}$ | $27.4_{\downarrow5.6}$ | $24.0_{\downarrow9.5}$ | $18.3_{\downarrow5.0}$ | $29.2_{\downarrow4.6}$ |
| | BoN | $71.0_{\uparrow0.0}$ | $8.8_{\uparrow0.0}$ | $37.7_{\uparrow0.0}$ | $32.0_{\uparrow0.0}$ | $33.0_{\uparrow0.0}$ | $33.5_{\uparrow0.0}$ | $23.3_{\uparrow0.0}$ | $33.8_{\uparrow0.0}$ |
| | Debate | $72.2_{\uparrow1.2}$ | $8.2_{\downarrow0.6}$ | $37.0_{\downarrow0.7}$ | $32.0_{\uparrow0.0}$ | $32.0_{\downarrow1.0}$ | $30.9_{\downarrow2.6}$ | $19.6_{\downarrow3.7}$ | $32.7_{\downarrow1.1}$ |
| | DyLAN | $83.2_{\uparrow12.2}$ | $12.4_{\uparrow3.6}$ | $47.0_{\uparrow9.3}$ | $38.1_{\uparrow6.1}$ | $31.9_{\downarrow1.1}$ | $30.2_{\downarrow3.3}$ | $20.0_{\downarrow3.3}$ | $31.2_{\downarrow2.6}$ |
| | MacNet | $86.8_{\uparrow15.8}$ | $16.0_{\uparrow7.2}$ | $59.7_{\uparrow22.0}$ | $49.1_{\uparrow17.1}$ | $34.8_{\uparrow1.8}$ | $37.8_{\uparrow4.3}$ | $26.7_{\uparrow3.4}$ | $39.9_{\uparrow6.1}$ |
| | MacNet | $86.8_{\uparrow15.8}$ | $16.0_{\uparrow7.2}$ | $59.7_{\uparrow22.0}$ | $49.1_{\uparrow17.1}$ | $34.8_{\uparrow1.8}$ | $37.8_{\uparrow4.3}$ | $26.7_{\uparrow3.4}$ | $39.9_{\uparrow6.1}$ |
| | M2CL (ours) | $94.5_{\uparrow23.5}$ | $32.7_{\uparrow23.9}$ | $88.5_{\uparrow50.8}$ | $66.1_{\uparrow34.1}$ | $48.7_{\uparrow15.7}$ | $47.1_{\uparrow13.6}$ | $36.2_{\uparrow12.9}$ | $45.5_{\uparrow11.7}$ |
| Llama-70B | Single | $68.9_{\downarrow14.1}$ | $31.6_{\downarrow3.2}$ | $27.6_{\downarrow32.0}$ | $35.5_{\downarrow12.5}$ | $43.7_{\downarrow10.2}$ | $38.3_{\downarrow11.5}$ | $28.6_{\downarrow4.6}$ | $46.8_{\downarrow6.2}$ |
| | BoN | $83.0_{\uparrow0.0}$ | $34.8_{\uparrow0.0}$ | $59.6_{\uparrow0.0}$ | $48.0_{\uparrow0.0}$ | $53.9_{\uparrow0.0}$ | $49.8_{\uparrow0.0}$ | $33.2_{\uparrow0.0}$ | $53.0_{\uparrow0.0}$ |
| | Debate | $82.9_{\downarrow0.1}$ | $34.6_{\downarrow0.2}$ | $58.5_{\downarrow1.1}$ | $46.8_{\downarrow1.2}$ | $52.2_{\downarrow1.7}$ | $51.2_{\uparrow1.4}$ | $33.0_{\downarrow0.2}$ | $52.8_{\downarrow0.2}$ |
| | DyLAN | $93.2_{\uparrow10.2}$ | $37.2_{\uparrow2.4}$ | $73.2_{\uparrow13.6}$ | $57.2_{\uparrow9.2}$ | $50.5_{\downarrow3.4}$ | $46.8_{\downarrow3.0}$ | $31.1_{\downarrow2.1}$ | $49.7_{\downarrow3.3}$ |
| | GPTSwarm | $93.0_{\uparrow10.0}$ | $37.4_{\uparrow2.6}$ | $69.7_{\uparrow10.1}$ | $55.5_{\uparrow7.5}$ | $51.7_{\downarrow2.2}$ | $47.5_{\downarrow2.3}$ | $32.4_{\downarrow0.8}$ | $51.6_{\downarrow1.4}$ |
| | MacNet | $86.7_{\uparrow3.7}$ | $38.9_{\uparrow4.1}$ | $68.3_{\uparrow8.7}$ | $49.8_{\uparrow1.8}$ | $61.6_{\uparrow7.7}$ | $70.8_{\uparrow21.0}$ | $43.8_{\uparrow10.6}$ | $67.1_{\uparrow14.1}$ |
| | M2CL (ours) | $93.9_{\uparrow10.9}$ | $51.5_{\uparrow16.7}$ | $91.3_{\uparrow31.7}$ | $97.2_{\uparrow49.2}$ | $84.9_{\uparrow31.0}$ | $81.7_{\uparrow31.9}$ | $61.5_{\uparrow28.3}$ | $78.1_{\uparrow25.1}$ |
| Qwen-7B | Single | $61.2_{\downarrow13.0}$ | $12.9_{\downarrow12.0}$ | $20.2_{\downarrow16.2}$ | $51.2_{\downarrow11.3}$ | $23.4_{\downarrow9.5}$ | $24.9_{\downarrow10.5}$ | $15.6_{\downarrow6.4}$ | $20.6_{\downarrow6.5}$ |
| | BoN | $74.2_{\uparrow0.0}$ | $24.9_{\uparrow0.0}$ | $36.4_{\uparrow0.0}$ | $62.5_{\uparrow0.0}$ | $32.9_{\uparrow0.0}$ | $35.4_{\uparrow0.0}$ | $22.0_{\uparrow0.0}$ | $27.1_{\uparrow0.0}$ |
| | Debate | $73.4_{\downarrow0.8}$ | $24.1_{\downarrow0.8}$ | $35.6_{\downarrow0.8}$ | $62.1_{\downarrow0.4}$ | $31.7_{\downarrow1.2}$ | $32.6_{\downarrow2.8}$ | $22.1_{\uparrow0.1}$ | $24.2_{\downarrow2.9}$ |
| | DyLAN | $77.4_{\uparrow3.2}$ | $26.0_{\uparrow1.1}$ | $38.9_{\uparrow2.5}$ | $64.8_{\uparrow2.3}$ | $30.4_{\downarrow2.5}$ | $28.7_{\downarrow6.7}$ | $19.2_{\downarrow2.8}$ | $22.7_{\downarrow4.4}$ |
| | GPTSwarm | $78.0_{\uparrow3.8}$ | $27.6_{\uparrow2.7}$ | $41.1_{\uparrow4.7}$ | $66.2_{\uparrow3.7}$ | $30.1_{\downarrow2.8}$ | $30.4_{\downarrow5.0}$ | $22.1_{\uparrow0.1}$ | $24.4_{\downarrow2.7}$ |
| | MacNet | $76.8_{\uparrow2.6}$ | $27.8_{\uparrow2.9}$ | $40.3_{\uparrow3.9}$ | $69.4_{\uparrow6.9}$ | $35.4_{\uparrow2.5}$ | $39.7_{\uparrow4.3}$ | $24.6_{\uparrow2.6}$ | $30.7_{\uparrow3.6}$ |
| | M2CL (ours) | $94.6_{\uparrow20.4}$ | $47.4_{\uparrow22.5}$ | $72.8_{\uparrow36.4}$ | $84.9_{\uparrow22.4}$ | $41.8_{\uparrow8.9}$ | $47.6_{\uparrow12.2}$ | $36.5_{\uparrow14.5}$ | $37.4_{\uparrow10.3}$ |
| Qwen-14B | Single | $67.2_{\downarrow12.5}$ | $21.6_{\downarrow6.2}$ | $21.2_{\downarrow11.6}$ | $56.7_{\downarrow12.7}$ | $28.3_{\downarrow9.8}$ | $30.9_{\downarrow11.4}$ | $18.5_{\downarrow7.6}$ | $25.5_{\downarrow3.8}$ |
| | BoN | $79.7_{\uparrow0.0}$ | $27.8_{\uparrow0.0}$ | $32.8_{\uparrow0.0}$ | $69.4_{\uparrow0.0}$ | $38.1_{\uparrow0.0}$ | $42.3_{\uparrow0.0}$ | $26.1_{\uparrow0.0}$ | $29.3_{\uparrow0.0}$ |
| | Debate | $78.7_{\downarrow1.0}$ | $26.7_{\downarrow1.1}$ | $33.4_{\downarrow0.6}$ | $70.1_{\uparrow0.7}$ | $37.7_{\downarrow0.4}$ | $40.5_{\downarrow1.8}$ | $26.7_{\uparrow0.6}$ | $30.4_{\uparrow1.1}$ |
| | DyLAN | $88.0_{\uparrow8.3}$ | $31.9_{\uparrow4.1}$ | $41.0_{\uparrow8.2}$ | $78.2_{\uparrow8.8}$ | $34.9_{\downarrow3.2}$ | $35.7_{\downarrow6.6}$ | $23.9_{\downarrow2.2}$ | $28.3_{\downarrow1.0}$ |
| | GPTSwarm | $88.0_{\uparrow8.3}$ | $31.8_{\uparrow4.0}$ | $40.4_{\uparrow7.6}$ | $78.4_{\uparrow9.0}$ | $35.5_{\downarrow2.6}$ | $36.8_{\downarrow5.5}$ | $25.1_{\downarrow1.0}$ | $28.7_{\downarrow0.6}$ |
| | MacNet | $83.1_{\uparrow3.4}$ | $30.8_{\uparrow3.0}$ | $35.1_{\uparrow2.3}$ | $74.8_{\uparrow5.4}$ | $42.1_{\uparrow4.0}$ | $49.3_{\uparrow7.0}$ | $35.5_{\uparrow9.4}$ | $33.8_{\uparrow4.5}$ |
| | M2CL (ours) | $95.9_{\uparrow16.2}$ | $52.7_{\uparrow24.9}$ | $69.0_{\uparrow36.2}$ | $93.3_{\uparrow23.9}$ | $51.2_{\uparrow13.1}$ | $59.4_{\uparrow17.1}$ | $43.9_{\uparrow17.8}$ | $46.0_{\uparrow16.7}$ |
| Qwen-72B | Single | $72.5_{\downarrow11.7}$ | $31.6_{\downarrow19.4}$ | $34.9_{\downarrow11.0}$ | $59.1_{\downarrow13.1}$ | $47.7_{\downarrow10.8}$ | $50.1_{\downarrow12.8}$ | $31.2_{\downarrow12.1}$ | $40.8_{\downarrow11.5}$ |
| | BoN | $84.2_{\uparrow0.0}$ | $51.0_{\uparrow0.0}$ | $45.9_{\uparrow0.0}$ | $72.2_{\uparrow0.0}$ | $58.5_{\uparrow0.0}$ | $62.9_{\uparrow0.0}$ | $43.3_{\uparrow0.0}$ | $52.3_{\uparrow0.0}$ |
| | Debate | $83.9_{\downarrow0.3}$ | $56.1_{\uparrow5.1}$ | $44.1_{\downarrow1.8}$ | $71.6_{\downarrow0.6}$ | $60.8_{\uparrow2.3}$ | $66.7_{\uparrow3.8}$ | $44.1_{\uparrow0.8}$ | $49.3_{\downarrow3.0}$ |
| | DyLAN | $92.7_{\uparrow8.5}$ | $65.8_{\uparrow14.8}$ | $53.6_{\uparrow7.7}$ | $81.4_{\uparrow9.2}$ | $56.1_{\downarrow2.4}$ | $59.1_{\downarrow3.8}$ | $41.1_{\downarrow2.2}$ | $46.3_{\downarrow6.0}$ |
| | GPTSwarm | $92.4_{\uparrow8.2}$ | $66.0_{\uparrow15.0}$ | $53.3_{\uparrow7.4}$ | $81.1_{\uparrow8.9}$ | $57.3_{\downarrow1.2}$ | $59.8_{\downarrow3.1}$ | $41.3_{\downarrow2.0}$ | $50.6_{\downarrow1.7}$ |
| | MacNet | $88.1_{\uparrow3.9}$ | $60.6_{\uparrow9.6}$ | $49.2_{\uparrow3.3}$ | $72.6_{\uparrow0.4}$ | $65.8_{\uparrow7.3}$ | $74.7_{\uparrow11.8}$ | $50.3_{\uparrow7.0}$ | $58.1_{\uparrow5.8}$ |
| | M2CL (ours) | $96.6_{\uparrow12.4}$ | $74.3_{\uparrow23.3}$ | $80.0_{\uparrow34.1}$ | $94.4_{\uparrow22.2}$ | $82.9_{\uparrow24.4}$ | $94.6_{\uparrow31.7}$ | $71.9_{\uparrow28.6}$ | $74.3_{\uparrow22.0}$ |

Table 7: Accuracy with varying number of LLMs on different datasets. The number of LLMs is 32 for all datasets. We exhibit the performance advantage with BoN and highlight the  best  result.

| Model | Method | MMLU | MATH | GPQA | Code | ALFWorld | SciWorld | GAIA | PDDL |
|---|---|---|---|---|---|---|---|---|---|
| Llama-7B | Single | 45.3$_{\downarrow5.2}$ | 2.9$_{\downarrow4.0}$ | 16.8$_{\downarrow10.7}$ | 15.8$_{\downarrow6.0}$ | 22.4$_{\downarrow4.4}$ | 18.8$_{\downarrow8.1}$ | 14.7$_{\downarrow5.1}$ | 23.2$_{\downarrow4.0}$ |
| | BoN | 50.5$_{\uparrow0.0}$ | 6.9$_{\uparrow0.0}$ | 27.5$_{\uparrow0.0}$ | 21.8$_{\uparrow0.0}$ | 26.8$_{\uparrow0.0}$ | 26.9$_{\uparrow0.0}$ | 19.8$_{\uparrow0.0}$ | 27.2$_{\uparrow0.0}$ |
| | Debate | 51.8$_{\uparrow1.3}$ | 7.7$_{\uparrow0.8}$ | 29.5$_{\uparrow2.0}$ | 23.5$_{\uparrow1.7}$ | 26.4$_{\downarrow0.4}$ | 25.6$_{\downarrow1.3}$ | 15.8$_{\downarrow4.0}$ | 26.2$_{\downarrow1.0}$ |
| | DyLAN | 54.7$_{\uparrow4.2}$ | 10.1$_{\uparrow3.2}$ | 36.2$_{\uparrow8.7}$ | 26.7$_{\uparrow4.9}$ | 24.9$_{\downarrow1.9}$ | 23.1$_{\downarrow3.8}$ | 17.6$_{\downarrow2.2}$ | 26.0$_{\downarrow1.2}$ |
| | GPTSwarm | 52.4$_{\uparrow1.9}$ | 7.9$_{\uparrow1.0}$ | 30.8$_{\uparrow3.3}$ | 23.7$_{\uparrow1.9}$ | 27.3$_{\uparrow0.5}$ | 27.9$_{\uparrow1.0}$ | 21.8$_{\uparrow2.0}$ | 30.4$_{\uparrow3.2}$ |
| | MacNet | 62.7$_{\uparrow12.2}$ | 14.0$_{\uparrow7.1}$ | 50.3$_{\uparrow22.8}$ | 37.2$_{\uparrow15.4}$ | 35.7$_{\uparrow8.9}$ | 37.5$_{\uparrow10.6}$ | 23.8$_{\uparrow4.0}$ | 37.5$_{\uparrow10.3}$ |
| | M2CL (ours) | 79.1$_{\uparrow28.6}$ | 28.8$_{\uparrow21.9}$ | 84.5$_{\uparrow57.0}$ | 56.5$_{\uparrow34.7}$ | 42.5$_{\uparrow15.7}$ | 40.7$_{\uparrow13.8}$ | 31.4$_{\uparrow11.6}$ | 38.9$_{\uparrow11.7}$ |
| Llama-14B | Single | 54.8$_{\downarrow16.2}$ | 3.9$_{\downarrow4.9}$ | 25.0$_{\downarrow12.7}$ | 23.7$_{\downarrow8.3}$ | 27.0$_{\downarrow5.8}$ | 23.5$_{\downarrow11.3}$ | 18.1$_{\downarrow6.1}$ | 28.9$_{\downarrow5.4}$ |
| | BoN | 71.0$_{\uparrow0.0}$ | 8.8$_{\uparrow0.0}$ | 37.7$_{\uparrow0.0}$ | 32.0$_{\uparrow0.0}$ | 32.8$_{\uparrow0.0}$ | 34.8$_{\uparrow0.0}$ | 24.2$_{\uparrow0.0}$ | 34.3$_{\uparrow0.0}$ |
| | Debate | 73.1$_{\uparrow2.1}$ | 9.5$_{\uparrow0.7}$ | 40.7$_{\uparrow3.0}$ | 34.3$_{\uparrow2.3}$ | 31.9$_{\downarrow0.9}$ | 31.7$_{\downarrow3.1}$ | 19.9$_{\downarrow4.3}$ | 33.0$_{\downarrow1.3}$ |
| | DyLAN | 83.6$_{\uparrow12.6}$ | 12.7$_{\uparrow3.9}$ | 47.7$_{\uparrow10.0}$ | 38.7$_{\uparrow6.7}$ | 32.1$_{\downarrow0.7}$ | 30.8$_{\downarrow4.0}$ | 19.8$_{\downarrow4.4}$ | 31.0$_{\downarrow3.3}$ |
| | MacNet | 90.7$_{\uparrow19.7}$ | 17.6$_{\uparrow8.8}$ | 67.2$_{\uparrow29.5}$ | 53.9$_{\uparrow21.9}$ | 37.1$_{\uparrow4.3}$ | 41.7$_{\uparrow6.9}$ | 29.1$_{\uparrow4.9}$ | 42.3$_{\uparrow8.0}$ |
| | MacNet | 90.7$_{\uparrow19.7}$ | 17.6$_{\uparrow8.8}$ | 67.2$_{\uparrow29.5}$ | 53.9$_{\uparrow21.9}$ | 37.1$_{\uparrow4.3}$ | 41.7$_{\uparrow6.9}$ | 29.1$_{\uparrow4.9}$ | 42.3$_{\uparrow8.0}$ |
| | M2CL (ours) | 95.8$_{\uparrow24.8}$ | 41.4$_{\uparrow32.6}$ | 94.7$_{\uparrow57.0}$ | 75.6$_{\uparrow43.6}$ | 52.2$_{\uparrow19.4}$ | 50.7$_{\uparrow15.9}$ | 38.5$_{\uparrow14.3}$ | 48.0$_{\uparrow13.7}$ |
| Llama-70B | Single | 68.9$_{\downarrow14.1}$ | 31.6$_{\downarrow3.2}$ | 27.6$_{\downarrow32.0}$ | 35.5$_{\downarrow12.5}$ | 43.1$_{\downarrow10.0}$ | 38.6$_{\downarrow11.5}$ | 28.1$_{\downarrow4.7}$ | 46.0$_{\downarrow7.2}$ |
| | BoN | 83.0$_{\uparrow0.0}$ | 34.8$_{\uparrow0.0}$ | 59.6$_{\uparrow0.0}$ | 48.0$_{\uparrow0.0}$ | 53.1$_{\uparrow0.0}$ | 50.1$_{\uparrow0.0}$ | 32.8$_{\uparrow0.0}$ | 53.2$_{\uparrow0.0}$ |
| | Debate | 85.7$_{\uparrow2.7}$ | 35.3$_{\uparrow0.5}$ | 63.4$_{\uparrow3.8}$ | 51.2$_{\uparrow3.2}$ | 52.6$_{\downarrow0.5}$ | 52.1$_{\uparrow2.0}$ | 32.9$_{\uparrow0.1}$ | 53.7$_{\uparrow0.5}$ |
| | DyLAN | 94.5$_{\uparrow11.5}$ | 37.3$_{\uparrow2.5}$ | 74.6$_{\uparrow15.0}$ | 57.8$_{\uparrow9.8}$ | 50.1$_{\downarrow3.0}$ | 47.2$_{\downarrow2.9}$ | 31.4$_{\downarrow1.4}$ | 49.7$_{\downarrow3.5}$ |
| | GPTSwarm | 96.4$_{\uparrow13.4}$ | 37.8$_{\uparrow3.0}$ | 73.4$_{\uparrow13.8}$ | 56.4$_{\uparrow8.4}$ | 52.2$_{\downarrow0.9}$ | 48.4$_{\downarrow1.7}$ | 32.5$_{\downarrow0.3}$ | 51.6$_{\downarrow1.6}$ |
| | MacNet | 90.0$_{\uparrow7.0}$ | 41.2$_{\uparrow6.4}$ | 73.9$_{\uparrow14.3}$ | 54.1$_{\uparrow6.1}$ | 64.5$_{\uparrow11.4}$ | 76.6$_{\uparrow26.5}$ | 46.5$_{\uparrow13.7}$ | 71.6$_{\uparrow18.4}$ |
| | M2CL (ours) | 97.0$_{\uparrow14.0}$ | 54.5$_{\uparrow19.7}$ | 95.1$_{\uparrow35.5}$ | 93.7$_{\uparrow45.7}$ | 88.5$_{\uparrow35.4}$ | 86.0$_{\uparrow35.9}$ | 65.7$_{\uparrow32.9}$ | 79.8$_{\uparrow26.6}$ |
| Qwen-7B | Single | 61.2$_{\downarrow13.0}$ | 12.9$_{\downarrow12.0}$ | 20.2$_{\downarrow16.2}$ | 51.2$_{\downarrow11.3}$ | 24.2$_{\downarrow8.6}$ | 24.5$_{\downarrow13.3}$ | 14.6$_{\downarrow7.7}$ | 20.4$_{\downarrow7.3}$ |
| | BoN | 74.2$_{\uparrow0.0}$ | 24.9$_{\uparrow0.0}$ | 36.4$_{\uparrow0.0}$ | 62.5$_{\uparrow0.0}$ | 32.8$_{\uparrow0.0}$ | 37.8$_{\uparrow0.0}$ | 22.3$_{\uparrow0.0}$ | 27.7$_{\uparrow0.0}$ |
| | Debate | 76.8$_{\uparrow2.6}$ | 26.8$_{\uparrow1.9}$ | 40.0$_{\uparrow3.6}$ | 64.3$_{\uparrow1.8}$ | 31.7$_{\downarrow1.1}$ | 33.4$_{\downarrow4.4}$ | 23.0$_{\uparrow0.7}$ | 25.3$_{\downarrow2.4}$ |
| | DyLAN | 78.2$_{\uparrow4.0}$ | 29.5$_{\uparrow4.6}$ | 40.3$_{\uparrow3.9}$ | 66.5$_{\uparrow4.0}$ | 30.7$_{\downarrow2.1}$ | 29.2$_{\downarrow8.6}$ | 19.0$_{\downarrow3.3}$ | 22.6$_{\downarrow5.1}$ |
| | GPTSwarm | 79.6$_{\uparrow5.4}$ | 29.9$_{\uparrow5.0}$ | 41.8$_{\uparrow5.4}$ | 67.8$_{\uparrow5.3}$ | 31.2$_{\downarrow1.6}$ | 31.3$_{\downarrow6.5}$ | 22.6$_{\uparrow0.3}$ | 25.0$_{\downarrow2.7}$ |
| | MacNet | 80.6$_{\uparrow6.4}$ | 31.5$_{\uparrow6.6}$ | 45.8$_{\uparrow9.4}$ | 73.7$_{\uparrow11.2}$ | 38.6$_{\uparrow5.8}$ | 43.1$_{\uparrow5.3}$ | 26.6$_{\uparrow4.3}$ | 34.3$_{\uparrow6.6}$ |
| | M2CL (ours) | 96.0$_{\uparrow21.8}$ | 47.1$_{\uparrow22.2}$ | 74.9$_{\uparrow38.5}$ | 87.3$_{\uparrow24.8}$ | 44.4$_{\uparrow11.6}$ | 50.4$_{\uparrow12.6}$ | 39.7$_{\uparrow17.4}$ | 40.2$_{\uparrow12.5}$ |
| Qwen-14B | Single | 67.2$_{\downarrow12.5}$ | 21.6$_{\downarrow6.2}$ | 21.2$_{\downarrow11.6}$ | 56.7$_{\downarrow12.7}$ | 29.4$_{\downarrow8.0}$ | 29.5$_{\downarrow15.6}$ | 18.9$_{\downarrow8.2}$ | 24.5$_{\downarrow4.5}$ |
| | BoN | 79.7$_{\uparrow0.0}$ | 27.8$_{\uparrow0.0}$ | 32.8$_{\uparrow0.0}$ | 69.4$_{\uparrow0.0}$ | 37.4$_{\uparrow0.0}$ | 45.1$_{\uparrow0.0}$ | 27.1$_{\uparrow0.0}$ | 29.0$_{\uparrow0.0}$ |
| | Debate | 82.7$_{\uparrow3.0}$ | 29.1$_{\uparrow1.3}$ | 35.5$_{\uparrow2.7}$ | 71.8$_{\uparrow2.4}$ | 39.0$_{\uparrow1.6}$ | 41.7$_{\downarrow3.4}$ | 28.7$_{\uparrow1.6}$ | 30.8$_{\uparrow1.8}$ |
| | DyLAN | 89.3$_{\uparrow9.6}$ | 32.6$_{\uparrow4.8}$ | 42.2$_{\uparrow9.4}$ | 78.9$_{\uparrow9.5}$ | 35.4$_{\downarrow2.0}$ | 36.2$_{\downarrow8.9}$ | 23.2$_{\downarrow3.9}$ | 29.3$_{\uparrow0.3}$ |
| | GPTSwarm | 89.0$_{\uparrow9.3}$ | 32.6$_{\uparrow4.8}$ | 42.2$_{\uparrow9.4}$ | 78.9$_{\uparrow9.5}$ | 36.4$_{\downarrow1.0}$ | 37.5$_{\downarrow7.6}$ | 26.6$_{\downarrow0.5}$ | 28.9$_{\downarrow0.1}$ |
| | MacNet | 87.8$_{\uparrow8.1}$ | 33.7$_{\uparrow5.9}$ | 40.1$_{\uparrow7.3}$ | 83.0$_{\uparrow13.6}$ | 44.8$_{\uparrow7.4}$ | 54.2$_{\uparrow9.1}$ | 39.8$_{\uparrow12.7}$ | 36.2$_{\uparrow7.2}$ |
| | M2CL (ours) | 96.1$_{\uparrow16.4}$ | 58.0$_{\uparrow30.2}$ | 70.5$_{\uparrow37.7}$ | 94.9$_{\uparrow25.5}$ | 54.9$_{\uparrow17.5}$ | 62.1$_{\uparrow17.0}$ | 49.0$_{\uparrow21.9}$ | 49.6$_{\uparrow20.6}$ |
| Qwen-72B | Single | 72.5$_{\downarrow11.7}$ | 31.6$_{\downarrow19.4}$ | 34.9$_{\downarrow11.0}$ | 59.1$_{\downarrow13.1}$ | 46.7$_{\downarrow12.6}$ | 49.0$_{\downarrow14.2}$ | 30.4$_{\downarrow13.5}$ | 40.1$_{\downarrow13.2}$ |
| | BoN | 84.2$_{\uparrow0.0}$ | 51.0$_{\uparrow0.0}$ | 45.9$_{\uparrow0.0}$ | 72.2$_{\uparrow0.0}$ | 59.3$_{\uparrow0.0}$ | 63.2$_{\uparrow0.0}$ | 43.9$_{\uparrow0.0}$ | 53.3$_{\uparrow0.0}$ |
| | Debate | 86.5$_{\uparrow2.3}$ | 62.9$_{\uparrow11.9}$ | 48.3$_{\uparrow2.4}$ | 74.5$_{\uparrow2.3}$ | 62.8$_{\uparrow3.5}$ | 67.8$_{\uparrow4.6}$ | 46.6$_{\uparrow2.7}$ | 50.2$_{\downarrow3.1}$ |
| | DyLAN | 92.8$_{\uparrow8.6}$ | 67.7$_{\uparrow16.7}$ | 54.8$_{\uparrow8.9}$ | 82.7$_{\uparrow10.5}$ | 56.0$_{\downarrow3.3}$ | 59.1$_{\downarrow4.1}$ | 42.9$_{\downarrow1.0}$ | 46.7$_{\downarrow6.6}$ |
| | GPTSwarm | 93.1$_{\uparrow8.9}$ | 68.1$_{\uparrow17.1}$ | 54.7$_{\uparrow8.8}$ | 82.2$_{\uparrow10.0}$ | 58.3$_{\downarrow1.0}$ | 62.6$_{\downarrow0.6}$ | 42.7$_{\downarrow1.2}$ | 51.4$_{\downarrow1.9}$ |
| | MacNet | 92.5$_{\uparrow8.3}$ | 72.1$_{\uparrow21.1}$ | 55.2$_{\uparrow9.3}$ | 79.9$_{\uparrow7.7}$ | 70.5$_{\uparrow11.2}$ | 79.5$_{\uparrow16.3}$ | 54.7$_{\uparrow10.8}$ | 62.3$_{\uparrow9.0}$ |
| | M2CL (ours) | 97.5$_{\uparrow13.3}$ | 74.7$_{\uparrow23.7}$ | 82.0$_{\uparrow36.1}$ | 95.9$_{\uparrow23.7}$ | 89.3$_{\uparrow30.0}$ | 95$_{\uparrow31.8}$ | 79.7$_{\uparrow35.8}$ | 80.0$_{\uparrow26.7}$ |

Table 8: Accuracy with varying number of LLMs on different datasets. The number of LLMs is 64 for all datasets. We exhibit the performance advantage with BoN and highlight the best result.

| Model | Method | MMLU | MATH | GPQA | Code | ALFWorld | SciWorld | GAIA | PDDL |
|---|---|---|---|---|---|---|---|---|---|
| Llama-7B | Single | $45.3_{\downarrow 5.2}$ | $2.9_{\downarrow 4.0}$ | $16.8_{\downarrow 10.7}$ | $15.8_{\downarrow 6.0}$ | $22.0_{\downarrow 5.2}$ | $18.8_{\downarrow 7.9}$ | $15.0_{\downarrow 4.8}$ | $22.5_{\downarrow 4.4}$ |
| | BoN | $50.5_{\uparrow 0.0}$ | $6.9_{\uparrow 0.0}$ | $27.5_{\uparrow 0.0}$ | $21.8_{\uparrow 0.0}$ | $27.2_{\uparrow 0.0}$ | $26.7_{\uparrow 0.0}$ | $19.8_{\downarrow 0.0}$ | $26.9_{\uparrow 0.0}$ |
| | Debate | $53.4_{\uparrow 2.9}$ | $9.2_{\uparrow 2.3}$ | $33.5_{\uparrow 6.0}$ | $25_{\uparrow 3.2}$ | $26.9_{\downarrow 0.3}$ | $26.3_{\downarrow 0.4}$ | $16.5_{\downarrow 3.3}$ | $26.9_{\uparrow 0.0}$ |
| | DyLAN | $55.1_{\uparrow 4.6}$ | $10.4_{\uparrow 3.5}$ | $36.9_{\uparrow 9.4}$ | $27_{\uparrow 5.2}$ | $24.6_{\downarrow 2.6}$ | $23.3_{\downarrow 3.4}$ | $17.9_{\downarrow 1.9}$ | $25.0_{\downarrow 1.9}$ |
| | GPTSwarm | $53.5_{\uparrow 3.0}$ | $9_{\uparrow 2.1}$ | $34_{\uparrow 6.5}$ | $25.3_{\uparrow 3.5}$ | $27.0_{\downarrow 0.2}$ | $28.8_{\uparrow 2.1}$ | $22.1_{\uparrow 2.3}$ | $31.0_{\uparrow 4.1}$ |
| | MacNet | $66.4_{\uparrow 15.9}$ | $15.3_{\uparrow 8.4}$ | $54.4_{\uparrow 26.9}$ | $40.0_{\uparrow 18.2}$ | $38.8_{\uparrow 11.6}$ | $39.6_{\uparrow 12.9}$ | $27.0_{\uparrow 7.2}$ | $38.4_{\uparrow 11.5}$ |
| | M2CL (ours) | $81.5_{\uparrow 31.0}$ | $35.4_{\uparrow 28.5}$ | $82.5_{\uparrow 55.0}$ | $60.6_{\uparrow 38.8}$ | $44.2_{\uparrow 17.0}$ | $42.9_{\uparrow 16.2}$ | $32.9_{\uparrow 13.1}$ | $40.9_{\uparrow 14.0}$ |
| Llama-14B | Single | $54.8_{\downarrow 16.2}$ | $3.9_{\downarrow 4.9}$ | $25.0_{\downarrow 12.7}$ | $23.7_{\downarrow 8.3}$ | $26.9_{\downarrow 5.7}$ | $23.6_{\downarrow 12.1}$ | $17.8_{\downarrow 6.7}$ | $28.6_{\downarrow 5.6}$ |
| | BoN | $71.0_{\uparrow 0.0}$ | $8.8_{\uparrow 0.0}$ | $37.7_{\uparrow 0.0}$ | $32.0_{\uparrow 0.0}$ | $32.6_{\uparrow 0.0}$ | $35.7_{\uparrow 0.0}$ | $24.5_{\uparrow 0.0}$ | $34.2_{\uparrow 0.0}$ |
| | Debate | $79.6_{\uparrow 8.6}$ | $11.1_{\uparrow 2.3}$ | $45.0_{\uparrow 7.3}$ | $36.5_{\uparrow 4.5}$ | $32.6_{\uparrow 0.0}$ | $32.0_{\downarrow 3.7}$ | $19.9_{\downarrow 4.6}$ | $32.6_{\downarrow 1.6}$ |
| | DyLAN | $85.0_{\uparrow 14.0}$ | $13.0_{\uparrow 4.2}$ | $48.8_{\uparrow 11.1}$ | $39.3_{\uparrow 7.3}$ | $31.9_{\downarrow 0.7}$ | $31.0_{\downarrow 4.7}$ | $19.9_{\downarrow 4.6}$ | $30.5_{\downarrow 3.7}$ |
| | MacNet | $90.8_{\uparrow 19.8}$ | $18.1_{\uparrow 9.3}$ | $67.5_{\uparrow 29.8}$ | $55.2_{\uparrow 23.2}$ | $39.1_{\uparrow 6.5}$ | $44.4_{\uparrow 8.7}$ | $31.1_{\uparrow 6.6}$ | $44.0_{\uparrow 9.8}$ |
| | MacNet | $90.8_{\uparrow 19.8}$ | $18.1_{\uparrow 9.3}$ | $67.5_{\uparrow 29.8}$ | $55.2_{\uparrow 23.2}$ | $39.1_{\uparrow 6.5}$ | $44.4_{\uparrow 8.7}$ | $31.1_{\uparrow 6.6}$ | $44.0_{\uparrow 9.8}$ |
| | M2CL (ours) | $96.8_{\uparrow 25.8}$ | $40.0_{\uparrow 31.2}$ | $93.1_{\uparrow 55.4}$ | $82.5_{\uparrow 50.5}$ | $54.8_{\uparrow 22.2}$ | $52.8_{\uparrow 17.1}$ | $40.3_{\uparrow 15.8}$ | $49.8_{\uparrow 15.6}$ |
| Llama-70B | Single | $68.9_{\downarrow 14.1}$ | $31.6_{\downarrow 3.2}$ | $27.6_{\downarrow 32.0}$ | $35.5_{\downarrow 12.5}$ | $43.0_{\downarrow 11.8}$ | $37.3_{\downarrow 13.7}$ | $28.2_{\downarrow 5.7}$ | $45.9_{\downarrow 6.8}$ |
| | BoN | $83.0_{\uparrow 0.0}$ | $34.8_{\uparrow 0.0}$ | $59.6_{\uparrow 0.0}$ | $48.0_{\uparrow 0.0}$ | $54.8_{\uparrow 0.0}$ | $51.0_{\uparrow 0.0}$ | $33.9_{\uparrow 0.0}$ | $52.7_{\uparrow 0.0}$ |
| | Debate | $90.0_{\uparrow 7.0}$ | $36.6_{\uparrow 1.8}$ | $69.0_{\uparrow 9.4}$ | $54.0_{\uparrow 6.0}$ | $53.6_{\downarrow 1.2}$ | $53.5_{\uparrow 2.5}$ | $32.7_{\downarrow 1.2}$ | $53.2_{\uparrow 0.5}$ |
| | DyLAN | $95.4_{\uparrow 12.4}$ | $37.6_{\uparrow 2.8}$ | $75.4_{\uparrow 15.8}$ | $58.9_{\uparrow 10.9}$ | $50.4_{\downarrow 4.4}$ | $47.6_{\downarrow 3.4}$ | $30.8_{\downarrow 3.1}$ | $50.4_{\downarrow 2.3}$ |
| | GPTSwarm | $95.7_{\uparrow 12.7}$ | $37.9_{\uparrow 3.1}$ | $78.9_{\uparrow 19.3}$ | $58.1_{\uparrow 10.1}$ | $52.4_{\downarrow 2.4}$ | $49.3_{\downarrow 1.7}$ | $32.3_{\downarrow 1.6}$ | $51.3_{\downarrow 1.4}$ |
| | MacNet | $95.3_{\uparrow 12.3}$ | $44.2_{\uparrow 9.4}$ | $84.2_{\uparrow 24.6}$ | $57.9_{\uparrow 9.9}$ | $67.9_{\uparrow 13.1}$ | $81.6_{\uparrow 30.6}$ | $49.0_{\uparrow 15.1}$ | $75.4_{\uparrow 22.7}$ |
| | M2CL (ours) | $96.3_{\uparrow 13.3}$ | $58.0_{\uparrow 23.2}$ | $96.9_{\uparrow 37.3}$ | $92.0_{\uparrow 44.0}$ | $90.8_{\uparrow 36.0}$ | $88.9_{\uparrow 37.9}$ | $68.2_{\uparrow 34.3}$ | $82.2_{\uparrow 29.5}$ |
| Qwen-7B | Single | $61.2_{\downarrow 13.0}$ | $12.9_{\downarrow 12.0}$ | $20.2_{\downarrow 16.2}$ | $51.2_{\downarrow 11.3}$ | $23.0_{\downarrow 11.5}$ | $24.4_{\downarrow 14.2}$ | $14.8_{\downarrow 8.2}$ | $20.0_{\downarrow 7.8}$ |
| | BoN | $74.2_{\uparrow 0.0}$ | $24.9_{\uparrow 0.0}$ | $36.4_{\uparrow 0.0}$ | $62.5_{\uparrow 0.0}$ | $34.5_{\uparrow 0.0}$ | $38.6_{\uparrow 0.0}$ | $23.0_{\uparrow 0.0}$ | $27.8_{\uparrow 0.0}$ |
| | Debate | $78.1_{\uparrow 3.9}$ | $28.9_{\uparrow 4.0}$ | $41.7_{\uparrow 5.3}$ | $66.6_{\uparrow 4.1}$ | $32.3_{\downarrow 2.2}$ | $34.1_{\downarrow 4.5}$ | $23.5_{\uparrow 0.5}$ | $26.0_{\downarrow 1.8}$ |
| | DyLAN | $79.8_{\uparrow 5.6}$ | $30.7_{\uparrow 5.8}$ | $43.5_{\uparrow 7.1}$ | $67.5_{\uparrow 5.0}$ | $30.7_{\downarrow 3.8}$ | $30.0_{\downarrow 8.6}$ | $19.8_{\downarrow 3.2}$ | $23.7_{\downarrow 4.1}$ |
| | GPTSwarm | $80.9_{\uparrow 6.7}$ | $29.9_{\uparrow 5.0}$ | $43.1_{\uparrow 6.7}$ | $68.0_{\uparrow 5.5}$ | $31.8_{\downarrow 2.7}$ | $32.7_{\downarrow 5.9}$ | $23.2_{\uparrow 0.2}$ | $25.3_{\downarrow 2.5}$ |
| | MacNet | $85.1_{\uparrow 10.9}$ | $34.7_{\uparrow 9.8}$ | $50.4_{\uparrow 14.0}$ | $79.8_{\uparrow 17.3}$ | $41.6_{\uparrow 7.1}$ | $46.7_{\uparrow 8.1}$ | $28.4_{\uparrow 5.4}$ | $37.1_{\uparrow 9.3}$ |
| | M2CL (ours) | $97.5_{\uparrow 23.3}$ | $50.7_{\uparrow 25.8}$ | $80.6_{\uparrow 44.2}$ | $90.2_{\uparrow 27.7}$ | $46.8_{\uparrow 12.3}$ | $53.3_{\uparrow 14.7}$ | $42.0_{\uparrow 19.0}$ | $42.2_{\uparrow 14.4}$ |
| Qwen-14B | Single | $67.2_{\downarrow 12.5}$ | $21.6_{\downarrow 6.2}$ | $21.2_{\downarrow 11.6}$ | $56.7_{\downarrow 12.7}$ | $28.9_{\downarrow 9.5}$ | $29.9_{\downarrow 15.7}$ | $17.8_{\downarrow 9.8}$ | $24.7_{\downarrow 4.8}$ |
| | BoN | $79.7_{\uparrow 0.0}$ | $27.8_{\uparrow 0.0}$ | $32.8_{\uparrow 0.0}$ | $69.4_{\uparrow 0.0}$ | $38.4_{\uparrow 0.0}$ | $45.6_{\uparrow 0.0}$ | $27.6_{\uparrow 0.0}$ | $29.5_{\uparrow 0.0}$ |
| | Debate | $86.6_{\uparrow 6.9}$ | $31.1_{\uparrow 3.3}$ | $39.9_{\uparrow 7.1}$ | $75.8_{\uparrow 6.4}$ | $38.8_{\uparrow 0.4}$ | $42.7_{\downarrow 2.9}$ | $29.0_{\uparrow 1.4}$ | $31.7_{\uparrow 2.2}$ |
| | DyLAN | $90.2_{\uparrow 10.5}$ | $33.6_{\uparrow 5.8}$ | $42.8_{\uparrow 10.0}$ | $80.1_{\uparrow 10.7}$ | $36.2_{\downarrow 2.2}$ | $37.0_{\downarrow 8.6}$ | $24.4_{\downarrow 3.2}$ | $28.7_{\downarrow 0.8}$ |
| | GPTSwarm | $90.2_{\uparrow 10.5}$ | $33.0_{\uparrow 5.2}$ | $43.2_{\uparrow 10.4}$ | $80.4_{\uparrow 11.0}$ | $35.9_{\downarrow 2.5}$ | $36.0_{\downarrow 9.6}$ | $26.5_{\downarrow 1.1}$ | $29.0_{\downarrow 0.5}$ |
| | MacNet | $93.7_{\uparrow 14.0}$ | $37.7_{\uparrow 9.9}$ | $46.7_{\uparrow 13.9}$ | $91.2_{\uparrow 21.8}$ | $47.5_{\uparrow 9.1}$ | $56.8_{\uparrow 11.2}$ | $42.7_{\uparrow 15.1}$ | $38.4_{\uparrow 8.9}$ |
| | M2CL (ours) | $99.1_{\uparrow 19.4}$ | $61.9_{\uparrow 34.1}$ | $80.4_{\uparrow 47.6}$ | $98.7_{\uparrow 29.3}$ | $56.4_{\uparrow 18.0}$ | $65.8_{\uparrow 20.2}$ | $51.8_{\uparrow 24.2}$ | $50.9_{\uparrow 21.4}$ |
| Qwen-72B | Single | $72.5_{\downarrow 11.7}$ | $31.6_{\downarrow 19.4}$ | $34.9_{\downarrow 11.0}$ | $59.1_{\downarrow 13.1}$ | $46.7_{\downarrow 13.4}$ | $48.3_{\downarrow 17.7}$ | $29.8_{\downarrow 15.1}$ | $40.4_{\downarrow 14.0}$ |
| | BoN | $84.2_{\uparrow 0.0}$ | $51.0_{\uparrow 0.0}$ | $45.9_{\uparrow 0.0}$ | $72.2_{\uparrow 0.0}$ | $60.1_{\uparrow 0.0}$ | $66.0_{\uparrow 0.0}$ | $44.9_{\uparrow 0.0}$ | $54.4_{\uparrow 0.0}$ |
| | Debate | $91.2_{\uparrow 7.0}$ | $62.3_{\uparrow 11.3}$ | $52.4_{\uparrow 6.5}$ | $78.9_{\uparrow 6.7}$ | $62.5_{\uparrow 2.4}$ | $69.1_{\uparrow 3.1}$ | $47.6_{\uparrow 2.7}$ | $50.5_{\downarrow 3.9}$ |
| | DyLAN | $95.1_{\uparrow 10.9}$ | $66.8_{\uparrow 15.8}$ | $55.7_{\uparrow 9.8}$ | $83.9_{\uparrow 11.7}$ | $56.9_{\downarrow 3.2}$ | $59.9_{\downarrow 6.1}$ | $43.7_{\downarrow 1.2}$ | $47.7_{\downarrow 6.7}$ |
| | GPTSwarm | $94.3_{\uparrow 10.1}$ | $68.8_{\uparrow 17.8}$ | $55.2_{\uparrow 9.3}$ | $84.0_{\uparrow 11.8}$ | $59.6_{\downarrow 0.5}$ | $63.2_{\downarrow 2.8}$ | $43.0_{\downarrow 1.9}$ | $52.5_{\downarrow 1.9}$ |
| | MacNet | $98.0_{\uparrow 13.8}$ | $73.9_{\uparrow 22.9}$ | $63.2_{\uparrow 17.3}$ | $86.7_{\uparrow 14.5}$ | $74.4_{\uparrow 14.3}$ | $85.1_{\uparrow 19.1}$ | $59.5_{\uparrow 14.6}$ | $66.6_{\uparrow 12.2}$ |
| | M2CL (ours) | $99.7_{\uparrow 15.5}$ | $79.7_{\uparrow 28.7}$ | $86.0_{\uparrow 40.1}$ | $97.5_{\uparrow 25.3}$ | $92.9_{\uparrow 32.8}$ | $95.9_{\uparrow 29.9}$ | $84.2_{\uparrow 39.3}$ | $83.9_{\uparrow 29.5}$ |

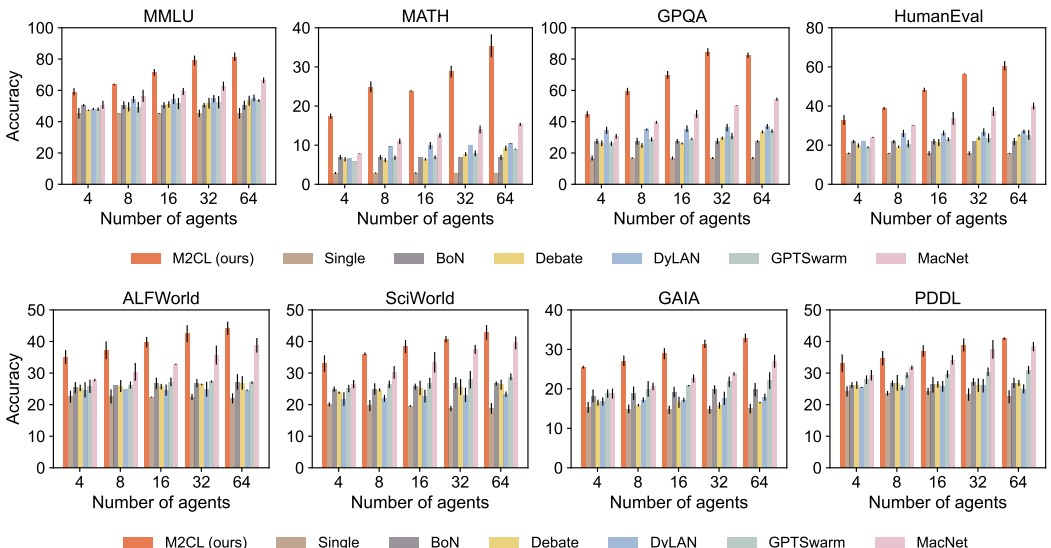

Figure 5: Performance of llama-7b as the base model with varying number of LLMs. Uncertainty intervals depict standard deviation over three seeds. M2CL exhibits higher performance and increasing tendency with more LLMs, demonstrating its great collaboration efficiency compared to existing methods. Of note, academic and agentic tasks reasoning are challenging because they require more diverse thinking perspectives and more rigorous analysis. The outperformance of M2CL reveals its capability of enabling LLMs to collaborate in changing discussion state.

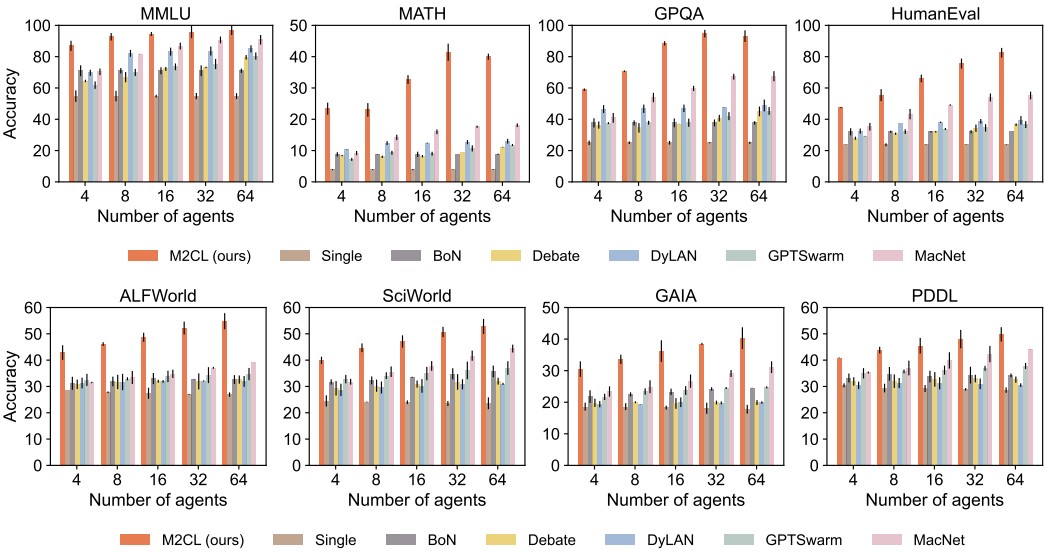

Figure 6: Performance of llama-13b as the base model with varying number of LLMs. Uncertainty intervals depict standard deviation over three seeds. M2CL exhibits higher performance and increasing tendency with more LLMs, demonstrating its great collaboration efficiency compared to existing methods.

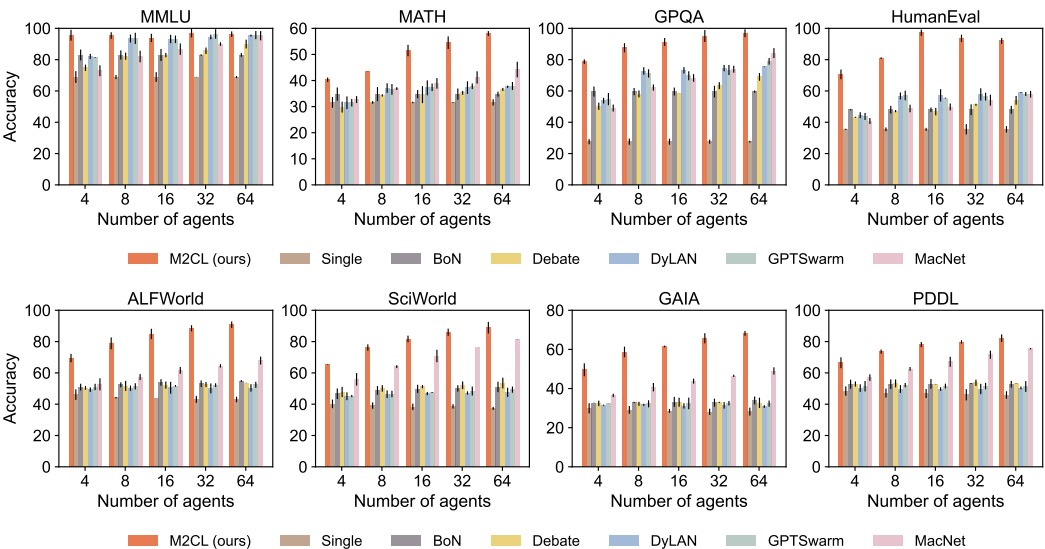

Figure 7: Performance as llama-70b as the base model with varying number of LLMs. Uncertainty intervals depict standard deviation over three seeds. M2CL exhibits higher performance and increasing tendency with more LLMs, demonstrating its great collaboration efficiency compared to existing methods.

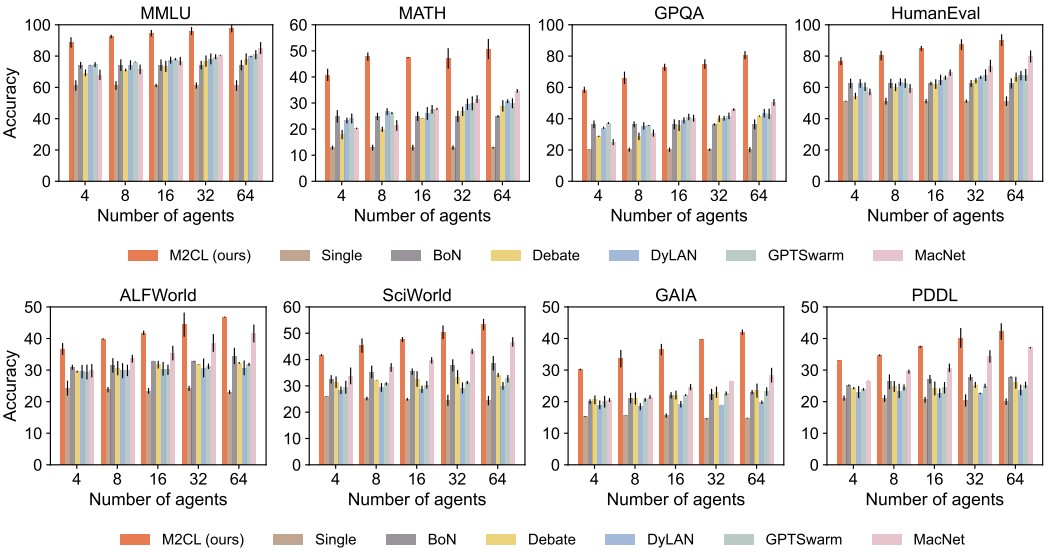

Figure 8: Performance of Qwen-7b as the base model with varying number of LLMs. Uncertainty intervals depict standard deviation over three seeds. M2CL exhibits higher performance and increasing tendency with more LLMs, demonstrating its great collaboration efficiency compared to existing methods.

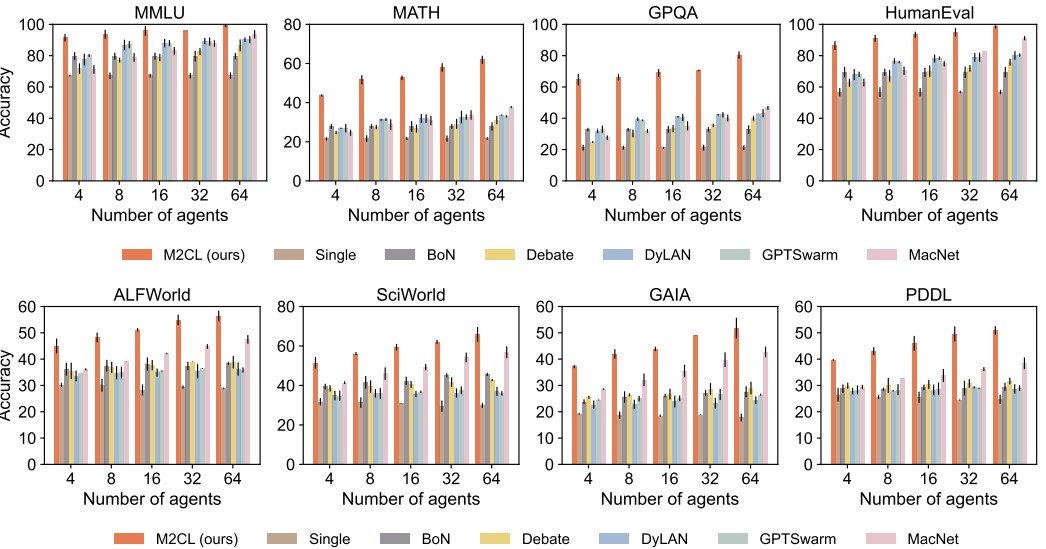

Figure 9: Performance of Qwen-14b as the base model with varying number of LLMs. Uncertainty intervals depict standard deviation over three seeds. M2CL exhibits higher performance and increasing tendency with more LLMs, demonstrating its great collaboration efficiency compared to existing methods.

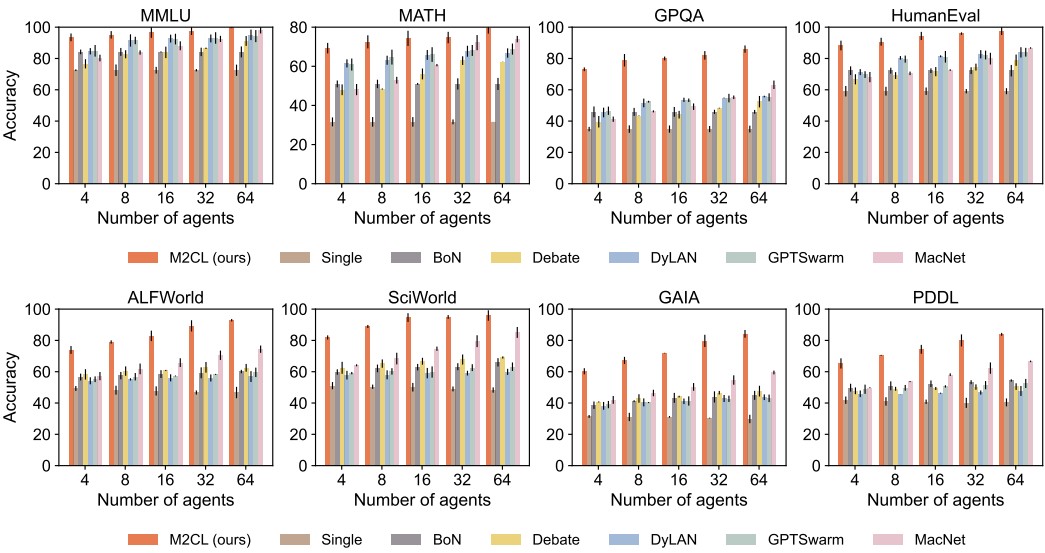

Figure 10: Performance of Qwen-72b as the base model with varying number of LLMs. Uncertainty intervals depict standard deviation over three seeds. M2CL exhibits higher performance and increasing tendency with more LLMs, demonstrating its great collaboration efficiency compared to existing methods.

### G.3 GUI AGENT

We evaluate `M2CL`'s performance on more challenging AndroidWorld, which requires GUI identification, long-horizon planning, and accurate action execution capability. Comparative results are shown in Table 9 and Fig. 11.

We observe that `M2CL` consistently outperforms existing baselines across different model scales up to $50\%$, with performance gains becoming more pronounced as the number of participating LLMs increases. The superior scalability of `M2CL` in this setting highlights its ability to exploit diverse responses and maintain consistent under complex, real-world style interactions.

| N | Model | Single | BoN | Debate | DyLAN | GPTSwarm | MacNet | M2CL |
|---|---|---|---|---|---|---|---|---|
| N=4 | 3B | $10.9_{\downarrow 6.1}$ | $17.0_{\uparrow 0.0}$ | $12.5_{\downarrow 4.5}$ | $13.0_{\downarrow 4.0}$ | $15.0_{\downarrow 2.0}$ | $16.8_{\downarrow 0.2}$ | $25.5_{\uparrow 8.5}$ |
| | 7B | $27.0_{\downarrow 11.0}$ | $38.0_{\uparrow 0.0}$ | $30.5_{\downarrow 7.5}$ | $31.0_{\downarrow 7.0}$ | $37.0_{\downarrow 1.0}$ | $39.6_{\uparrow 1.6}$ | $44.9_{\uparrow 6.9}$ |
| N=8 | 3B | $10.9_{\downarrow 6.1}$ | $17.0_{\uparrow 0.0}$ | $15.7_{\downarrow 1.3}$ | $15.2_{\downarrow 1.8}$ | $15.6_{\downarrow 1.4}$ | $19.7_{\uparrow 2.7}$ | $32.0_{\uparrow 15.0}$ |
| | 7B | $27.0_{\downarrow 11.0}$ | $38.0_{\uparrow 0.0}$ | $33.0_{\downarrow 5.0}$ | $34.5_{\downarrow 3.5}$ | $38.0_{\uparrow 0.0}$ | $44.0_{\uparrow 6.0}$ | $50.0_{\uparrow 12.0}$ |
| N=16 | 3B | $10.9_{\downarrow 6.1}$ | $17.0_{\uparrow 0.0}$ | $16.9_{\downarrow 0.1}$ | $18.4_{\uparrow 1.4}$ | $18.1_{\uparrow 1.1}$ | $22.6_{\uparrow 5.6}$ | $38.0_{\uparrow 21.0}$ |
| | 7B | $27.0_{\downarrow 11.0}$ | $38.0_{\uparrow 0.0}$ | $37.3_{\downarrow 0.7}$ | $37.8_{\downarrow 0.2}$ | $42.5_{\uparrow 4.5}$ | $47.9_{\uparrow 9.9}$ | $55.0_{\uparrow 17.0}$ |
| N=32 | 3B | $10.9_{\downarrow 6.1}$ | $17.0_{\uparrow 0.0}$ | $18.0_{\uparrow 1.0}$ | $21.5_{\uparrow 4.5}$ | $20.6_{\uparrow 3.6}$ | $25.2_{\uparrow 8.2}$ | $42.0_{\uparrow 25.0}$ |
| | 7B | $27.0_{\downarrow 11.0}$ | $38.0_{\uparrow 0.0}$ | $41.5_{\uparrow 3.5}$ | $40.0_{\uparrow 2.0}$ | $45.0_{\uparrow 7.0}$ | $52.8_{\uparrow 14.8}$ | $59.0_{\uparrow 21.0}$ |
| N=64 | 3B | $10.9_{\downarrow 6.1}$ | $17.0_{\uparrow 0.0}$ | $20.1_{\uparrow 3.1}$ | $23.6_{\uparrow 6.6}$ | $24.2_{\uparrow 7.2}$ | $28.6_{\uparrow 11.6}$ | $45.0_{\uparrow 28.0}$ |
| | 7B | $27.0_{\downarrow 11.0}$ | $38.0_{\uparrow 0.0}$ | $41.7_{\uparrow 3.7}$ | $42.2_{\uparrow 4.2}$ | $49.3_{\uparrow 11.3}$ | $55.2_{\uparrow 17.2}$ | $62.0_{\uparrow 24.0}$ |

Table 9: Accuracy with varying number of LLMs from $4$ to $64$ on AndroidWorld. We exhibit the performance advantage with `BoN` and highlight the  best  result.

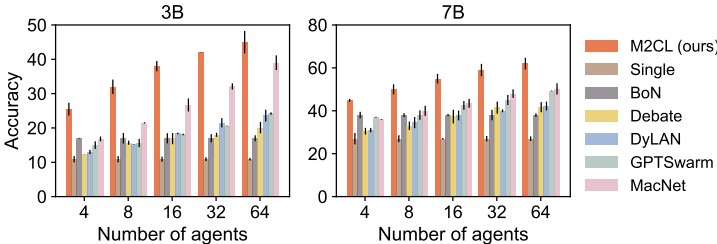

Figure 11: Performance of Qwen2.5-VL (3B and 7B) as the base model with varying number of LLMs. `M2CL` exhibits higher performance and increasing tendency with more LLMs, demonstrating its great collaboration efficiency compared to existing methods. Uncertainty intervals depict standard deviation over three seeds.

### G.4 CONTEXT CONSTRAINT

To assess the effect of context constraint, we carry out experiments by varying the context constraint $\beta$ from 0 to 10. As illustrated in Figs. 12 to 16, a larger value of $\beta$ results in a high degree of consistency among LLMs, leading them to produce similar answers. Conversely, a smaller value of $\beta$ is associated with reduced collaboration among LLMs. Therefore, it is important to adjust $\beta$ to control the degree of consistency among LLMs for better collaboration.

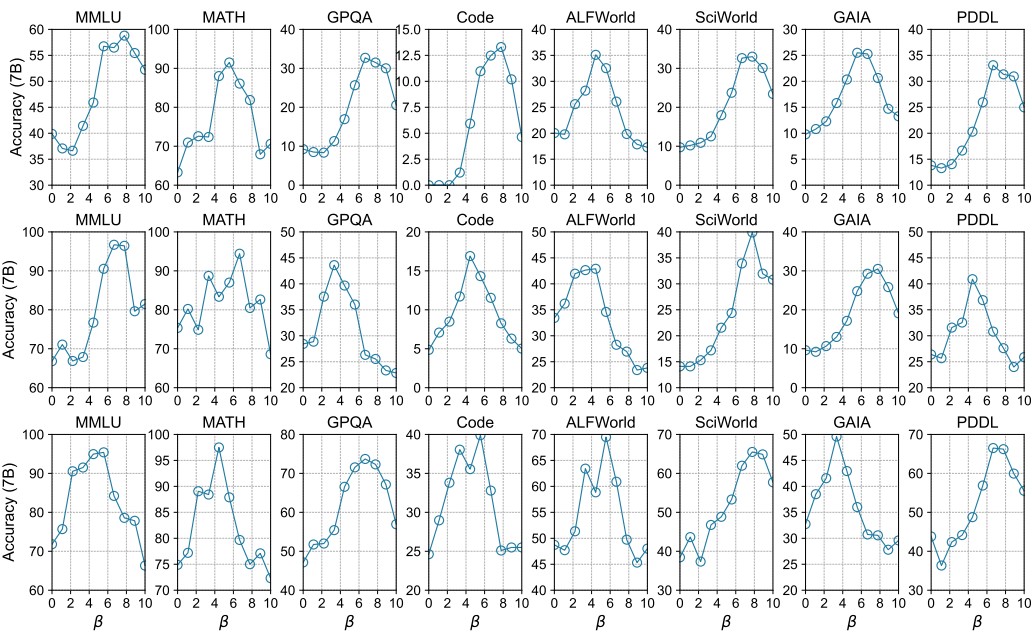

Figure 12: Performance with varying context constraint when 4 LLMs participate. All the curves display the same trend of rising first and then falling, which is consistent with our theory.

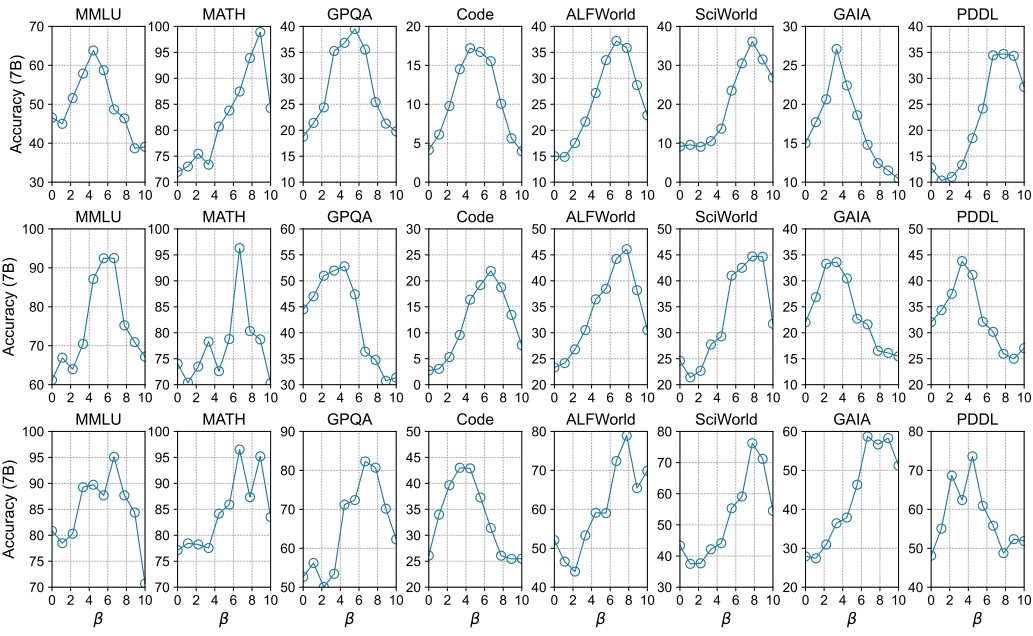

Figure 13: Performance with varying context constraint when 8 LLMs participate.

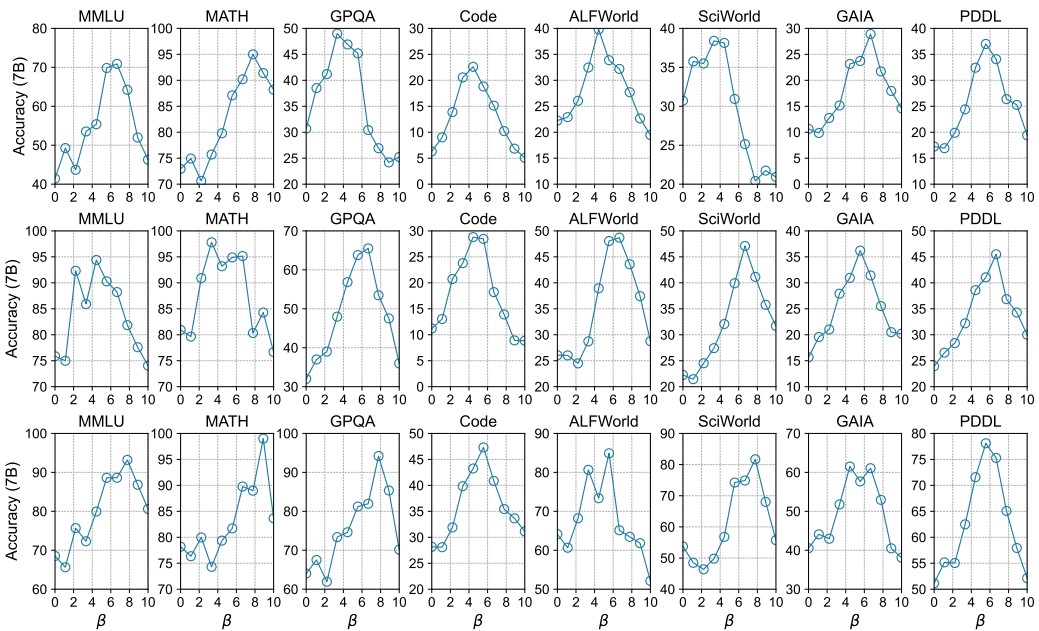

Figure 14: Performance with varying context constraint when 16 LLMs participate.

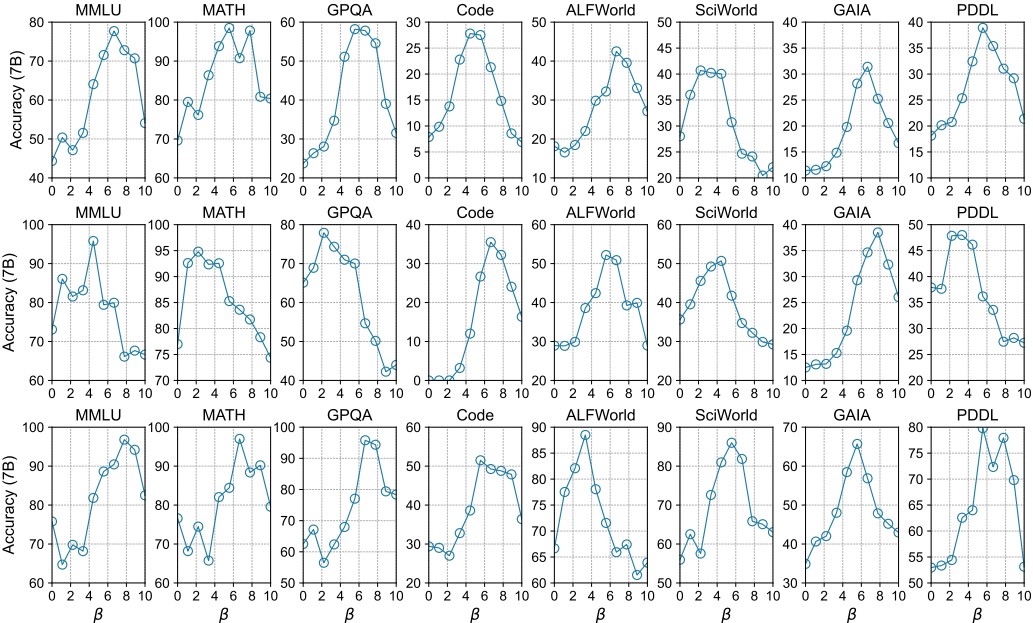

Figure 15: Performance with varying context constraint when 32 LLMs participate.

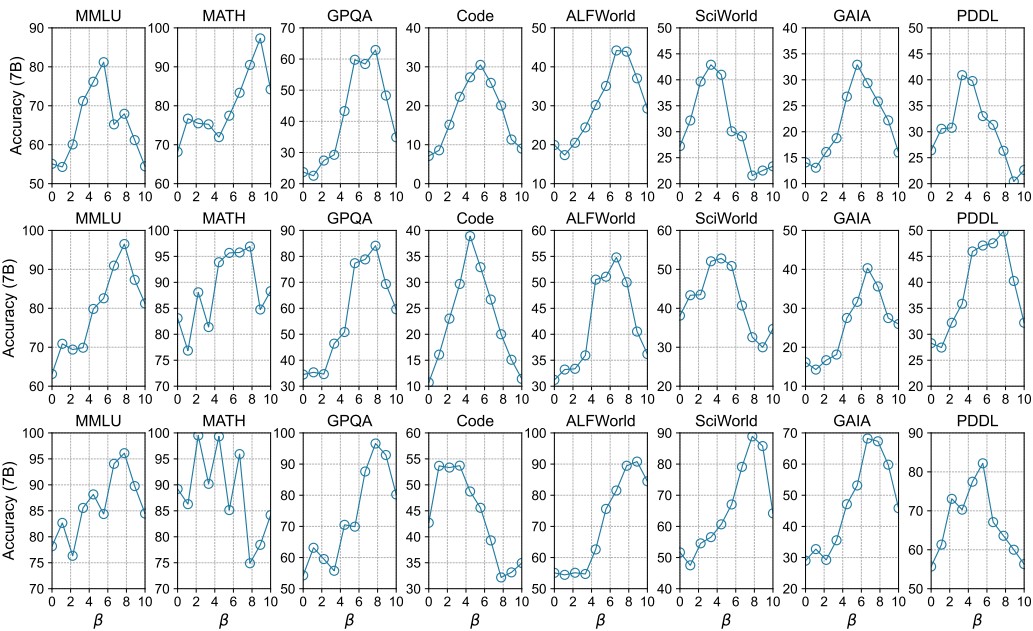

Figure 16: Performance with varying context constraint when 64 LLMs participate.

## G.5 DISCREPANCY INTENSITY BY ROUNDS

To validate that M2CL can collaborate LLMs to reach an agreement by rounds, we visualize the discrepancy intensity. As illustrated in Figs. 17 to 21, the initial discrepancy intensity of M2CL is higher as its context initialization can make the discussion more creative. The discrepancy intensity of M2CL increases faster because the dynamic adjustment of the context provides LLMs with better ability to effectively receive information from other LLMs, resulting in reduced disagreement and faster collaboration among LLMs to reach a consensus.

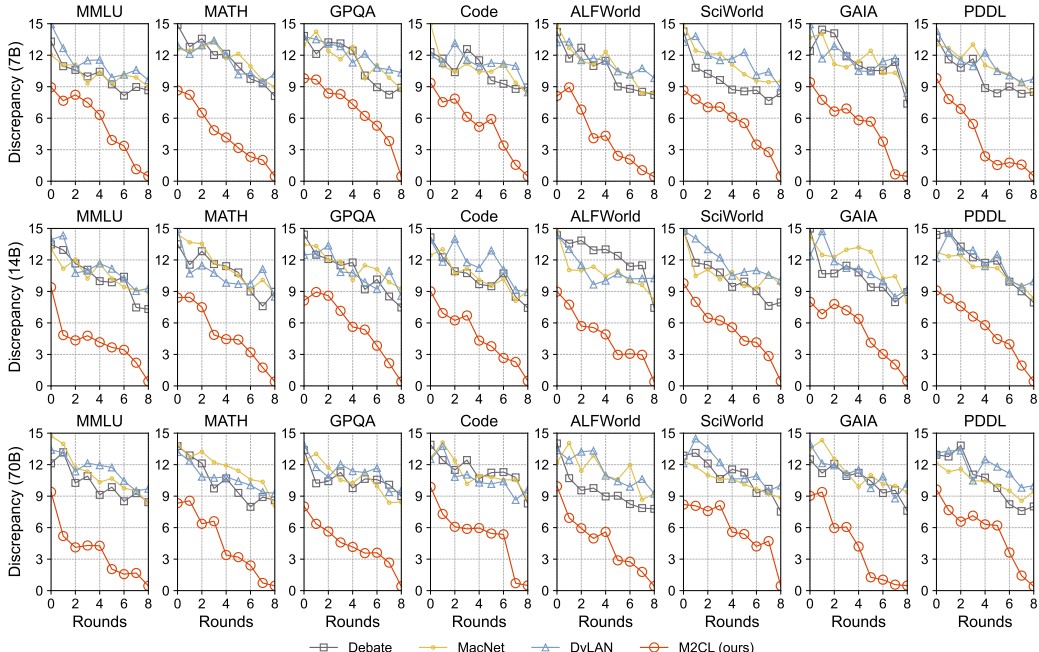

Figure 17: Comparative results on discrepancy intensity with varying model size (from top to bottom correspond to 7B, 14B, and 70B). The number of agents is set as 4. Lower values represent a lower degree of disagreement. M2CL can improve consistency with fewer rounds. Of note, M2CL displays both a lower initial value and a faster decreasing speed, indicating its capability of assigning appropriate contexts based on the given question and current discussion situation.

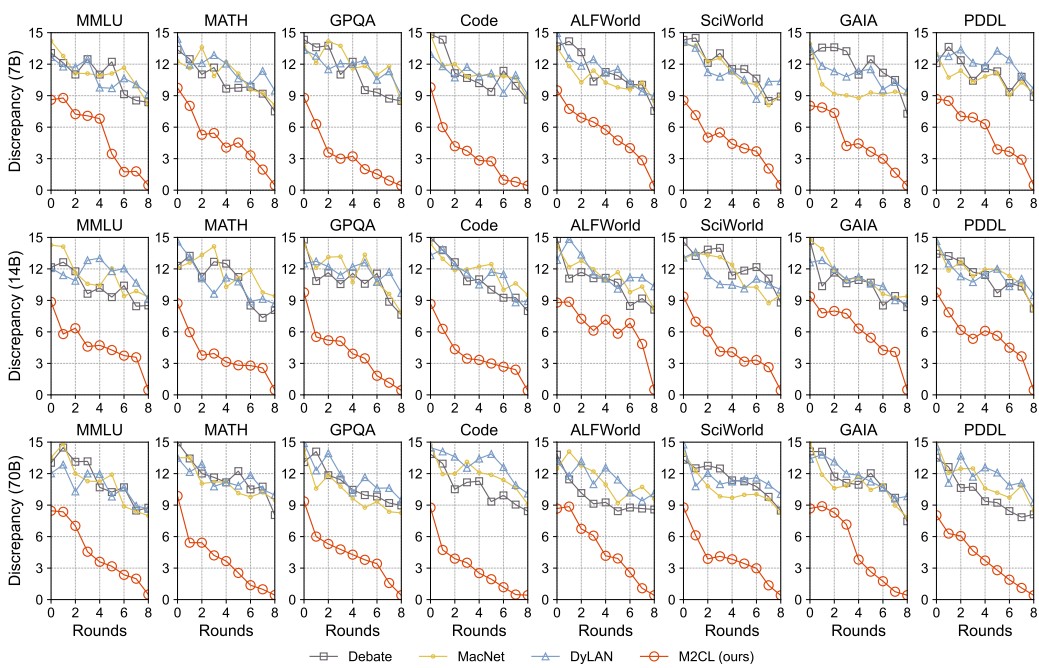

Figure 18: Comparative results on discrepancy intensity with varying model size (from top to bottom correspond to 7B, 14B, and 70B). The number of agents is set as 8. Lower values represent a lower degree of disagreement. M2CL can improve consistency with fewer rounds.

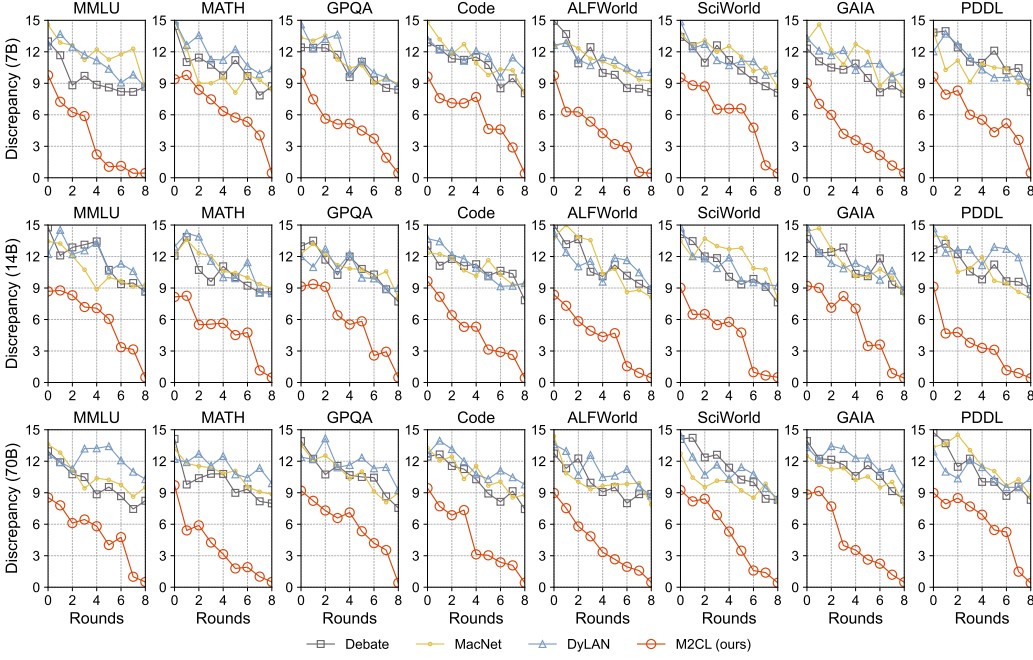

Figure 19: Comparative results on discrepancy intensity with varying model size (from top to bottom correspond to 7B, 14B, and 70B). The number of agents is set as 16. Lower values represent a lower degree of disagreement. M2CL can improve consistency with fewer rounds.

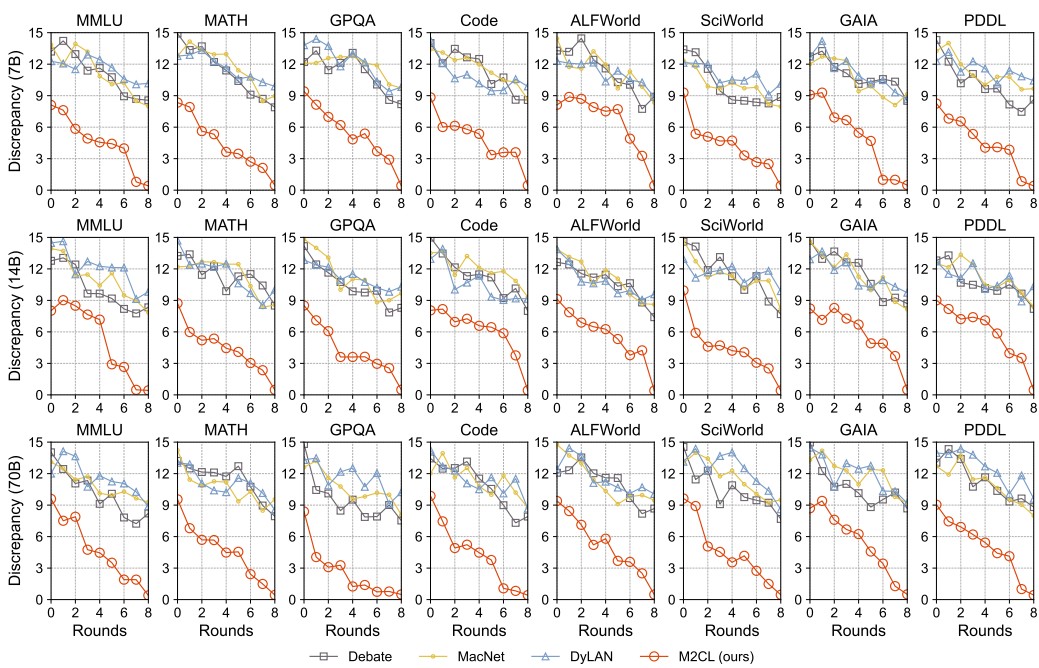

Figure 20: Comparative results on discrepancy intensity with varying model size (from top to bottom correspond to 7B, 14B, and 70B) The number of agents is set as 32. Lower values represent a lower degree of disagreement. M2CL can improve consistency with fewer rounds.

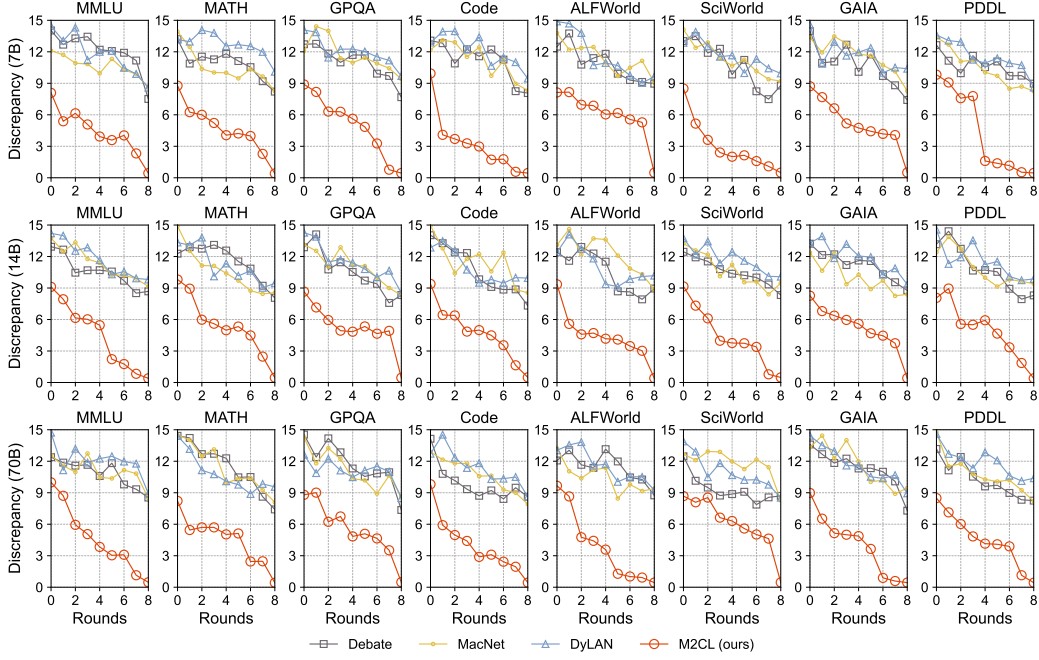

Figure 21: Comparative results on discrepancy intensity with varying model size (from top to bottom correspond to 7B, 14B, and 70B). The number of agents is set as 64. Lower values represent a lower degree of disagreement. M2CL can improve consistency with fewer rounds.

## G.6 TRANSFERABILITY OF CONTEXTS

To further study the generalization of the generated contexts, we implement the multi-agent system using GPT-4 as the base model with the context generator trained on llama-7B and compare its performance with using initial contexts. As illustrated in Table 10, the generated contexts outperform initial contexts, indicating that the trained context generator can be expanded to more models for improving overall performance through LLMs' collaboration.

| Dataset | Method | MMLU | MATH | GPQA | Code | ALFWorld | SciWorld | GAIA | PDDL |
|---------|--------|------|------|------|------|----------|----------|------|------|
| $n = 4$ | Initial | $84.3_{\downarrow 0.0}$ | $41.1_{\downarrow 0.0}$ | $33.3_{\downarrow 0.0}$ | $62.7_{\downarrow 0.0}$ | $70.2_{\downarrow 0.0}$ | $65.4_{\downarrow 0.0}$ | $58.7_{\downarrow 0.0}$ | $73.5_{\downarrow 0.0}$ |
|         | Generated | $95.0_{\uparrow 10.7}$ | $58.0_{\uparrow 16.9}$ | $52.1_{\uparrow 18.8}$ | $81.0_{\uparrow 18.3}$ | $82.0_{\uparrow 11.8}$ | $76.5_{\uparrow 11.1}$ | $65.4_{\uparrow 6.7}$ | $85.2_{\uparrow 11.7}$ |
| $n = 8$ | Initial | $86.4_{\downarrow 0.0}$ | $43.9_{\downarrow 0.0}$ | $38.8_{\downarrow 0.0}$ | $67.0_{\downarrow 0.0}$ | $72.8_{\downarrow 0.0}$ | $68.0_{\downarrow 0.0}$ | $61.0_{\downarrow 0.0}$ | $76.0_{\downarrow 0.0}$ |
|         | Generated | $98.2_{\uparrow 11.8}$ | $68.5_{\uparrow 24.6}$ | $62.5_{\uparrow 23.7}$ | $89.1_{\uparrow 22.1}$ | $82.5_{\uparrow 9.7}$ | $77.8_{\uparrow 9.8}$ | $64.3_{\uparrow 3.3}$ | $88.2_{\uparrow 12.2}$ |
| $n = 16$ | Initial | $87.9_{\downarrow 0.0}$ | $44.8_{\downarrow 0.0}$ | $40.1_{\downarrow 0.0}$ | $68.5_{\downarrow 0.0}$ | $74.3_{\downarrow 0.0}$ | $70.5_{\downarrow 0.0}$ | $63.0_{\downarrow 0.0}$ | $78.2_{\downarrow 0.0}$ |
|          | Generated | $98.5_{\uparrow 10.6}$ | $70.3_{\uparrow 25.5}$ | $64.3_{\uparrow 24.2}$ | $90.3_{\uparrow 21.8}$ | $84.5_{\uparrow 10.2}$ | $78.7_{\uparrow 8.2}$ | $66.1_{\uparrow 3.1}$ | $90.0_{\uparrow 11.8}$ |
| $n = 32$ | Initial | $89.1_{\downarrow 0.0}$ | $45.4_{\downarrow 0.0}$ | $41.5_{\downarrow 0.0}$ | $69.7_{\downarrow 0.0}$ | $75.0_{\downarrow 0.0}$ | $71.0_{\downarrow 0.0}$ | $63.5_{\downarrow 0.0}$ | $78.5_{\downarrow 0.0}$ |
|          | Generated | $98.8_{\uparrow 9.7}$ | $72.0_{\uparrow 26.6}$ | $66.0_{\uparrow 24.5}$ | $91.3_{\uparrow 21.6}$ | $86.0_{\uparrow 11.0}$ | $79.5_{\uparrow 8.5}$ | $66.8_{\uparrow 3.3}$ | $91.2_{\uparrow 12.7}$ |
| $n = 64$ | Initial | $88.5_{\downarrow 0.0}$ | $45.4_{\downarrow 0.0}$ | $41.8_{\downarrow 0.0}$ | $70.5_{\downarrow 0.0}$ | $75.2_{\downarrow 0.0}$ | $71.5_{\downarrow 0.0}$ | $64.0_{\downarrow 0.0}$ | $79.2_{\downarrow 0.0}$ |
|          | Generated | $98.9_{\uparrow 10.4}$ | $72.8_{\uparrow 27.4}$ | $66.9_{\uparrow 25.1}$ | $91.7_{\uparrow 21.2}$ | $86.5_{\uparrow 11.3}$ | $80.2_{\uparrow 8.7}$ | $67.2_{\uparrow 3.2}$ | $91.5_{\uparrow 12.3}$ |

Table 10: Performance of transferring generated contexts trained on llama-7B to GPT-4 with varying number of LLMs.

## G.7 ABLATION STUDIES AND COMPLEMENTARY EXPERIMENTS

### G.7.1 TRAINING CONVERGENCE OF CONTEXT GENERATOR

We verify the convergence of solving Problem (16) by displaying the cumulative utility. As shown in Fig. 22, it works well in all dataset and often converges in 60 training steps.

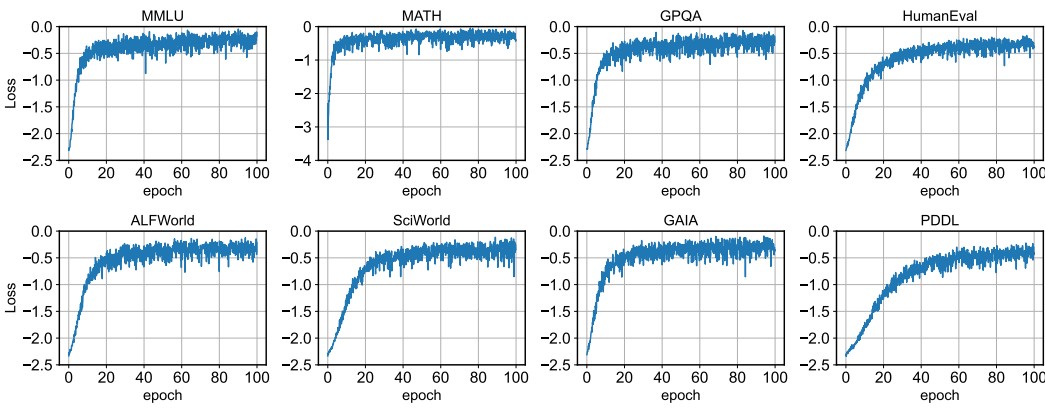

Figure 22: The value of $L(\theta)$ when training llama-7B with 8 LLMs participating.

### G.7.2 ABLATION

In this section, we assess the effect of key components by ablating them under the same setting.

Without context initialization, LLMs fail to develop specialized expertise, resulting in homogeneous policies that inefficiently duplicate effort, demonstrating poor adaptability when faced with novel questions. Tables 11 to 13 underscore the importance of context initialization before discussion to enhance foundational multi-agent capabilities.

Without tuning $\alpha$ during discussion rounds, LLMs tend to reach an agreement in the first round, leading to responses that lack creativity and diversity, which ultimately reduces problem-solving ability. Tables 11 to 13 underscore the importance of tuning $\alpha$ during discussion rounds for training.

Without context evolution, LLMs receive no guidance consider inter-LLM dependencies, making it difficult for them to fully utilize the information contributed by other LLMs. This context misalignment leads to discussion inconsistency and poor collaboration efficiency. Tables 11 to 13 highlight the necessity of context evolution for enabling effective inter-LLM collaboration and achieving consistent improvements across discussion rounds.

| Dataset | Method | MMLU | MATH | GPQA | Code | ALFWorld | SciWorld | GAIA | PDDL |
|---|---|---|---|---|---|---|---|---|---|
| $n=4$ | M2CL | $59.1_{\uparrow 10.6}$ | $17.4_{\uparrow 9.9}$ | $44.7_{\uparrow 16.7}$ | $32.9_{\uparrow 11.8}$ | $35.1_{\uparrow 8.8}$ | $33.0_{\uparrow 8.3}$ | $25.5_{\uparrow 8.1}$ | $33.1_{\uparrow 6.2}$ |
| | w/o init. | $56.4_{\uparrow 7.9}$ | $13.3_{\uparrow 5.8}$ | $44.5_{\uparrow 16.5}$ | $31.3_{\uparrow 10.2}$ | $31.8_{\uparrow 5.5}$ | $30.1_{\uparrow 5.4}$ | $23.2_{\uparrow 5.8}$ | $30.5_{\uparrow 3.6}$ |
| | w/o $\alpha$ | $53.9_{\uparrow 5.4}$ | $9.4_{\uparrow 1.9}$ | $37.8_{\uparrow 9.8}$ | $26.0_{\uparrow 4.9}$ | $22.5_{\downarrow 3.8}$ | $21.1_{\downarrow 3.6}$ | $20.3_{\uparrow 2.9}$ | $23.4_{\downarrow 3.5}$ |
| | w/o evolve | $48.5_{\downarrow 0.0}$ | $7.5_{\downarrow 0.0}$ | $28.0_{\downarrow 0.0}$ | $21.1_{\downarrow 0.0}$ | $26.3_{\downarrow 0.0}$ | $24.7_{\downarrow 0.0}$ | $17.4_{\downarrow 0.0}$ | $26.9_{\downarrow 0.0}$ |
| $n=8$ | M2CL | $63.8_{\uparrow 12.9}$ | $24.9_{\uparrow 16.8}$ | $59.3_{\uparrow 31.1}$ | $38.8_{\uparrow 17.6}$ | $37.2_{\uparrow 10.2}$ | $36.1_{\uparrow 10.3}$ | $27.1_{\uparrow 10.1}$ | $34.7_{\uparrow 7.0}$ |
| | w/o init. | $63.8_{\uparrow 12.9}$ | $17.2_{\uparrow 9.1}$ | $54.9_{\uparrow 26.7}$ | $37.2_{\uparrow 16.0}$ | $34.7_{\uparrow 7.7}$ | $33.1_{\uparrow 7.3}$ | $24.5_{\uparrow 7.5}$ | $31.5_{\uparrow 3.8}$ |
| | w/o $\alpha$ | $57.4_{\uparrow 6.5}$ | $12.9_{\uparrow 4.8}$ | $43.0_{\uparrow 14.8}$ | $29.8_{\uparrow 8.6}$ | $22.7_{\downarrow 4.3}$ | $26.8_{\uparrow 1.0}$ | $20.9_{\uparrow 3.9}$ | $24.8_{\downarrow 2.9}$ |
| | w/o evolve | $50.9_{\downarrow 0.0}$ | $8.1_{\downarrow 0.0}$ | $28.2_{\downarrow 0.0}$ | $21.2_{\downarrow 0.0}$ | $27.0_{\downarrow 0.0}$ | $25.8_{\downarrow 0.0}$ | $17.0_{\downarrow 0.0}$ | $27.7_{\downarrow 0.0}$ |
| $n=16$ | M2CL | $71.5_{\uparrow 18.5}$ | $23.9_{\uparrow 15.7}$ | $69.9_{\uparrow 39.4}$ | $48.3_{\uparrow 24.3}$ | $39.8_{\uparrow 12.7}$ | $38.4_{\uparrow 11.8}$ | $28.9_{\uparrow 11.1}$ | $37.0_{\uparrow 9.5}$ |
| | w/o init. | $70.9_{\uparrow 17.9}$ | $22.6_{\uparrow 14.4}$ | $69.4_{\uparrow 38.9}$ | $45.3_{\uparrow 21.3}$ | $36.3_{\uparrow 9.2}$ | $35.3_{\uparrow 8.7}$ | $26.2_{\uparrow 8.4}$ | $33.9_{\uparrow 6.4}$ |
| | w/o $\alpha$ | $60.5_{\uparrow 7.5}$ | $14.2_{\uparrow 6.0}$ | $46.4_{\uparrow 15.9}$ | $32.8_{\uparrow 8.8}$ | $26.6_{\downarrow 0.5}$ | $23.7_{\downarrow 2.9}$ | $22.2_{\uparrow 4.4}$ | $24.9_{\downarrow 2.6}$ |
| | w/o evolve | $53.0_{\downarrow 0.0}$ | $8.2_{\downarrow 0.0}$ | $30.5_{\downarrow 0.0}$ | $24.0_{\downarrow 0.0}$ | $27.1_{\downarrow 0.0}$ | $26.6_{\downarrow 0.0}$ | $17.8_{\downarrow 0.0}$ | $27.5_{\downarrow 0.0}$ |
| $n=32$ | M2CL | $95.8_{\uparrow 41.3}$ | $41.4_{\uparrow 31.6}$ | $94.7_{\uparrow 59.7}$ | $75.6_{\uparrow 48.8}$ | $42.5_{\uparrow 14.5}$ | $40.7_{\uparrow 13.6}$ | $31.4_{\uparrow 14.0}$ | $38.9_{\uparrow 11.4}$ |
| | w/o init. | $77.7_{\uparrow 23.2}$ | $27.8_{\uparrow 18.0}$ | $83.5_{\uparrow 48.5}$ | $53.2_{\uparrow 26.4}$ | $40.4_{\uparrow 12.4}$ | $36.7_{\uparrow 9.6}$ | $28.8_{\uparrow 11.4}$ | $35.3_{\uparrow 7.8}$ |
| | w/o $\alpha$ | $64.4_{\uparrow 9.9}$ | $18.1_{\uparrow 8.3}$ | $50.0_{\uparrow 15.0}$ | $34.8_{\uparrow 8.0}$ | $32.5_{\uparrow 4.5}$ | $28.6_{\uparrow 1.5}$ | $24.6_{\uparrow 7.2}$ | $28.4_{\uparrow 0.9}$ |
| | w/o evolve | $54.5_{\downarrow 0.0}$ | $9.8_{\downarrow 0.0}$ | $35.0_{\downarrow 0.0}$ | $26.8_{\downarrow 0.0}$ | $28.0_{\downarrow 0.0}$ | $27.1_{\downarrow 0.0}$ | $17.4_{\downarrow 0.0}$ | $27.5_{\downarrow 0.0}$ |
| $n=64$ | M2CL | $81.5_{\uparrow 25.3}$ | $35.4_{\uparrow 23.6}$ | $82.5_{\uparrow 44.1}$ | $60.6_{\uparrow 32.0}$ | $44.2_{\uparrow 15.6}$ | $42.9_{\uparrow 14.9}$ | $32.9_{\uparrow 14.8}$ | $40.9_{\uparrow 12.6}$ |
| | w/o init. | $81.2_{\uparrow 25.0}$ | $30.5_{\uparrow 18.7}$ | $80.6_{\uparrow 42.2}$ | $57.2_{\uparrow 28.6}$ | $40.7_{\uparrow 12.1}$ | $39.5_{\uparrow 11.5}$ | $30.4_{\uparrow 12.3}$ | $37.4_{\uparrow 9.1}$ |
| | w/o $\alpha$ | $63.3_{\uparrow 7.1}$ | $19.2_{\uparrow 7.4}$ | $49.7_{\uparrow 11.3}$ | $37.0_{\uparrow 8.4}$ | $31.7_{\uparrow 3.1}$ | $32.9_{\uparrow 4.9}$ | $24.6_{\uparrow 6.5}$ | $27.8_{\downarrow 0.5}$ |
| | w/o evolve | $56.2_{\downarrow 0.0}$ | $11.8_{\downarrow 0.0}$ | $38.4_{\downarrow 0.0}$ | $28.6_{\downarrow 0.0}$ | $28.6_{\downarrow 0.0}$ | $28.0_{\downarrow 0.0}$ | $18.1_{\downarrow 0.0}$ | $28.3_{\downarrow 0.0}$ |

Table 11: Ablation study on context initialization, tuning $\alpha$, and context evolution when using llama-7B with varying number of LLMs.

| Dataset | Method | MMLU | MATH | GPQA | Code | ALFWorld | SciWorld | GAIA | PDDL |
|---|---|---|---|---|---|---|---|---|---|
| $n=4$ | M2CL (ours) | $86.9_{\uparrow 20.2}$ | $23.4_{\uparrow 13.5}$ | $58.9_{\uparrow 20.3}$ | $47.5_{\uparrow 17.6}$ | $42.9_{\uparrow 10.9}$ | $39.9_{\uparrow 9.5}$ | $30.5_{\uparrow 9.6}$ | $40.9_{\uparrow 8.1}$ |
| | w/o init. | $86.7_{\uparrow 20.0}$ | $16.9_{\uparrow 7.0}$ | $57.9_{\uparrow 19.3}$ | $45.2_{\uparrow 15.3}$ | $40.7_{\uparrow 8.7}$ | $47.8_{\uparrow 17.4}$ | $33.7_{\uparrow 12.8}$ | $37.0_{\uparrow 4.2}$ |
| | w/o $\alpha$ | $81.5_{\uparrow 14.8}$ | $13.2_{\uparrow 3.3}$ | $50.9_{\uparrow 12.3}$ | $37.3_{\uparrow 7.4}$ | $32.1_{\uparrow 0.1}$ | $31.9_{\uparrow 1.5}$ | $24.2_{\uparrow 3.3}$ | $24.3_{\downarrow 8.5}$ |
| | w/o evolve | $66.7_{\downarrow 0.0}$ | $9.9_{\downarrow 0.0}$ | $38.6_{\downarrow 0.0}$ | $29.9_{\downarrow 0.0}$ | $32.0_{\downarrow 0.0}$ | $30.4_{\downarrow 0.0}$ | $20.9_{\downarrow 0.0}$ | $32.8_{\downarrow 0.0}$ |
| $n=8$ | M2CL (ours) | $92.7_{\uparrow 23.2}$ | $23.0_{\uparrow 13.5}$ | $70.7_{\uparrow 32.6}$ | $55.4_{\uparrow 22.1}$ | $46.1_{\uparrow 13.0}$ | $44.7_{\uparrow 13.0}$ | $33.6_{\uparrow 12.2}$ | $43.8_{\uparrow 10.6}$ |
| | w/o init. | $92.5_{\uparrow 23.0}$ | $21.9_{\uparrow 12.4}$ | $70.3_{\uparrow 32.2}$ | $53.3_{\uparrow 20.0}$ | $45.3_{\uparrow 12.2}$ | $51.8_{\uparrow 20.1}$ | $38.4_{\uparrow 17.0}$ | $39.9_{\uparrow 6.7}$ |
| | w/o $\alpha$ | $85.4_{\uparrow 15.9}$ | $16.9_{\uparrow 7.4}$ | $53.9_{\uparrow 15.8}$ | $43.3_{\uparrow 10.0}$ | $34.7_{\uparrow 1.6}$ | $37.6_{\uparrow 5.9}$ | $26.2_{\uparrow 4.8}$ | $29.0_{\downarrow 4.2}$ |
| | w/o evolve | $69.5_{\downarrow 0.0}$ | $9.5_{\downarrow 0.0}$ | $38.1_{\downarrow 0.0}$ | $33.3_{\downarrow 0.0}$ | $33.1_{\downarrow 0.0}$ | $31.7_{\downarrow 0.0}$ | $21.4_{\downarrow 0.0}$ | $33.2_{\downarrow 0.0}$ |
| $n=16$ | M2CL (ours) | $94.9_{\uparrow 22.9}$ | $27.3_{\uparrow 16.2}$ | $73.2_{\uparrow 33.8}$ | $61.5_{\uparrow 25.3}$ | $49.7_{\uparrow 15.3}$ | $48.9_{\uparrow 15.6}$ | $36.0_{\uparrow 12.9}$ | $45.2_{\uparrow 10.2}$ |
| | w/o init. | $94.7_{\uparrow 22.7}$ | $25.0_{\uparrow 13.9}$ | $72.3_{\uparrow 32.9}$ | $60.4_{\uparrow 24.2}$ | $48.5_{\uparrow 14.1}$ | $54.5_{\uparrow 21.2}$ | $39.1_{\uparrow 16.0}$ | $41.8_{\uparrow 6.8}$ |
| | w/o $\alpha$ | $87.2_{\uparrow 15.2}$ | $18.7_{\uparrow 7.6}$ | $57.7_{\uparrow 18.3}$ | $47.6_{\uparrow 11.4}$ | $36.3_{\uparrow 1.9}$ | $39.9_{\uparrow 6.6}$ | $27.0_{\uparrow 3.9}$ | $31.0_{\downarrow 4.0}$ |
| | w/o evolve | $72.0_{\downarrow 0.0}$ | $11.1_{\downarrow 0.0}$ | $39.4_{\downarrow 0.0}$ | $36.2_{\downarrow 0.0}$ | $34.4_{\downarrow 0.0}$ | $33.3_{\downarrow 0.0}$ | $23.1_{\downarrow 0.0}$ | $35.0_{\downarrow 0.0}$ |
| $n=32$ | M2CL (ours) | $95.5_{\uparrow 21.8}$ | $27.9_{\uparrow 15.3}$ | $74.0_{\uparrow 32.3}$ | $63.1_{\uparrow 24.6}$ | $50.8_{\uparrow 15.5}$ | $49.9_{\uparrow 14.7}$ | $36.6_{\uparrow 12.3}$ | $46.7_{\uparrow 10.5}$ |
| | w/o init. | $95.3_{\uparrow 21.6}$ | $25.7_{\uparrow 13.1}$ | $73.0_{\uparrow 31.3}$ | $62.0_{\uparrow 23.5}$ | $49.4_{\uparrow 14.1}$ | $55.2_{\uparrow 20.0}$ | $39.8_{\uparrow 15.5}$ | $43.0_{\uparrow 6.8}$ |
| | w/o $\alpha$ | $87.6_{\uparrow 13.9}$ | $19.1_{\uparrow 6.5}$ | $58.5_{\uparrow 16.8}$ | $48.1_{\uparrow 9.6}$ | $36.7_{\uparrow 1.4}$ | $40.5_{\uparrow 5.3}$ | $27.4_{\uparrow 3.1}$ | $31.7_{\downarrow 4.5}$ |
| | w/o evolve | $73.7_{\downarrow 0.0}$ | $12.6_{\downarrow 0.0}$ | $41.7_{\downarrow 0.0}$ | $38.5_{\downarrow 0.0}$ | $35.3_{\downarrow 0.0}$ | $35.2_{\downarrow 0.0}$ | $24.3_{\downarrow 0.0}$ | $36.2_{\downarrow 0.0}$ |
| $n=64$ | M2CL (ours) | $95.8_{\uparrow 21.6}$ | $28.1_{\uparrow 14.9}$ | $74.8_{\uparrow 31.3}$ | $63.5_{\uparrow 23.5}$ | $51.2_{\uparrow 15.2}$ | $50.3_{\uparrow 13.9}$ | $36.9_{\uparrow 11.9}$ | $47.1_{\uparrow 10.3}$ |
| | w/o init. | $95.6_{\uparrow 21.4}$ | $25.9_{\uparrow 12.7}$ | $73.5_{\uparrow 30.0}$ | $62.2_{\uparrow 22.2}$ | $49.6_{\uparrow 13.6}$ | $55.6_{\uparrow 19.2}$ | $40.1_{\uparrow 15.1}$ | $43.3_{\uparrow 6.5}$ |
| | w/o $\alpha$ | $87.8_{\uparrow 13.6}$ | $19.4_{\uparrow 6.2}$ | $59.1_{\uparrow 15.6}$ | $48.6_{\uparrow 8.6}$ | $37.0_{\uparrow 1.0}$ | $40.8_{\uparrow 4.4}$ | $27.7_{\uparrow 2.7}$ | $32.0_{\downarrow 4.8}$ |
| | w/o evolve | $74.2_{\downarrow 0.0}$ | $13.2_{\downarrow 0.0}$ | $43.5_{\downarrow 0.0}$ | $40.0_{\downarrow 0.0}$ | $36.0_{\downarrow 0.0}$ | $36.4_{\downarrow 0.0}$ | $25.0_{\downarrow 0.0}$ | $36.8_{\downarrow 0.0}$ |

Table 12: Ablation study on context initialization, tuning $\alpha$, and context evolution when using llama-13B with varying number of LLMs.

| Dataset | Method | MMLU | MATH | GPQA | Code | ALFWorld | SciWorld | GAIA | PDDL |
|---|---|---|---|---|---|---|---|---|---|
| $n = 4$ | M2CL (ours) | $95.6_{\uparrow 18.7}$ | $40.3_{\uparrow 9.4}$ | $78.7_{\uparrow 25.6}$ | $70.6_{\uparrow 24.7}$ | $69.5_{\uparrow 17.2}$ | $65.5_{\uparrow 15.9}$ | $49.6_{\uparrow 15.5}$ | $66.5_{\uparrow 12.2}$ |
| | w/o init. | $95.4_{\uparrow 18.5}$ | $39.9_{\uparrow 9.0}$ | $88.5_{\uparrow 35.4}$ | $67.8_{\uparrow 21.9}$ | $64.8_{\uparrow 12.5}$ | $62.9_{\uparrow 13.3}$ | $47.0_{\uparrow 12.9}$ | $62.6_{\uparrow 8.3}$ |
| | w/o $\alpha$ | $86.7_{\uparrow 9.8}$ | $36.9_{\uparrow 6.0}$ | $72.3_{\uparrow 19.2}$ | $53.9_{\uparrow 8.0}$ | $43.3_{\uparrow 9.0}$ | $47.1_{\downarrow 2.5}$ | $38.0_{\downarrow -0.1}$ | $40.3_{\downarrow 14.0}$ |
| | w/o evolve | $76.9_{\downarrow 0.0}$ | $30.9_{\downarrow 0.0}$ | $53.1_{\downarrow 0.0}$ | $45.9_{\downarrow 0.0}$ | $52.3_{\downarrow 0.0}$ | $49.6_{\downarrow 0.0}$ | $34.1_{\downarrow 0.0}$ | $54.3_{\downarrow 0.0}$ |
| $n = 8$ | M2CL (ours) | $95.4_{\uparrow 11.9}$ | $43.5_{\uparrow 8.3}$ | $87.6_{\uparrow 26.6}$ | $81.0_{\uparrow 30.5}$ | $78.9_{\uparrow 24.5}$ | $76.2_{\uparrow 23.5}$ | $58.7_{\uparrow 23.8}$ | $73.6_{\uparrow 18.3}$ |
| | w/o init. | $95.1_{\uparrow 11.6}$ | $43.0_{\uparrow 7.8}$ | $86.3_{\uparrow 25.3}$ | $80.0_{\uparrow 29.5}$ | $75.8_{\uparrow 21.4}$ | $71.7_{\uparrow 18.5}$ | $56.1_{\uparrow 21.2}$ | $71.3_{\uparrow 16.0}$ |
| | w/o $\alpha$ | $94.0_{\uparrow 10.5}$ | $39.3_{\uparrow 4.1}$ | $78.0_{\uparrow 17.0}$ | $63.8_{\uparrow 13.3}$ | $49.3_{\downarrow 5.1}$ | $49.0_{\downarrow 3.7}$ | $42.6_{\uparrow 7.7}$ | $47.8_{\downarrow 7.5}$ |
| | w/o evolve | $83.5_{\downarrow 0.0}$ | $35.2_{\downarrow 0.0}$ | $61.0_{\downarrow 0.0}$ | $50.5_{\downarrow 0.0}$ | $54.4_{\downarrow 0.0}$ | $52.7_{\downarrow 0.0}$ | $34.9_{\downarrow 0.0}$ | $55.3_{\downarrow 0.0}$ |
| $n = 16$ | M2CL (ours) | $93.9_{\uparrow 9.9}$ | $51.5_{\uparrow 15.2}$ | $91.3_{\uparrow 29.5}$ | $97.2_{\uparrow 45.4}$ | $84.9_{\uparrow 29.4}$ | $81.7_{\uparrow 27.4}$ | $61.5_{\uparrow 25.6}$ | $78.1_{\uparrow 22.8}$ |
| | w/o init. | $93.2_{\uparrow 9.2}$ | $47.3_{\uparrow 11.0}$ | $90.9_{\uparrow 29.1}$ | $96.9_{\uparrow 45.1}$ | $81.3_{\uparrow 25.8}$ | $79.3_{\uparrow 23.0}$ | $56.6_{\uparrow 20.7}$ | $73.5_{\uparrow 18.2}$ |
| | w/o $\alpha$ | $93.2_{\uparrow 9.2}$ | $40.5_{\uparrow 4.2}$ | $78.6_{\uparrow 16.8}$ | $72.2_{\uparrow 20.4}$ | $64.2_{\downarrow -8.7}$ | $64.8_{\uparrow 10.5}$ | $47.0_{\uparrow 11.1}$ | $59.3_{\uparrow 4.0}$ |
| | w/o evolve | $84.0_{\downarrow 0.0}$ | $36.3_{\downarrow 0.0}$ | $61.8_{\downarrow 0.0}$ | $51.8_{\downarrow 0.0}$ | $55.5_{\downarrow 0.0}$ | $54.3_{\downarrow 0.0}$ | $35.9_{\downarrow 0.0}$ | $55.3_{\downarrow 0.0}$ |
| $n = 32$ | M2CL (ours) | $97.0_{\uparrow 10.2}$ | $54.5_{\uparrow 17.3}$ | $95.1_{\uparrow 28.5}$ | $93.7_{\uparrow 38.2}$ | $88.5_{\uparrow 32.3}$ | $86.0_{\uparrow 30.5}$ | $65.7_{\uparrow 29.5}$ | $79.8_{\uparrow 23.5}$ |
| | w/o init. | $96.8_{\uparrow 10.0}$ | $51.5_{\uparrow 14.3}$ | $94.8_{\uparrow 27.9}$ | $93.4_{\uparrow 37.9}$ | $85.8_{\uparrow 29.6}$ | $81.8_{\uparrow 26.3}$ | $61.2_{\uparrow 24.3}$ | $77.1_{\uparrow 20.8}$ |
| | w/o $\alpha$ | $95.2_{\uparrow 8.4}$ | $44.0_{\uparrow 6.8}$ | $83.9_{\uparrow 17.3}$ | $68.6_{\uparrow 13.1}$ | $57.3_{\downarrow -1.1}$ | $57.9_{\uparrow 2.4}$ | $50.9_{\uparrow 14.7}$ | $61.3_{\uparrow 5.0}$ |
| | w/o evolve | $86.8_{\downarrow 0.0}$ | $37.2_{\downarrow 0.0}$ | $66.6_{\downarrow 0.0}$ | $55.5_{\downarrow 0.0}$ | $56.2_{\downarrow 0.0}$ | $55.5_{\downarrow 0.0}$ | $36.2_{\downarrow 0.0}$ | $56.3_{\downarrow 0.0}$ |
| $n = 64$ | M2CL (ours) | $96.3_{\uparrow 5.7}$ | $58.0_{\uparrow 19.3}$ | $96.9_{\uparrow 25.1}$ | $92.0_{\uparrow 34.2}$ | $90.8_{\uparrow 33.5}$ | $88.9_{\uparrow 31.9}$ | $68.2_{\uparrow 32.0}$ | $82.2_{\uparrow 26.1}$ |
| | w/o init. | $96.1_{\uparrow 5.5}$ | $53.7_{\uparrow 15.0}$ | $96.9_{\uparrow 25.1}$ | $91.7_{\uparrow 33.9}$ | $88.7_{\uparrow 31.4}$ | $85.5_{\uparrow 28.5}$ | $65.0_{\uparrow 28.8}$ | $77.5_{\uparrow 21.4}$ |
| | w/o $\alpha$ | $95.7_{\uparrow 5.1}$ | $43.3_{\uparrow 4.6}$ | $82.8_{\uparrow 11.0}$ | $69.6_{\uparrow 11.8}$ | $66.9_{\downarrow -9.6}$ | $70.3_{\uparrow 13.3}$ | $41.8_{\uparrow 5.6}$ | $55.6_{\uparrow -0.5}$ |
| | w/o evolve | $90.6_{\downarrow 0.0}$ | $38.7_{\downarrow 0.0}$ | $71.8_{\downarrow 0.0}$ | $57.8_{\downarrow 0.0}$ | $57.3_{\downarrow 0.0}$ | $57.0_{\downarrow 0.0}$ | $36.2_{\downarrow 0.0}$ | $56.1_{\downarrow 0.0}$ |

Table 13: Ablation study on context initialization, tuning $\alpha$, and context evolution when using llama-70B with varying number of LLMs.

### G.7.3 RUN-TIME

To demonstrate the efficiency of our context initialization, we verify its runtime with varying the number of LLMs. Then, we evaluate the runtime of M2CL compared with baseline algorithms

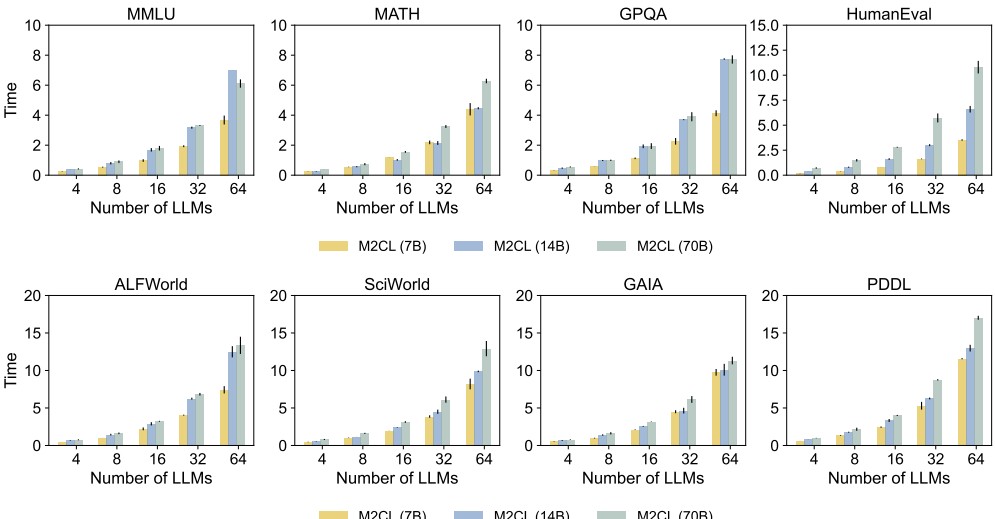

Figure 23: Runtime of initialization. Uncertainty intervals depict standard deviation over three seeds.

for average testing time, utilizing the same model size on 8 NVIDIA H800 GPUs. As illustrated by Fig. 24, the runtime of M2CL is slightly longer than other multi-LLM discussion methods as the runtime of context generators is negligible compared with the inference time of LLM, which substantiates the low computational cost of M2CL.

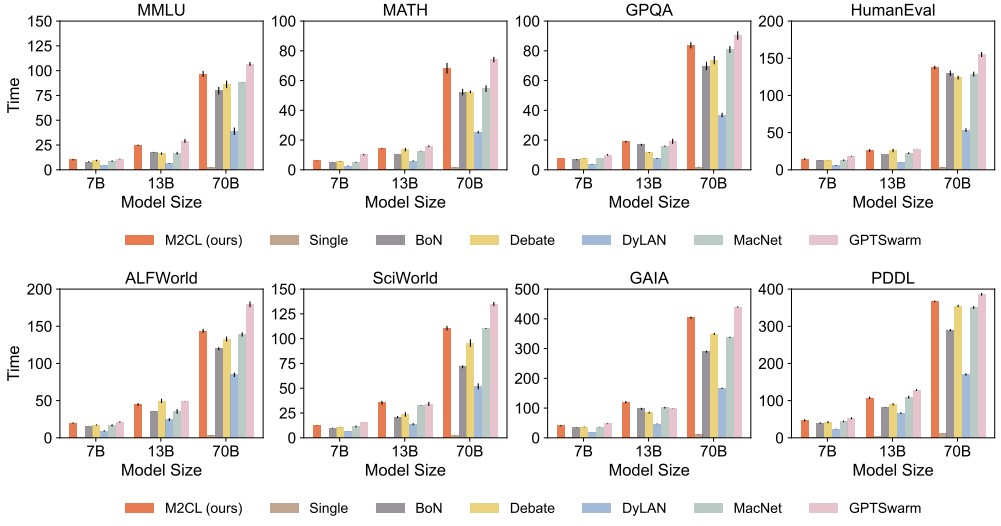

Figure 24: Runtime when varying the size of the LLama series models. The number of LLMs is 8. Uncertainty intervals depict standard deviation over three seeds.

## H    CASE STUDY

We used a problem from the MATH Hendrycks et al. (2021). The number of LLMs is set as 8. For each LLM, we present their instructions, responses, and final answers for 4 discussion rounds.

### H.1    CASE STUDY OF M2CL (OURS)

We provide the case of M2CL solving the problem. As illustrated in Figs. 25 to 28, we observe that the generated contexts evolve progressively to enforce stricter collaboration among LLMs. Initially, the instructions merely encourage LLMs to pay attention to others' responses. In subsequent rounds, they guide LLMs to cross-check each other's answers and eventually require reaching full agreement. Correspondingly, the LLMs' outputs transition from diverse, potentially conflicting answers to a single, consistent solution. This behavior demonstrates that our method effectively modulates context evolution to guide toward consensus without sacrificing initial creativity.

**Problem:** In the diagram, square ABCD has sides of length 4, and $\triangle ABE$ is equilateral. Line segments BE and AC intersect at P. Point Q is on BC so that PQ is perpendicular to BC and PQ=x. A=(0,0); B=(4,0); C=(4,4); D=(0,4); E=(2,3.464); P=(2.535,2.535); Q=(4,2.535); Determine the measure of $\angle BPC$.
**Solution:** Since $\triangle ABE$ is equilateral, we know that $\angle ABE = 60°$. Therefore, $\angle PBC = \angle ABC - \angle ABE = 90° - 60° = 30°$. Since AB=BC, we know that $\triangle ABC$ is a right isosceles triangle and $\angle BAC = \angle BCA = 45°$. Then, $\angle BCP = \angle BCA = 45°$ and $\angle BPC = 180° - \angle PBC - \angle BCP = 180° - 30° - 45° = \boxed{105°}$.

**Agent 1**

**Instruction:** You are an expert in Geometric Constructions. Your job is to use the given lengths and properties of shapes like squares and equilateral triangles to visualize and understand the arrangement of points, lines, and angles. It helps in setting up geometric relationships based on the diagram and given conditions.
**Response:** Start with $\triangle ABE$ as equilateral, making $\angle ABE = 60°$. In square ABCD, $\angle ABC = 90°$, so $\angle PBC = \angle ABC - \angle ABE = 90° - 60° = 30°$. Since AB = BC, $\triangle ABC$ is a right isosceles triangle, meaning $\angle BAC = \angle BCA = 45°$. Therefore, $\angle BCP = 45°$. Calculate $\angle BPC = 180° - 30° - 45° = 105°$.
**Final Answer:** $\boxed{105°}$

**Agent 2**

**Instruction:** You are an expert in Analytic Geometry. Your job is to represent the positions of points, derive equations for lines, and calculate intersection points and angles. By solving for the coordinates of point P and applying vector or trigonometric methods, one can determine the angle between lines BP and CP.
**Response:** Assign coordinates $A = (0, 0)$, $B = (4, 0)$, $C = (4, 4)$, $E = (2, 2\sqrt{3})$. The line BE has slope $m_{BE} = \sqrt{3}$, and equation $y = \sqrt{3}(x - 4)$. The line AC has slope $m_{AC} = 1$, and equation $y = x + 4$. Solve for intersection P: $\sqrt{3}(x - 4) = x + 4$. Solve for x = 4. Substitute x = 4 to find P = (4, 0). Now calculate the slopes of BP and CP, use $\tan \theta$ for $\angle BPC$. After some calculations, conclude that $\angle BPC = 120°$.
**Final Answer:** $\boxed{120°}$

**Agent 3**

**Instruction:** You are an expert in Vector Mathematics. Your job is to provide a powerful tool for describing spatial relationships between points. In this case, vectors are used to represent the directions from points B and C to P, allowing the calculation of the angle between the two vectors.
**Response:** Represent vectors $\overrightarrow{BP} = (1.465, 2.535)$ and $\overrightarrow{CP} = (1.465, 1.465)$. Use the dot product formula: $\overrightarrow{BP} \cdot \overrightarrow{CP} = 1.465 \cdot 1.465 + 2.535 \cdot 1.465$. After calculating the dot product and magnitudes, use $\cos \theta = \frac{\text{dot product}}{\text{magnitudes}}$. Solve for $\theta$ and conclude $\angle BPC = 90°$.
**Final Answer:** $\boxed{90°}$

**Agent 4**

**Instruction:** You are an expert in Symmetry and Transformational Geometry. Your job is to explore the geometric properties of the figure, identifying symmetrical properties and transformations that can simplify the calculation of angles, such as recognizing the symmetry of the square and its impact on angles.
**Response:** In square ABCD and with $\triangle ABE$ being equilateral, $\angle ABE = 60°$. Subtract this from $\angle ABC = 90°$ to get $\angle PBC = 30°$. Use the symmetry of $\triangle ABC$ (since AB = BC) and determine that $\angle BCP = 60°$. Conclude $\angle BPC = 180° - 30° - 60° = 90°$.
**Final Answer:** $\boxed{90°}$

**Agent 5**

**Instruction:** You are an expert in Trigonometry. Your job is to solve problems involving angles and distances. It enables the use of trigonometric ratios and formulas, such as the tangent or sine rule, to find unknown angles and sides in geometric configurations.
**Response:** Slope of $BE = \sqrt{3}$, and slope of $AC = 1$. Use the formula for the angle between two lines: $\tan \theta = \left| \frac{\sqrt{3}+1}{1-\sqrt{3}} \right|$. Solve for $\theta$ using arc-tangent and conclude $\angle BPC = 120°$.
**Final Answer:** $\boxed{120°}$

**Agent 6**

**Instruction:** You are an expert in Computational Geometry and Visualization. Your job is to plot the points and lines on a coordinate plane, often using software tools, to visually interpret the geometric configuration and help confirm or measure angles directly.
**Response:** Plot the equations of BE and AC using graphing software. Measure $\angle BPC$ at the intersection point P. Upon measuring, find $\angle BPC = 90°$.
**Final Answer:** $\boxed{90°}$

**Agent 7**

**Instruction:** You are an expert in Geometric Analysis and Area-Based Methods. Your job is to rely on known formulas for areas of triangles and other polygons. By relating the geometry of the shape to its area, one can deduce missing information, such as angles, based on geometric properties.
**Response:** Use the triangle area formula to calculate the area of $\triangle BPC$ from the coordinates of B, C, and P: Area $= \frac{1}{2}|x_1(y_2 - y_3) + x_2(y_3 - y_1) + x_3(y_1 - y_2)|$. Compare the calculated area with known triangle properties.
Use the side lengths and triangle geometry to determine $\angle BPC = 120°$.
**Final Answer:** $\boxed{120°}$

**Agent 8**

**Instruction:** You are an expert in Algebraic Geometry and Triangle Theory. Your job is to apply algebraic techniques, such as the Law of Cosines, to solve for angles when side lengths are known. This method connects algebraic formulas directly with geometric interpretation to determine angles in triangles like $\triangle BPC$.
**Response:** In $\triangle BPC$, calculate side lengths BP, CP, and BC = 4. Use the Law of Cosines: $\cos \angle BPC = \frac{BP^2 + CP^2 - BC^2}{2 BP \cdot CP}$. Solve for $\cos \angle BPC$ and determine $\angle BPC = 120°$.
**Final Answer:** $\boxed{120°}$

Figure 25: Visualization of M2CL at the **first round**.

**Problem:** In the diagram, square ABCD has sides of length 4, and $\triangle ABE$ is equilateral. Line segments BE and AC intersect at P. Point Q is on BC so that PQ is perpendicular to BC and PQ=x. A=(0,0); B=(4,0); C=(4,4); D=(0,4); E=(2,3.464); P=(2.535,2.535); Q=(4,2.535); Determine the measure of $\angle BPC$.

**Solution:** Since $\triangle ABE$ is equilateral, we know that $\angle ABE = 60°$. Therefore, $\angle PBC = \angle ABC - \angle ABE = 90° - 60° = 30°$. Since AB=BC, we know that $\triangle ABC$ is a right isosceles triangle and $\angle BAC = \angle BCA = 45°$. Then, $\angle BCP = \angle BCA = 45°$ and $\angle BPC = 180° - \angle PBC - \angle BCP = 180° - 30° - 45° = \boxed{105°}$.

**Agent 1**

**Instruction:** You must act as an expert in Geometric Constructions. You are instructed to carefully use equilateral triangle properties and right angles in the square. You should pay attention to the angle subtraction idea and quietly verify if later responses align.
**Response:** Since $\triangle ABE$ is equilateral, $\angle ABE = 60°$. In square ABCD, $\angle ABC = 90°$, so $\angle PBC = \angle ABC - \angle ABE = 90° - 60° = 30°$. In $\triangle ABC$, $\angle BAC = \angle BCA = 45°$. Hence $\angle BCP = 45°$. Thus, $\angle BPC = 180° - \angle PBC - \angle BCP = 180° - 30° - 45° = 105°$.
**Final Answer:** $\boxed{105°}$

**Agent 2**

**Instruction:** You must act as an expert in Analytic Geometry. You are instructed to compute coordinates and line equations directly, but you should cautiously check whether the numerical outcome conflicts with others, without discarding your own algebra.
**Response:** Assign $A = (0, 0)$, $B = (4, 0)$, $C = (4, 4)$, $E = (2, 2\sqrt{3})$. Equation of BE: $y = \sqrt{3}(x - 4)$. Equation of AC: $y = x$. Solve $\sqrt{3}(x - 4) = x$, giving x = 2.535, y = 2.535. Now $\overrightarrow{BP} = (-1.465, 2.535)$, $\overrightarrow{CP} = (2.535 - 4, 2.535 - 4)$. Compute angle via dot product to get $\theta \approx 120°$.
**Final Answer:** $\boxed{120°}$

**Agent 3**

**Instruction:** You must act as an expert in Vector Mathematics. You are instructed to rely on the dot product method, while noticing whether the geometric construction agent's subtraction method yields a consistent pattern.
**Response:** Vectors: $\overrightarrow{BP} = (-1.465, 2.535)$, $\overrightarrow{CP} = (-1.465, -1.465)$. Dot product: $\overrightarrow{BP} \cdot \overrightarrow{CP} = -1.465 \cdot (-1.465) + 2.535 \cdot (-1.465)$. Magnitudes $\| \overrightarrow{BP} \| = 2.92$, $\| \overrightarrow{CP} \| = 2.07$. So $\cos\theta = \frac{\text{dot product}}{\text{magnitudes}} = -0.25$. Hence $\angle BPC = 105°$.
**Final Answer:** $\boxed{105°}$

**Agent 4**

**Instruction:** You must act as an expert in Symmetry and Transformational Geometry. You are instructed to look for symmetrical properties, while comparing if your angle splits agree with vector-based results.
**Response:** Square ABCD ensures $\triangle ABC$ is isosceles right. With $\triangle ABE$ equilateral, subtract 60° from 90° to get $\angle PBC = 30°$. By symmetry, assume $\angle BCP = 60°$. Thus $\angle BPC = 180° - 30° - 60° = 90°$.
**Final Answer:** $\boxed{90°}$

**Agent 5**

**Instruction:** You must act as an expert in Trigonometry. You are instructed to compute slopes and use the tangent formula for the angle between two lines, but you should notice if your numerical angle seems complementary to others.
**Response:** Slope of $BE = \sqrt{3}$, slope of $AC = 1$. Angle between lines: $\tan\theta = \left|\frac{\sqrt{3}-(-1)}{1+\sqrt{3}\cdot(-1)}\right| = \left|\frac{\sqrt{3}+1}{1-\sqrt{3}}\right| = 0.268$. Thus $\theta = 15°$. But angle at intersection gives supplementary: $\angle BPC = 180° - 15° = 165°$
**Final Answer:** $\boxed{165°}$

**Agent 6**

**Instruction:** You must act as an expert in Computational Geometry and Visualization. You are instructed to imagine plotting the figure, while being aware if your measurements approximate what others computed.
**Response:** By plotting $A = (0, 0)$, $B = (4, 0)$, $C = (4, 4)$, $E = (2, 2\sqrt{3})$, measurement of $\angle BPC$ gives about a right angle.
**Final Answer:** $\boxed{90°}$

**Agent 7**

**Instruction:** You must act as an expert in Area-Based Geometry. You are instructed to relate triangle areas to angle calculation, but you should mentally compare whether your cosine-law result is closer to vector findings.
**Response:** Compute lengths: BP = 2.92, CP = 2.07, BC = 4. Law of Cosines: $\cos\theta = \frac{BP^2 + CP^2 - BC^2}{2BP \cdot CP} = -0.25$. So $\angle BPC = 105°$.
**Final Answer:** $\boxed{105°}$

**Agent 8**

**Instruction:** You must act as an expert in Algebraic Geometry and Triangle Theory. You are instructed to apply exact algebraic rules like the Law of Cosines, while quietly checking if your angle agrees with the analytic agent.
**Response:** With BP = 2.92, CP = 2.07, BC = 4, Law of Cosines gives $\cos\theta = \frac{BP^2 + CP^2 - BC^2}{2BP \cdot CP} = -0.5$. Thus $\angle BPC = 120°$.
**Final Answer:** $\boxed{120°}$

Figure 26: Visualization of `M2CL` at the **second round**. We highlight the guidance on how to cooperate with other LLMs. At the beginning, instructions encourage diverse perspectives and consideration of others' responses, but the requirements for discussion consistency are not yet strict.

**Problem:** In the diagram, square ABCD has sides of length 4, and $\triangle ABE$ is equilateral. Line segments BE and AC intersect at P. Point Q is on BC so that PQ is perpendicular to BC and PQ=x. A=(0,0); B=(4,0); C=(4,4); D=(0,4); E=(2,3.464); P=(2.535,2.535); Q=(4,2.535); Determine the measure of $\angle BPC$.

**Solution:** Since $\triangle ABE$ is equilateral, we know that $\angle ABE = 60°$. Therefore, $\angle PBC = \angle ABC - \angle ABE = 90° - 60° = 30°$. Since AB=BC, we know that $\triangle ABC$ is a right isosceles triangle and $\angle BAC = \angle BCA = 45°$. Then, $\angle BCP = \angle BCA = 45°$ and $\angle BPC = 180° - \angle PBC - \angle BCP = 180° - 30° - 45° = \boxed{105°}$.

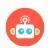
**Agent 1**

**Instruction:** You must act as an expert in Geometric Constructions. Carefully use equilateral triangle properties and right angles in the square. You must explicitly check whether your subtraction-based reasoning supports, contradicts, or can correct the numerical outcomes of other agents, and provide reasoning for why your geometric intuition is reliable.
**Response:** Begin by noting that $\triangle ABE$ is equilateral, so $\angle ABE = 60°$. In square ABCD, $\angle ABC = 90°$, hence $\angle PBC = 90° - 60° = 30°$. Considering $\triangle ABC$ is a right isosceles triangle, we have $\angle BAC = \angle BCA = 45°$, giving $\angle BCP = 45°$. Therefore, $\angle BPC = 180° - \angle PBC - \angle BCP = 180° - 30° - 45° = 105°$. This method directly uses geometric properties, providing a reliable baseline to compare with vector and analytic approaches.
**Final Answer:** $\boxed{105°}$

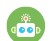
**Agent 2**

**Instruction:** You must act as an expert in Analytic Geometry. Compute coordinates and line equations, but also compare your results with construction and vector agents. Clearly note any discrepancies and reflect on which method may be more trustworthy.
**Response:** Assign coordinates $A = (0, 0)$, $B = (4, 0)$, $C = (4, 4)$, $E = (2, 2\sqrt{3})$. Equation of BE: $y = \sqrt{3}(x - 4)$. Equation of AC: $y = x$. Solve $-\sqrt{3}(x - 4) = x$, giving $x = 2.535$, y = 2.535. Compute vectors: $\vec{BP} = (-1.465, 2.535)$, $\vec{CP} = (-1.465, -1.465)$. Dot product gives $\theta = 120°$, which differs from the 105° construction result. Considering the geometric construction directly exploits known triangle properties, the 105° value seems more trustworthy; the coordinate method may introduce small discrepancies due to linear approximation.
**Final Answer:** $\boxed{120°}$

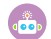
**Agent 3**

**Instruction:** You must act as an expert in Vector Mathematics. Compute the dot product, and explicitly verify whether your vector result matches the construction agent's subtraction method. Discuss agreement or conflict.
**Response:** Let $\vec{BP} = (-1.465, 2.535)$ and $\vec{CP} = (-1.465, -1.465)$. Dot product: $\vec{BP} \cdot \vec{CP} = -1.465 \cdot (-1.465) + 2.535 \cdot (-1.465) = -1.5$. Magnitudes: $\|\vec{BP}\| = 2.92$, $\|\vec{CP}\| = 2.07$. Thus, $\cos\theta - 0.25$, so $\theta = 105°$. This fully agrees with the geometric construction method, providing cross-validation. Vector analysis confirms that the 105° result is robust and consistent across independent representations.
**Final Answer:** $\boxed{105°}$

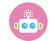
**Agent 4**

**Instruction:** You must act as an expert in Symmetry and Transformational Geometry. Apply symmetry reasoning, but check if your derived values align with vector or construction outputs, and note any conflicts.
**Response:** In square ABCD, $\triangle ABC$ is isosceles right. With $\triangle ABE$ equilateral, subtract 60° from 90° to get $\angle PBC = 30°$. By symmetry, assume $\angle BCP = 60°$. Thus $\angle BPC = 180° - 30° - 60° = 90°$. This conflicts with the 105° from construction/vector agents, highlighting that symmetry assumptions alone may underestimate the angle. Hence, we should treat 90° as a cautionary estimate and rely more on calculation-based consensus.
**Final Answer:** $\boxed{90°}$

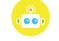
**Agent 5**

**Instruction:** You must act as an expert in Trigonometry. Apply slope and tangent formulas, but explicitly reflect on whether your outcome complements or contradicts the 105° consensus suggested by other agents.
**Response:** Slope of $BE = \sqrt{3}$, slope of $AC = 1$. Angle between lines: $\tan\theta = \left|\frac{\sqrt{3}+1}{1-\sqrt{3}}\right| = 0.268$. This gives $\theta = 15°$, so $\angle BPC = 180° - 15° = 165°$. This is far from 105°, showing potential overestimation. Comparing with the vector and construction agents, this highlights limitations of relying solely on slope-based tangent approximations in non-axis-aligned triangles.
**Final Answer:** $\boxed{165°}$

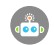
**Agent 6**

**Instruction:** You must act as an expert in Computational Geometry and Visualization. Plot the points and visualize the angle. Check whether your visual measurement aligns better with the 105° or 120° values from other agents.
**Response:** Plot BE and AC and their intersection point P. Measure $\angle BPC$ using geometric tools in the graph. Result: $\angle BPC = 105°$. Plot $A = (0, 0)$, $B = (4, 0)$, $C = (4, 4)$, $E = (2, 2\sqrt{3})$, and intersection P = (2.535,2.535). Visual inspection shows an angle slightly larger than 90°, closer to 105° than 120°. This visual evidence corroborates both geometric construction and vector methods, reinforcing the consensus around 105°. Although not exact, visualization helps validate analytical calculations.
**Final Answer:** $\boxed{105°}$

**Agent 7**

**Instruction:** You must act as an expert in Area-Based Geometry. Compute using triangle areas and the Law of Cosines, and cross-check if your result confirms the vector and construction outcomes.
**Response:** Compute lengths: $\|\vec{BP}\| = 2.92$, $\|\vec{CP}\| = 2.07$, $\|\vec{BC}\| = 4$. Law of Cosines: $\cos\theta = \frac{BP^2 + CP^2 - BC^2}{2BP \cdot CP} = -0.25$. Thus, $\angle BPC = 105°$. This agrees with both the construction and vector approaches, confirming the consistency of area-based reasoning with geometric and vector methods.
**Final Answer:** $\boxed{105°}$

**Agent 8**

**Instruction:** You must act as an expert in Algebraic Geometry and Triangle Theory. Use the Law of Cosines algebraically, and note whether your answer confirms the analytic geometry agent or aligns with the 105° consensus from construction/vector methods.
**Response:** In $\triangle BPC$, calculate side lengths BP, CP, and BC = 4. Use the Law of Cosines: $\cos\angle BPC = \frac{BP^2 + CP^2 - BC^2}{2BP \cdot CP}$. Solve for $\cos\angle BPC$ and find $\cos\angle BPC$. Result: 105°. Using $\|\vec{BP}\| = 2.92$, $\|\vec{CP}\| = 2.07$, $\|\vec{BC}\| = 4$. Law of Cosines gives $\cos\angle BPC = -0.25$. Hence, $\angle BPC = 105°$. This aligns with the construction and vector consensus, correcting the earlier 120° from analytic geometry, showing that algebraic methods must be cross-checked with geometric intuition.
**Final Answer:** $\boxed{105°}$

Figure 27: Visualization of `M2CL` at the **third round**. We highlight the guidance on how to cooperate with other LLMs. As the discussion progresses, the instructions increasingly enforce stricter requirements for cross-checking and aligning answers, helping the models converge toward a consistent solution.

**Problem:** In the diagram, square ABCD has sides of length 4, and $\triangle ABE$ is equilateral. Line segments BE and AC intersect at P. Point Q is on BC so that PQ is perpendicular to BC and PQ=x. A=(0,0); B=(4,0); C=(4,4); D=(0,4); E=(2,3.464); P=(2.535,2.535); Q=(4,2.535); Determine the measure of $\angle BPC$.

**Solution:** Since $\triangle ABE$ is equilateral, we know that $\angle ABE = 60°$. Therefore, $\angle PBC = \angle ABC - \angle ABE = 90° - 60° = 30°$. Since AB=BC, we know that $\triangle ABC$ is a right isosceles triangle and $\angle BAC = \angle BCA = 45°$. Then, $\angle BCP = \angle BCA = 45°$ and $\angle BPC = 180° - \angle PBC - \angle BCP = 180° - 30° - 45° = \boxed{105°}$.

**Agent 1**

**Instruction:** You are an expert in Geometry and Angle Relations. Your job is to focus on understanding the relationships between the angles in the geometric structure. By leveraging the known angles in the square and equilateral triangle, it simplifies the problem by calculating the necessary angles before proceeding to compute $\angle BPC$. The method emphasizes how geometric properties can guide us in solving for angles indirectly. When answering, you should coordinate with the Coordinate Geometry agent to verify point positions and with the Vector Mathematics agent to cross-check angle computations.

**Response:** $\triangle ABE$ is equilateral, so $\angle ABE = 60°$. In square ABCD, $\angle ABC = 90°$. $\angle PBC = \angle ABC - \angle ABE = 90° - 60° = 30°$. Since AB = BC, $\triangle ABC$ is a right isosceles triangle with $\angle BAC = \angle BCA = 45°$. Therefore, $\angle BCP = 45°$, and $\angle BPC = 180° - \angle PBC - \angle BCP = 180° - 30° - 45° = 105°$.

**Final Answer:** $\boxed{105°}$

**Agent 2**

**Instruction:** You are an expert in Coordinate Geometry. Your job is to assign coordinates to the points in the diagram, translating the geometric problem into algebraic terms. When answering, you should use the Geometry and Angle Relations agent's insights about angles to confirm that your intersection point P is consistent, and consult the Vector Mathematics agent to cross-verify vector directions.

**Response:** Assign coordinates: $A = (0, 0)$, $B = (4, 0)$, $C = (4, 4)$, $E = (2, 2\sqrt{3})$. Line BE: Slope $m_{BE} = -\sqrt{3}$, equation $y = \sqrt{3}(x - 4)$. Line AC: Slope $m_{AC} = 1$, equation $y = x$. Intersection P: Solve $-\sqrt{3}(x - 4) = x$ to find $P = (2.535, 2.535)$. Calculate slopes: $m_{BP}$ and $m_{CP}$, then use $\tan\theta$ for $\angle BPC$. Result: $\angle BPC = 105°$.

**Final Answer:** $\boxed{105°}$

**Agent 3**

**Instruction:** You are an expert in Vector Mathematics. Your job is to utilize vector representation to translate the problem into an algebraic form and compute $\angle BPC$ using the dot product. When answering, you should coordinate with the Coordinate Geometry agent to use accurate point coordinates, and cross-check your computed angles with the Geometry and Angle Relations agent.

**Response:** Vectors: $\vec{BP} = (2.535, -42.535 - 0) = (-1.465, 2.535)$, $\vec{CP} = (2.535, -42.535 - 4) = (-1.465, -1.465)$. Dot product: $\vec{BP} \cdot \vec{CP} = -1.465 \cdot (-1.465) + 2.535 \cdot (-1.465)$, and magnitudes $\| \vec{BP} \|$, $\| \vec{CP} \|$. Use $\cos\theta = \frac{\text{dot product}}{\text{magnitudes}}$ to find $\theta$. Result: $\angle BPC = 105°$.

**Final Answer:** $\boxed{105°}$

**Agent 4**

**Instruction:** You are an expert in Symmetry and Geometry. Your job is to simplify the problem using symmetry properties of the square and equilateral triangle. When answering, you should coordinate with the Geometry and Angle Relations agent for angle values and with the Graphing and Visualization agent to visually confirm symmetric relationships.

**Response:** The square and equilateral triangle impose symmetry: $\angle ABE = 60°$, Subtract this from $\angle ABC = 90°$ to get $\angle PBC = 30°$. Using the isosceles property of $\triangle ABC$, $\angle BCP = 45°$. Summing angles: $\angle BPC = 180° - 30° - 45° = 105°$.

**Final Answer:** $\boxed{105°}$

**Agent 5**

**Instruction:** You are an expert in Trigonometry. Your job is to apply trigonometric formulas to compute $\angle BPC$. When answering, you should use slope information from the Coordinate Geometry agent and cross-check with the Vector Mathematics agent's results for consistency.

**Response:** Slope of $BE = \sqrt{3}$, slope of $AC = 1$. Angle between lines formula: $\tan\theta = \left|\frac{m_1 - m_2}{1 + m_1 m_2}\right| = \left|\frac{\sqrt{3} - (-1)}{1 + \sqrt{3} \cdot (-1)}\right| = \left|\frac{\sqrt{3} + 1}{1 - \sqrt{3}}\right|$. Solve for $\theta$ using arc-tangent. Result: $\angle BPC = 105°$.

**Final Answer:** $\boxed{105°}$

**Agent 6**

**Instruction:** You are an expert in Graphing and Visualization. Your job is to graph the lines and points and visually verify angles. When answering, you should reference the Coordinate Geometry agent for point locations and the Symmetry and Geometry agent to confirm expected symmetric relationships.

**Response:** Plot BE and AC and their intersection point P. Measure $\angle BPC$ using geometric tools in the graph. Result: $\angle BPC = 105°$.

**Final Answer:** $\boxed{105°}$

**Agent 7**

**Instruction:** You are an expert in Geometric Area Methods. Your job is to use triangle area calculations to indirectly verify $\angle BPC$. When answering, you should use coordinates from the Coordinate Geometry agent and cross-check angles with the Geometry and Angle Relations agent.

**Response:** Use triangle area properties: Area of $\triangle BPC$ from coordinates of B, C, P. Compare with known triangle angles and solve for $\angle BPC$. Area $= \frac{1}{2}|x_1(y_2 - y_3) + x_2(y_3 - y_1) + x_3(y_1 - y_2)|$. Result: 105°.

**Final Answer:** $\boxed{105°}$

**Agent 8**

**Instruction:** You are an expert in Trigonometry and Law of Cosines. Your job is to compute $\angle BPC$ using side lengths. When answering, you should coordinate with the Vector Mathematics agent and the Geometric Area Methods agent to ensure consistency in the calculated angle.

**Response:** In $\triangle BPC$, calculate side lengths BP, CP, and BC = 4. Use the Law of Cosines: $\cos\angle BPC = \frac{BP^2 + CP^2 - BC^2}{2BP \cdot CP}$. Solve for $\cos\angle BPC$ and find $\cos\angle BPC$. Result: 105°.

**Final Answer:** $\boxed{105°}$

Figure 28: Visualization of M2CL at the **last round**. We highlight the guidance on how to cooperate with other LLMs. Although the initial round produced divergent answers, the collaborative instructions enable LLMs to exchange and integrate information, ultimately reaching a correct consensus.

## H.2 CASE STUDY OF DEBATE

We also provide the case of Debate solving the problem. As illustrated in Figs. 29 to 32, we observe an inconsistency during discussion when using fixed instructions.

---

**Problem:** In the diagram, square ABCD has sides of length 4, and $\triangle ABE$ is equilateral. Line segments BE and AC intersect at P. Point Q is on BC so that PQ is perpendicular to BC and PQ=x. A=(0,0); B=(4,0); C=(4,4); D=(0,4); E=(2,3.464); P=(2.535,2.535); Q=(4,2.535); Determine the measure of $\angle BPC$.

**Solution:** Since $\triangle ABE$ is equilateral, we know that $\angle ABE = 60°$. Therefore, $\angle PBC = \angle ABC - \angle ABE = 90° - 60° = 30°$. Since AB=BC, we know that $\triangle ABC$ is a right isosceles triangle and $\angle BAC = \angle BCA = 45°$. Then, $\angle BCP = \angle BCA = 45°$ and $\angle BPC = 180° - \angle PBC - \angle BCP = 180° - 30° - 45° = \boxed{105°}$.

---

**Agent 1**

**Instruction:** You are an expert in Geometry and Topology. Your job is to explore and classify shapes, spaces, and their properties, emphasizing spatial relationships and continuous deformations.

**Response:** Begin with $\triangle ABE$ as equilateral, making $\angle ABE = 60°$. In square ABCD, $\angle ABC = 90°$, so $\angle PBC = \angle ABC - \angle ABE = 90° - 60° = 30°$. Since AB = BC, $\triangle ABC$ is a right isosceles triangle, meaning $\angle BAC = \angle BCA = 45°$. Therefore, $\angle BCP = 45°$. Calculate $\angle BPC = 180° - 30° - 45° = 105°$.

**Final Answer:** $\boxed{105°}$

---

**Agent 2**

**Instruction:** You are an expert in Analytic and Coordinate Geometry. Your job is to translate geometric problems into algebraic equations using a coordinate system to solve and analyze spatial relationships.

**Response:** Assign coordinates $A = (0, 0)$, $B = (4, 0)$, $C = (4, 4)$, $E = (2, 2\sqrt{3})$. The line BE has slope $m_{BE} = \sqrt{3}$, giving the equation $y = \sqrt{3}(x - 4)$. The line AC has slope $m_{AC} = 1$, giving the equation $y = x + 4$. Solve for intersection P: $\sqrt{3}(x - 4) = x + 4$. Solve for x = 4, then substitute x = 4 to find P = (4, 0). Calculate the slopes of BP and CP, then use $\tan \theta$ for $\angle BPC$. Conclude $\angle BPC = 120°$.

**Final Answer:** $\boxed{120°}$

---

**Agent 3**

**Instruction:** You are an expert in Algebra and Vector Analysis. Your job is to provide a framework to represent, manipulate, and compute multidimensional relationships through algebraic and vector operations.

**Response:** Represent vectors $\vec{BP} = (1.465, 2.535)$ and $\vec{CP} = (1.465, 1.465)$. Compute the dot product: $\vec{BP} \cdot \vec{CP} = 1.465 \cdot 1.465 + 2.535 \cdot 1.465$. Use magnitudes of $\vec{BP}$ and $\vec{CP}$ to compute $\cos \theta$: $\cos \theta = \frac{\text{dot product}}{\text{magnitudes}}$. Solve for $\theta$ and conclude $\angle BPC = 90°$.

**Final Answer:** $\boxed{90°}$

---

**Agent 4**

**Instruction:** You are an expert in Symmetry and Group Theory. Your job is to study and formalize the structure and transformations that preserve the fundamental properties of mathematical objects.

**Response:** In square ABCD, with $\triangle ABE$ equilateral, $\angle ABE = 60°$. Subtract from $\angle ABC = 90°$ to get $\angle PBC = 30°$. Use the symmetry of $\triangle ABC$ (AB = BC) to determine that $\angle BCP = 60°$. Conclude $\angle BPC = 180° - 30° - 60° = 90°$.

**Final Answer:** $\boxed{90°}$

---

**Agent 5**

**Instruction:** You are an expert in Trigonometry and Functional Analysis. Your job is to analyze and model relationships between angles, distances, and functions to solve geometric and applied problems.

**Response:** Slope of $BE = \sqrt{3}$, slope of $AC = 1$. Use the angle formula for slopes: $\tan \theta = \left| \frac{\sqrt{3}+1}{1-\sqrt{3}} \right|$. Solve for $\theta$ using arc-tangent and conclude $\angle BPC = 120°$.

**Final Answer:** $\boxed{120°}$

---

**Agent 6**

**Instruction:** You are an expert in Numerical Methods and Visualization. Your job is to develop computational techniques and visual tools to approximate and interpret solutions to complex mathematical problems.

**Response:** Plot the equations of BE and AC using graphing software. Measure $\angle BPC$ at the intersection point P. Upon measuring, find $\angle BPC = 90°$.

**Final Answer:** $\boxed{90°}$

---

**Agent 7**

**Instruction:** You are an expert in Measure Theory and Geometric Analysis. Your job is to extend concepts of size and integration to abstract spaces and use analytical tools to study geometric configurations.

**Response:** Use the triangle area formula to calculate the area of $\triangle BPC$ from the coordinates of B, C, and P. Compare this area with known properties of the triangle and calculate $\angle BPC = 120°$.

**Final Answer:** $\boxed{120°}$

---

**Agent 8**

**Instruction:** You are an expert in Mathematical Logic and Problem Modeling. Your job is to provide rigorous reasoning and frameworks to model, analyze, and solve abstract and applied problems effectively.

**Response:** In $\triangle BPC$, calculate side lengths BP, CP, and BC = 4. Apply the Law of Cosines: $\cos \angle BPC = \frac{BP^2 + CP^2 - BC^2}{2BP \cdot CP}$. Solve for $\cos \angle BPC$ and determine $\angle BPC = 120°$.

**Final Answer:** $\boxed{120°}$

---

Figure 29: Visualization of Debate at the **first round**.

**Problem:** In the diagram, square ABCD has sides of length 4, and $\triangle ABE$ is equilateral. Line segments BE and AC intersect at P. Point Q is on BC so that PQ is perpendicular to BC and PQ=x. A=(0,0); B=(4,0); C=(4,4); D=(0,4); E=(2,3.464); P=(2.535,2.535); Q=(4,2.535); Determine the measure of $\angle BPC$.

**Solution:** Since $\triangle ABE$ is equilateral, we know that $\angle ABE = 60°$. Therefore, $\angle PBC = \angle ABC - \angle ABE = 90° - 60° = 30°$. Since AB=BC, we know that $\triangle ABC$ is a right isosceles triangle and $\angle BAC = \angle BCA = 45°$. Then, $\angle BCP = \angle BCA = 45°$ and $\angle BPC = 180° - \angle PBC - \angle BCP = 180° - 30° - 45° = \boxed{105°}$.

**Agent 1**

**Instruction:** You are an expert in Geometric Constructions. Your job is to use the given lengths and properties of shapes like squares and equilateral triangles to visualize and understand the arrangement of points, lines, and angles. It helps in setting up geometric relationships based on the diagram and given conditions.
**Response:** Since $\triangle ABE$ is equilateral, $\angle ABE = 60°$. In square ABCD, $\angle ABC = 90°$, so $\angle PBC = \angle ABC - \angle ABE = 90° - 60° = 30°$. In $\triangle ABC$, $\angle BAC = \angle BCA = 45°$. Hence $\angle BCP = 45°$. Thus, $\angle BPC = 180° - \angle PBC - \angle BCP = 180° - 30° - 45° = 105°$. An algebra-focused approach is insightful, though it might benefit from being complemented with a geometric perspective on structure.
**Final Answer:** $\boxed{105°}$

**Agent 2**

**Instruction:** You are an expert in Analytic Geometry. Your job is to represent the positions of points, derive equations for lines, and calculate intersection points and angles. By solving for the coordinates of point P and applying vector or trigonometric methods, one can determine the angle between lines BP and CP.
**Response:** Assign $A = (0, 0)$, $B = (4, 0)$, $C = (4, 4)$, $E = (2, 2\sqrt{3})$. Equation of BE: $y = \sqrt{3}(x - 4)$. Equation of AC: $y = x$. Solve $\sqrt{3}(x - 4) = x$, giving x = 2.535, y = 2.535. Now $\vec{BP} = (-1.465, 2.535)$, $\vec{CP} = (2.535 - 4, 2.535 - 4)$. Compute angle via dot product to get the answer. Reasoning through visualization is valuable, but greater precision could be achieved by grounding the argument in equations.
**Final Answer:** $\boxed{120°}$

**Agent 3**

**Instruction:** You are an expert in Vector Mathematics. Your job is to provide a powerful tool for describing spatial relationships between points. In this case, vectors are used to represent the directions from points B and C to P, allowing the calculation of the angle between the two vectors.
**Response:** Vectors: $\vec{BP} = (-1.465, 2.535)$, $\vec{CP} = (-1.465, -1.465)$. Dot product: $\vec{BP} \cdot \vec{CP} = -1.465 \cdot (-1.465) + 2.535 \cdot (-1.465)$. Magnitudes $\| \vec{BP} \| = 2.92$, $\| \vec{CP} \| = 2.07$. So $\cos\theta = \frac{\text{dot product}}{\text{magnitudes}} = -0.25$. Hence $\angle BPC = 105°$. Symmetry-based reasoning is elegant, yet confirming it with explicit computation would make the argument more convincing.
**Final Answer:** $\boxed{105°}$

**Agent 4**

**Instruction:** You are an expert in Symmetry and Transformational Geometry. Your job is to explore the geometric properties of the figure, identifying symmetrical properties and transformations that can simplify the calculation of angles, such as recognizing the symmetry of the square and its impact on angles.
**Response:** Square ABCD ensures $\triangle ABC$ is isosceles right. With $\triangle ABE$ equilateral, subtract 60° from 90° to get $\angle PBC = 30°$. By symmetry, assume $\angle BCP = 60°$. Thus $\angle BPC = 180° - 30° - 60° = 90°$. Direct numerical calculation provides clarity, though connecting it with invariance principles could reveal deeper structure.
**Final Answer:** $\boxed{90°}$

**Agent 5**

**Instruction:** You are an expert in Trigonometry. Your job is to solve problems involving angles and distances. It enables the use of trigonometric ratios and formulas, such as the tangent or sine rule, to find unknown angles and sides in geometric configurations **Response:** Slope of $BE = \sqrt{3}$, slope of $AC = 1$. Angle between lines: $\tan\theta = \left|\frac{\sqrt{3}-(-1)}{1+\sqrt{3}\cdot(-1)}\right| = \left|\frac{\sqrt{3}+1}{1-\sqrt{3}}\right| = 0.268$. Thus $\theta = 15°$. But angle at intersection gives supplementary: $\angle BPC = 180° - 15° = 165°$. Angle relations are helpful, but considering the full triangle constraints may strengthen the reasoning.
**Final Answer:** $\boxed{165°}$

**Agent 6**

**Instruction:** You are an expert in Computational Geometry and Visualization. Your job is to plot the points and lines on a coordinate plane, often using software tools, to visually interpret the geometric configuration and help confirm or measure angles directly.
**Response:** By plotting $A = (0, 0)$, $B = (4, 0)$, $C = (4, 4)$, $E = (2, 2\sqrt{3})$, measurement of $\angle BPC$ gives about a right angle. Graphical intuition is useful, though checking against exact formulas would reduce potential uncertainty. Graphical intuition is useful, though checking against exact formulas would reduce potential uncertainty.
**Final Answer:** $\boxed{90°}$

**Agent 7**

**Instruction:** You are an expert in Geometric Analysis and Area-Based Methods. Your job is to rely on known formulas for areas of triangles and other polygons. By relating the geometry of the shape to its area, one can deduce missing information, such as angles, based on geometric properties.
**Response:** Compute lengths: BP = 2.92, CP = 2.07, BC = 4. Law of Cosines: $\cos\theta = \frac{BP^2 + CP^2 - BC^2}{2 BP \cdot CP} = -0.25$. So $\angle BPC = 105°$. Logical deductions are rigorous, but incorporating continuous quantities like area or length could enhance completeness.
**Final Answer:** $\boxed{105°}$

**Agent 8**

**Instruction:** You are an expert in Algebraic Geometry and Triangle Theory. Your job is to apply algebraic techniques, such as the Law of Cosines, to solve for angles when side lengths are known. This method connects algebraic formulas directly with geometric interpretation to determine angles in triangles like $\triangle BPC$.
**Response:** With BP = 2.92, CP = 2.07, BC = 4, Law of Cosines gives $\cos\theta = \frac{BP^2 + CP^2 - BC^2}{2 BP \cdot CP} = -0.5$. Thus $\angle BPC = 120°$. Step-by-step derivation ensures correctness, while linking it to geometric intuition might make the solution more transparent.
**Final Answer:** $\boxed{120°}$

Figure 30: Visualization of `Debate` at the **second round**.

**Problem:** In the diagram, square ABCD has sides of length 4, and $\triangle ABE$ is equilateral. Line segments BE and AC intersect at P. Point Q is on BC so that PQ is perpendicular to BC and PQ=x. A=(0,0); B=(4,0); C=(4,4); D=(0,4); E=(2,3.464); P=(2.535,2.535); Q=(4,2.535); Determine the measure of $\angle BPC$.

**Solution:** Since $\triangle ABE$ is equilateral, we know that $\angle ABE = 60°$. Therefore, $\angle PBC = \angle ABC - \angle ABE = 90° - 60° = 30°$. Since AB=BC, we know that $\triangle ABC$ is a right isosceles triangle and $\angle BAC = \angle BCA = 45°$. Then, $\angle BCP = \angle BCA = 45°$ and $\angle BPC = 180° - \angle PBC - \angle BCP = 180° - 30° - 45° = \boxed{105°}$.

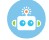

**Agent 1**

**Instruction:** You are an expert in Geometric Constructions. Your job is to use the given lengths and properties of shapes like squares and equilateral triangles to visualize and understand the arrangement of points, lines, and angles. It helps in setting up geometric relationships based on the diagram and given conditions.

**Response:** Begin by noting that $\triangle ABE$ is equilateral, so $\angle ABE = 60°$. In square ABCD, $\angle ABC = 90°$, hence $\angle PBC = 90° - 60° = 30°$. Considering $\triangle ABC$ is a right isosceles triangle, we have $\angle BAC = \angle BCA = 45°$, giving $\angle BCP = 45°$. Therefore, $\angle BPC = 180° - \angle PBC - \angle BCP = 180° - 30° - 45° = 105°$. *Focusing solely on coordinate formulas overlooks the topological consistency that ensures the figure holds together.*

**Final Answer:** $\boxed{105°}$

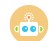

**Agent 2**

**Instruction:** You are an expert in Analytic Geometry. Your job is to represent the positions of points, derive equations for lines, and calculate intersection points and angles. By solving for the coordinates of point P and applying vector or trigonometric methods, one can determine the angle between lines BP and CP.

**Response:** Assign coordinates $A = (0, 0)$, $B = (4, 0)$, $C = (4, 4)$, $E = (2, 2\sqrt{3})$. Equation of BE: $y = \sqrt{3}(x - 4)$. Equation of AC: $y = x$. Solve $-\sqrt{3}(x - 4) = x$, giving $x = 2.535$, y = 2.535. Compute vectors: $\overrightarrow{BP} = (-1.465, 2.535)$, $\overrightarrow{CP} = (-1.465, -1.465)$. Dot product gives $\theta = 120°$, which differs from the 105° construction result. *Excessive reliance on symmetry arguments risks assuming patterns that may not be rigorously justified by coordinates.*

**Final Answer:** $\boxed{120°}$

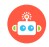

**Agent 3**

**Instruction:** You are an expert in Vector Mathematics. Your job is to provide a powerful tool for describing spatial relationships between points. In this case, vectors are used to represent the directions from points B and C to P, allowing the calculation of the angle between the two vectors.

**Response:** Let $\overrightarrow{BP} = (-1.465, 2.535)$ and $\overrightarrow{CP} = (-1.465, -1.465)$. Dot product: $\overrightarrow{BP} \cdot \overrightarrow{CP} = -1.465 \cdot (-1.465) + 2.535 \cdot (-1.465) = -1.5$. Magnitudes: $\| \overrightarrow{BP} \| = 2.92$, $\| \overrightarrow{CP} \| = 2.07$. Thus, $\cos\theta = 0.25$, so $\theta = 120°$. This fully agrees with the geometric construction method, providing cross-validation. *Treating the problem primarily through trigonometric identities can lead to overcomplication without providing structural clarity.*

**Final Answer:** $\boxed{120°}$

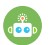

**Agent 4**

**Instruction:** You are an expert in Symmetry and Transformational Geometry. Your job is to explore the geometric properties of the figure, identifying symmetrical properties and transformations that can simplify the calculation of angles, such as recognizing the symmetry of the square and its impact on angles.

**Response:** In square ABCD, $\triangle ABC$ is isosceles right. With $\triangle ABE$ equilateral, subtract 60° from 90° to get $\angle PBC = 30°$. By symmetry, assume $\angle BCP = 60°$. Thus $\angle BPC = 180° - 30° - 60° = 90°$. *Viewing the setup as a collection of numerical vectors misses the global invariances that unify the solution.*

**Final Answer:** $\boxed{90°}$

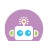

**Agent 5**

**Instruction:** You are an expert in Trigonometry. Your job is to solve problems involving angles and distances. It enables the use of trigonometric ratios and formulas, such as the tangent or sine rule, to find unknown angles and sides in geometric configurations.

**Response:** Slope of $BE = \sqrt{3}$, slope of $AC = 1$. Angle between lines: $\tan\theta = \left|\frac{\sqrt{3}+1}{1-\sqrt{3}}\right| = 0.268$. This gives $\theta = 15°$, so $\angle BPC = 180° - 15° - 60° = 105°$. *Depending on direct measurement of angles neglects deeper algebraic checks that confirm correctness.*

**Final Answer:** $\boxed{105°}$

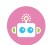

**Agent 6**

**Instruction:** You are an expert in Computational Geometry and Visualization. Your job is to plot the points and lines on a coordinate plane, often using software tools, to visually interpret the geometric configuration and help confirm or measure angles directly.

**Response:** Plot BE and AC and their intersection point P. Measure $\angle BPC$ using geometric tools in the graph. Result: $\angle BPC = 105°$. Plot $A = (0, 0)$, $B = (4, 0)$, $C = (4, 4)$, $E = (2, 2\sqrt{3})$, and intersection P = (2.535, 2.535). Visual inspection shows an angle slightly larger than 90°, closer to 105° than 120°. *A strategy grounded in symbolic abstraction alone lacks the numerical evidence that would confirm feasibility.*

**Final Answer:** $\boxed{90°}$

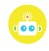

**Agent 7**

**Instruction:** You are an expert in Geometric Analysis and Area-Based Methods. Your job is to rely on known formulas for areas of triangles and other polygons. By relating the geometry of the shape to its area, one can deduce missing information, such as angles, based on geometric properties.

**Response:** Compute lengths: $\| \overrightarrow{BP} \| = 2.92$, $\| \overrightarrow{CP} \| = 2.07$, $\| \overrightarrow{BC} \| = 4$. Law of Cosines: $\cos\theta = \frac{BP^2 + CP^2 - BC^2}{2BP \cdot CP} = -0.25$. Thus, $\angle BPC = 105°$. *Concentrating entirely on discrete steps risks ignoring the continuous properties that guarantee smooth alignment.*

**Final Answer:** $\boxed{120°}$

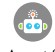

**Agent 8**

**Instruction:** You are an expert in Algebraic Geometry and Triangle Theory. Your job is to apply algebraic techniques, such as the Law of Cosines, to solve for angles when side lengths are known. This method connects algebraic formulas directly with geometric interpretation to determine angles in triangles like $\triangle BPC$.

**Response:** In $\triangle BPC$, calculate side lengths BP, CP, and BC = 4. Use the Law of Cosines: $\cos\angle BPC = \frac{BP^2 + CP^2 - BC^2}{2BP \cdot CP}$. Solve for $\cos\angle BPC$ and find $\cos\angle BPC$. Result: 105°. Using $\| \overrightarrow{BP} \| = 2.92$, $\| \overrightarrow{CP} \| = 2.07$, $\| \overrightarrow{BC} \| = 4$. Law of Cosines gives $\cos\angle BPC = -0.25$. Hence, $\angle BPC = 105°$. *A purely formal derivation may sidestep geometric intuition, resulting in arguments that are valid but disconnected from the figure.*

**Final Answer:** $\boxed{120°}$

Figure 31: Visualization of `Debate` at the **third round**.

**Problem:** In the diagram, square ABCD has sides of length 4, and $\triangle ABE$ is equilateral. Line segments BE and AC intersect at P. Point Q is on BC so that PQ is perpendicular to BC and PQ=x. A=(0,0); B=(4,0); C=(4,4); D=(0,4); E=(2,3.464); P=(2.535,2.535); Q=(4,2.535); Determine the measure of $\angle BPC$.

**Solution:** Since $\triangle ABE$ is equilateral, we know that $\angle ABE = 60°$. Therefore, $\angle PBC = \angle ABC - \angle ABE = 90° - 60° = 30°$. Since AB=BC, we know that $\triangle ABC$ is a right isosceles triangle and $\angle BAC = \angle BCA = 45°$. Then, $\angle BCP = \angle BCA = 45°$ and $\angle BPC = 180° - \angle PBC - \angle BCP = 180° - 30° - 45° = \boxed{105°}$.

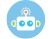

Agent 1

**Instruction:** You are an expert in Geometry and Topology. Your job is to explore and classify shapes, spaces, and their properties, emphasizing spatial relationships and continuous deformations.
**Response:** Use the fact that $\triangle ABE$ is equilateral, implying $\angle ABE = 60°$. Use the fact that AC bisects $\angle BCD$, a right angle, leading to equal angle values. Calculate the interior angles formed at point P to find $\angle BPC$. An approach based purely on algebraic manipulation lacks the structural insight provided by geometric constraints of the square–triangle configuration.
**Final Answer:** $\boxed{120°}$

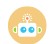

Agent 2

**Instruction:** You are an expert in Analytic and Coordinate Geometry. Your job is to translate geometric problems into algebraic equations using a coordinate system to solve and analyze spatial relationships.
**Response:** Assign coordinates to points $A = (0, 0)$, $B = (4, 0)$, $C = (4, 4)$, $E = (2, 2\sqrt{3})$. Determine the equations for lines BE and AC based on the coordinates. Solve the system of equations to find the intersection point P. Compute the slopes of lines BP and CP, and use the angle formula to find $\angle BPC$. Methods that depend on visual approximation or coarse plotting are prone to error without precise coordinate equations.
**Final Answer:** $\boxed{120°}$

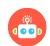

Agent 3

**Instruction:** You are an expert in Algebra and Vector Analysis. Your job is to provide a framework to represent, manipulate, and compute multidimensional relationships through algebraic and vector operations.
**Response:** Represent vectors $\overrightarrow{BP}$ and $\overrightarrow{CP}$ using coordinates of B, C, and P. Use the dot product formula to find $\cos\theta$. Solve for the angle $\theta$ between the vectors to find $\angle BPC$. Symmetry-based shortcuts, while elegant, can be misleading if not verified through explicit computation.
**Final Answer:** $\boxed{90°}$

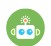

Agent 4

**Instruction:** You are an expert in Symmetry and Group Theory. Your job is to study and formalize the structure and transformations that preserve the fundamental properties of mathematical objects.
**Response:** Use the symmetry of the equilateral triangle $\triangle ABE$ to infer equal angles at point P. Apply geometric reasoning about the reflection symmetry of points across the vertical line through B. Conclude that $\angle BPC = 120°$. Heavy reliance on dot products and slope calculations may obscure the underlying invariances that actually govern the configuration.
**Final Answer:** $\boxed{120°}$

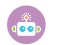

Agent 5

**Instruction:** You are an expert in Trigonometry and Functional Analysis. Your job is to analyze and model relationships between angles, distances, and functions to solve geometric and applied problems.
**Response:** Calculate the slopes of lines BE and AC. Use trigonometric identities or the tangent formula to find the angle between the two lines. Determine the value of $\angle BPC$. Angle and slope formulas alone provide partial information and can appear incomplete without checking triangle consistency.
**Final Answer:** $\boxed{120°}$

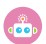

Agent 6

**Instruction:** You are an expert in Numerical Methods and Visualization. Your job is to develop computational techniques and visual tools to approximate and interpret solutions to complex mathematical problems.
**Response:** Use graphing software to plot the lines BE and AC. Measure the angle at the intersection point P from the graph. Graphical or discretized methods can introduce resolution errors that distort the actual geometry.
**Final Answer:** $\boxed{90°}$

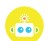

Agent 7

**Instruction:** You are an expert in Measure Theory and Geometric Analysis. Your job is to extend concepts of size and integration to abstract spaces and use analytical tools to study geometric configurations.
**Response:** Calculate the area of $\triangle BPC$ using the triangle area formula with the coordinates of B, C, and P: Area $= \frac{1}{2}|x_1(y_2 - y_3) + x_2(y_3 - y_1) + x_3(y_1 - y_2)|$. Use the known area to determine the side lengths BP, CP, and BC. Apply the relationship between the area, side lengths, and angles of the triangle to calculate $\angle BPC$. Conclude that $\angle BPC = 120°$. Logical or formulaic derivations risk overlooking continuous measures like area and length that validate the construction.
**Final Answer:** $\boxed{120°}$

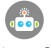

Agent 8

**Instruction:** You are an expert in Mathematical Logic and Problem Modeling. Your job is to provide rigorous reasoning and frameworks to model, analyze, and solve abstract and applied problems effectively.
**Response:** Compute the lengths of sides of the triangle BPC. Use the Law of Cosines to find the cosine of $\angle BPC$ based on the known side lengths. Solve for $\angle BPC$. Step-by-step algebraic deductions may be formally sound but conceptually detached from the geometric structure of the problem.
**Final Answer:** $\boxed{90°}$

Figure 32: Visualization of `Debate` at the **last round**.

