# OpenReview forum: "Context Learning for Multi-Agent Discussion"
_ICLR.cc/2026/Conference — ICLR 2026 Poster_

### Official Review · Reviewer_g8ch · 2025-10-30

**Soundness:** 3
**Presentation:** 3
**Contribution:** 3
**Rating:** 6
**Confidence:** 4

**Summary:**

This paper addresses the issue of discussion inconsistency in Multi-Agent Discussion (MAD) systems, where multiple Large Language Models (LLMs) collaborate through structured dialogue but often fail to reach a coherent consensus due to context misalignment among agents. To tackle this, this paper proposes M2CL (Multi-LLM Context Learning) that learns a context generator for each agent, enabling dynamic, round-by-round generation of optimized context instructions through automatic information organization and refinement. By introducing a self-adaptive mechanism that balances context coherence and output diversity, M2CL prevents premature convergence to “majority noise” and guides agents toward the correct consensus.

**Strengths:**

1.The paper clearly diagnoses discussion inconsistency，a core limitation of existing Multi-Agent Discussion systems to reach an agreement on a coherent solution.

2.The paper provides analytical understanding of how initial and evolving contexts influence discussion dynamics, grounding the method in a clear theoretical framework rather than purely empirical design.

3.Based on theoretical insight, the paper proposes M2CL framework learn dynamic, per-agent context generators that evolve during discussion.

4.The paper used dual gradient descent to automatically balance coherence and diversity, which is crucial for avoiding premature consensus and enabling more robust reasoning convergence.

5.Extensive experiments are conducted across three domains and nine datasets, covering LLM reasoning, embodied agentic, and mobile GUI AndroidWorld. The results demonstrate substantial and consistent performance improvements over previous MAD methods in terms of the number of collaborating LLMs, model size, and other aspects.

**Weaknesses:**

1.Lines 45–47: I have some concerns about whether the conclusion “fail to reach an agreement on a coherent solution, easily making the collaborative decision dominated by noise rather than principled reasoning” can be fully supported by Fig. 2. On the one hand, whether a higher level of discrepancy might actually be beneficial in some multi-agent discussion scenarios. Diverse responses could help explore different reasoning paths. Would it be helpful to check whether successful cases show lower final discrepancy than failed ones, to better support the claim that reducing discrepancy improves consensus?

2.Line 85: The paper states “A straightforward solution is to manually adapt instructions in contexts as the discussion progresses.” However, an arguably more intuitive approach would be to assign each LLM an additional discussion-summary LLM responsible for dynamically aggregating intermediate discussion results and generating updated context instructions. I am curious whether any prior work has adopted such a setup, and if so, what specific limitations those approaches faced compared with the proposed M2CL framework.

3.The derivation of Eq. (19) may contain a potential issue: the missing first term in the inequality could influence whether the inequality still holds. It would be helpful for the authors to clarify whether omitting this term affects the mathematical validity of the inequality, and if not, to explain why it can be safely ignored. Otherwise, this omission may undermine the soundness of subsequent theoretical conclusions that rely on Eq. (19).
a−ac=i=1Nωi∗aCib−ac≤i=1Nωi∗aCib−i=1Nωi∗aCit+i=1Nωi∗aCit−ac

4.While the paper presents impressive empirical results and a novel mechanism for context initialization in multi-LLM coordination, the theoretical reasoning connecting Eq. (9)–Eq. (10) seems to rely on several implicit assumptions. Eq. (9) ensures that f(a([A;P]))≈vp, and Eq. (10) ensures that F([Iib;P])≈f(a([Iib;P])). However, there is no theoretical guarantee that f(a([A;P]))≈i∈SωiF([Iib;P]) or f(a([A;P]))≈i∈Sωif(a([Iib;P])).
Therefore, it remains unclear whether the proposed procedure can truly achieve the “orthogonal basis selection” objective. I have some reservations about this logical gap, even though the empirical evidence is strong.

**Questions:**

Refer to the weaknesses.

---

> ### Author Response · Authors · 2025-11-21
>
> **Q1. Lines 45–47: I have some concerns about whether the conclusion “fail to reach an agreement on a coherent solution, easily making the collaborative decision dominated by noise rather than principled reasoning” can be fully supported by Fig.2. On the one hand, whether a higher level of discrepancy might actually be beneficial in some multi-agent discussion scenarios. Diverse responses could help explore different reasoning paths. Would it be helpful to check whether successful cases show lower final discrepancy than failed ones, to better support the claim that reducing discrepancy improves consensus?**
>
> Following the reviewer's suggestion, we have compared the discrepancy of the last round between successful cases and failed ones. The experimental setup is the same as in Fig.2 and the results are presented in the following table.
>
> |  Method  | MMLU | MATH  | GPQA  | Code  |
> |  ----  | ----  |  ----  | ----  | ----  |
> | Success  | 3.4 | 4.2 | 3.7 | 2.6 |
> | Failed  | 22.7 | 10.2 |11.2 | 18.6 |
>
> The results demonstrate that the discrepancy of success cases is significantly lower than the discrepancy of failed ones. In fact, a higher level of discrepancy is beneficial at the beginning of the discussion, as this expanding search space of solutions is the advantage of multi-agent discussion. As the discussion proceeds, agents may suffer from this inconsistency and fail to cooperate with each other (as shown in Fig.2). In our method, the adjustment of the weight parameter $\alpha$ is designed so that the context generators initially provide diverse contexts for exploration and guide agents toward a consensus as the discussion progresses.
>
> **Q2. Line 85: The paper states “A straightforward solution is to manually adapt instructions in contexts as the discussion progresses.” However, an arguably more intuitive approach would be to assign each LLM an additional discussion-summary LLM responsible for dynamically aggregating intermediate discussion results and generating updated context instructions. I am curious whether any prior work has adopted such a setup, and if so, what specific limitations those approaches faced compared with the proposed M2CL framework.**
>
> While directly using an LLM for summarization may seem intuitive, this approach faces two key limitations. First, the quality of the summary becomes a critical bottleneck: since the summary agent cannot directly solve problems, it may amplify errors or “majority noise” from the discussion rather than guide agents toward a correct consensus. As demonstrated in the case study in Section H, when most LLMs initially provide incorrect answers, a summary LLM lacks a reliable ground truth and risks reinforcing the prevailing “majority noise,” making it extremely difficult to generate an accurate and useful summary. Second, deploying a full-scale summary LLM for each agent incurs substantial parameter and computational costs, which severely limit scalability.
>
> In contrast, M2CL learns a compact context generator that produces targeted instructions instead of functioning as a black-box summarizer. These instructions are dynamically tailored to each agent’s initial context and the evolving discussion state, providing concrete, actionable guidance for refining reasoning. For example: “You are instructed to compute coordinates and line equations directly, but carefully check whether your numerical results conflict with others, without discarding your own algebra.” This approach directs agents to leverage their expertise while explicitly coordinating with others.
>
> In the experiment, as shown in Fig.5-11, M2CL outperforms GPTSwarm, which utilizes a summary LLM, demonstrating its effectiveness in generating guidance on reaching consensus to enhance the capability of multi-agent discussion.
>
> **Q3. The derivation of Eq.(19) may contain a potential issue: the missing first term in the inequality could influence whether the inequality still holds. It would be helpful for the authors to clarify whether omitting this term affects the mathematical validity of the inequality, and if not, to explain why it can be safely ignored. Otherwise, this omission may undermine the soundness of subsequent theoretical conclusions that rely on Eq.(19).**
>
> We would like to clarify that in the derivation of Eq.(19), no term was omitted and the inequality remains mathematically valid. Specifically, for the derivation of Eq.(19), we add and subtract $\sum_{i=1}^Na(C_i^b)$ in the first line. In the second line, we use the triangle inequality and the definition of $\omega^*$. In the third line, we scale $\omega^*$ to 1. Finally, we utilize the smoothness property of the activation function to bound the second term and complete the derivation. Therefore, the inequality is strictly held.

---

> ### Author Response · Authors · 2025-11-21
>
> **Q4. It remains unclear whether the proposed procedure can truly achieve the “orthogonal basis selection” objective. I have some reservations about this logical gap, even though the empirical evidence is strong.**
>
> The dimension of the activation matrix $a([I^b_i;P])\in\mathbb{R}^{d_{model}\times n}$ is far larger than the number of selected contexts $N$. In such a high-dimensional space, a set of matrices that aims to best reconstruct the correct activation $a_c$ naturally tends toward forming a set of basis-like directions. Since $a_c$ is unavailable during inference, Eq.(8) replaces it with a projection function $f(\cdot)$ that maps activations into the problem embedding space. The objectives of Eq.(7) and Eq.(8) are designed to be equivalent, and the orthogonality among activations is preserved. Further, we introduces $\mathcal{F}$ Eq.(11) to distill the composition of $f$ and $a$ to reduce computational cost of the activation $a$ and this process also keeps the orthogonality property.

---

> ### Author Response · Authors · 2025-11-27
>
> Dear Reviewer,
>
> As the author-reviewer discussion period will end soon, we would appreciate it if you could check our response to your review comments. This way, if you have further questions and comments, we can still reply before the author-reviewer discussion period ends. Thank you very much for your time!

---

### Official Review · Reviewer_xVos · 2025-10-30

**Soundness:** 4
**Presentation:** 4
**Contribution:** 4
**Rating:** 8
**Confidence:** 3

**Summary:**

This paper proposes M2CL, a multi-round context collaboration framework for multi-LLM systems (MAD) that balances diversity vs. consistency across agents: diverse initialization broadens solution coverage, while temporal consistency and dual-driven scheduling steer the system toward agreement. The method reports strong gains across multiple datasets.

**Strengths:**

1. The paper addresses the fundamental and practically important diversity–consistency trade-off in multi-agent LLM collaboration.
2. This work presents a principled methodology: it secures diversity by solving for orthogonal, sufficiently covering initial profiles, fosters inter-agent consistency via temporal cross-round coherence, and employs a dual-driven scheduling mechanism to dynamically steer the trade-off between the two.
3. It demonstrates consistent performance gains across multiple datasets.

**Weaknesses:**

1. Although a lightweight distilled $F(\cdot)$ is used, the cost may still grow combinatorially as the profile pool expands.
2. How the contextual pool is constructed, whether it is shared across tasks, and whether it can be aligned with baselines.

**Questions:**

1. Could you provide the exact input–output specification and examples. What patterns does it learn—tone control, attention to specific evidence?
2. Are generators trained per dataset or jointly? Is there cross-dataset generalization? Could you report train/val/test sizes?
3. Will you release checkpoints, data, the initial context pool ?

---

> ### Author Response · Authors · 2025-11-21
>
> **Q1. Although a lightweight distilled $F(\cdot)$ is used, the cost may still grow combinatorially as the profile pool expands.**
>
> We appreciate the reviewer’s concern. In practice, once the problem $P$ is given, we precompute all representations $\mathcal{F}([I^b_i;P])$ for the entire context pool. Since $\mathcal{F}(\cdot)$ is extremely lightweight, each forward pass is very cheap, and this preprocessing step avoids repeatedly recomputing activations during the selection process.
>
> Although the initialization involves a combinatorial search over subsets, this selection process can be parallelized because $\mathcal{F}(\cdot)$ has a small memory cost. Moreover, the time cost of single validation is efficient, which bounds the time cost of initialization. As shown in Fig.23 in Section G, the initialization stage that involves $F(\cdot)$ accounts for around 10% of the total inference time, confirming that its cost is negligible compared with the overall reasoning process.
>
> **Q2. How the contextual pool is constructed, whether it is shared across tasks, and whether it can be aligned with baselines.**
>
> The context pool used in our M2CL framework is constructed using GPT-4o, where we prompt it to generate a large collection of high-quality initial contexts across diverse domains, including mathematics, science, coding, and embodied reasoning. This ensures that the pool provides a broad coverage of reasoning perspectives and is shared across all tasks.
>
> For a fair comparison, in all baseline methods that require context initialization (such as DyLAN and GPTSwarm), we employ the same initial context. This design ensures that the observed improvements stem from our proposed context learning mechanism rather than differences in the initial context quality.
>
> **Q3. Could you provide the exact input–output specification and examples. What patterns does it learn—tone control, attention to specific evidence?**
>
> The exact input–output specification and representative examples are provided in Section H of the appendix. As illustrated in the highlighted examples, the learned instructions guide agents to follow the intermediate results from others. For instance, in the red-highlighted text — “You should pay attention to the angle subtraction idea and quietly verify if later responses align” — the context instruct the agent to focus on a specific peer’s reasoning and to integrate that evidence with its own initial context.
>
> **Q4. Are generators trained per dataset or jointly? Is there cross-dataset generalization? Could you report train/val/test sizes?**
>
> In our main experiments, the context generators are trained per dataset, and we use 20% of the questions in each dataset for training and the remaining 80% for testing.
>
> Following the reviewer's suggestion, we additionally conducted a joint-training experiment where $8$ agents participated in the discussion with their generators trained simultaneously on Math, GPQA, and Code tasks. We present the following results.
>
> |  Method  | MATH  | GPQA  | Code  |
> |  ----  | ----  |  ----  | ----  |
> | Debate  | 6.2 | 24.8 | 19.2 |
> | DyLAN  | 9.7 | 35.0 | 26.0 |
> | GPTSwarm  | 6.8 | 28.7 | 20.7 |
> | MacNet  | 11.0 | 39.6 | 30.0 |
> | M2CL(jointly)  | 23.6 | 57.5 | 37.4 |
> | M2CL(per-dataset)  | 24.9 | 59.3 | 38.9 |
>
> As demonstrated in the above table, M2CL with jointly training consistently and significantly outperforms baselines, while exhibiting only a slight degradation compared with per-dataset training. We attribute this to the generator’s ability to learn collaboration-guiding instructions that capture general coordination and communication patterns among agents (see detailed case study in Section H).
>
> To verify the cross-dataset capability of the context generators, we conduct experiment on transferring the context generator trained on GPQA into other datasets (MATH, code, and ALFWorld) with $8$ agents participating and present the results in the following table.
>
> |  Method  | MATH  | Code  | ALFWorld  |
> |  ----  | ----  |  ----  | ----  |
> | Debate  | 6.2 | 19.2 | 25.9 |
> | DyLAN  | 9.7 | 26.0 | 25.0 |
> | GPTSwarm  | 6.8 | 20.7 | 26.2 |
> | MacNet  | 11.0 | 30.0 | 30.4 |
> | M2CL  | 22.9 | 36.5 | 36.0 |
>
> As demonstrated in the above table, M2CL significantly outperforms all baseline methods across various datasets, exhibiting strong cross-dataset generalization capability. This result suggests that the generator captures domain-invariant collaboration patterns, rather than task-specific knowledge, allowing effective transfer to unseen domains without retraining.
>
> **Q5. Will you release checkpoints, data, the initial context pool ?**
>
> Yes, we plan to release all resources upon acceptance. Specifically, we will open-source the full implementation, the checkpoint of the context initialization and generator, and the complete context pool. All codes, scripts, and data necessary to reproduce the experiments will be made publicly available to facilitate future research and ensure reproducibility.

---

> ### Author Response · Authors · 2025-11-27
>
> Dear Reviewer,
>
> As the author-reviewer discussion period will end soon, we would appreciate it if you could check our response to your review comments. This way, if you have further questions and comments, we can still reply before the author-reviewer discussion period ends. Thank you very much for your time!

---

> ### Comment · Reviewer_xVos · 2025-11-28
>
> Thank you for addressing my concerns. I will maintain my positive score.

---

### Official Review · Reviewer_uL5G · 2025-10-31

**Soundness:** 3
**Presentation:** 4
**Contribution:** 3
**Rating:** 8
**Confidence:** 3

**Summary:**

The paper starts from the observation that discussion inconsistency in MAD prevents conclusions from converging, and proposes M2CL, a multi-LLM context learning method that learns a context generator for each agent to dynamically produce context instructions at every discussion round. Grounded in theoretical analysis, the method posits that agents’ activations should be as orthogonal as possible during initialization, and that subsequent evolution should reduce inter-agent activation discrepancies while controlling deviation from the initial context. During the initialization, the paper selects a near-orthogonal set of initial instructions from a predefined pool of multi-perspective prompts, minimizing the weighted reconstruction error of the selected instructions’ activations. After that, the paper proposes a round-level context evolution objective: on one hand, constraining the distance between the current round’s context and the initial context; on the other, aligning activations by encouraging the current round’s instruction to be consistent with the previous round’s own output, thereby indirectly reducing inter-agent divergence. The objective is further reformulated as a constrained problem with a dual variable $\alpha$, and alternating updates with approximate dual gradients are used to achieve adaptive trade-off. The paper evaluates M2CL across academic reasoning, embodied tasks, and mobile control scenarios , demonstrating its effectiveness.

**Strengths:**

1. The proposed method is theoretically driven and structurally clear.  The paper formalizes how to preserve diversity while fostering consensus, alleviating discussion inconsistency and non-convergence in MAD.
2. During the initialization, M2CL selects near-orthogonal instructions from a multi-perspective prompt pool.  The round-level update objective further combines a distance-to-initial constraint with activation alignment, and employs a dual variable $\alpha$ to dynamically balance diversity and consistency.
3. Thorough experiments on nine benchmarks spanning academic reasoning, embodied tasks, and mobile control, demonstrates the effectiveness  of the proposed method .

**Weaknesses:**

1. The paper uses output activation alignment as a proxy for discussion consistency and contribution. This proxy is not the task ground truth, there may exist cases where embeddings are close yet the logic still conflicts, or semantics are diverse yet complementary.
2. In initialization, one must select a near-orthogonal subset of initial instructions from a multi-perspective prompt pool and minimize the weighted reconstruction error. If the pool lacks coverage or contains heavy semantic redundancy, it is difficult in practice to pick a subset that balances diversity and reconstructability.
3. Implementing initialization requires the activation mapping $a(\cdot)$, the projection $f(\cdot)$ and the distilled projection $F(\cdot)$. Access to answer-side activations are needed and the stability of projection training may limit the initilization quality.

**Questions:**

See weaknesses.

**Details Of Ethics Concerns:**

The paper raises no ethical concerns.

---

> ### Author Response · Authors · 2025-11-21
>
> **Q1. The paper uses output activation alignment as a proxy for discussion consistency and contribution. This proxy is not the task ground truth, there may exist cases where embeddings are close yet the logic still conflicts, or semantics are diverse yet complementary.**
>
> We agree that output activation alignment is only a proxy for discussion consistency and cannot perfectly represent the task ground truth. However, due to the smoothness property of neural representations, the true semantic discrepancy between model outputs is bounded by their activation difference. In other words, when activations are well aligned, their underlying semantic representations tend to be close in semantic meaning, even if surface-level expressions vary.
>
> Moreover, directly measuring token- or semantic-level similarity would introduce other issues. For example, in a math problem, two expressions that differ only by a small superscript or symbol may appear extremely similar at the token level, while in fact representing fundamentally different meanings. Activation-based alignment, by contrast, captures deep representational similarity learned through the model’s internal reasoning process, making it more robust to superficial linguistic variations.
>
> **Q2. In initialization, one must select a near-orthogonal subset of initial instructions from a multi-perspective prompt pool and minimize the weighted reconstruction error. If the pool lacks coverage or contains heavy semantic redundancy, it is difficult in practice to pick a subset that balances diversity and reconstructability.**
>
> In our work, the pool exhibits substantial semantic diversity as it includes contexts for diverse areas such as mathematics, science, and programming, and within each domain we further design domain-specific sub-contexts. For example, in mathematics, the pool includes experts specialized in Geometric Constructions, Analytic Geometry, Vector Mathematics, and more. Consequently, when selecting a near-orthogonal subset during initialization, it is feasible to obtain contexts that are semantically distinct yet complementary, ensuring good diversity while maintaining reconstructability.
>
> **Q3. Implementing initialization requires the activation mapping $a(\cdot)$, the projection $f(\cdot)$ and the distilled projection $F(\cdot)$. Access to answer-side activations is needed and the stability of projection training may limit the initialization quality.**
>
> We agree that implementing the initialization procedure requires the activation mapping $a(\cdot)$, the projection $f(\cdot)$, and the distilled projection $F(\cdot)$. However, in practice this issue is mitigated for two reasons. First, the mapping $f(\cdot)$ only needs to project activations into a coarse problem-space representation (i.e., $v_P$), rather than reconstructing fine-grained token-level information. This greatly stabilizes the projection training. Second, the distilled model $F(\cdot)$ is intentionally designed to be lightweight and smooth, making it robust to minor noise in activation extraction and allowing consistent performance across datasets.
>
> Most importantly, our ablation study in Tab.13 empirically confirms that the initialization produced by this pipeline is effective and significantly improves performance. Removing or perturbing the initialization leads to clear drops in accuracy across all tasks, verifying that our projection-based initialization is both stable and beneficial in practice.

---

> ### Author Response · Authors · 2025-11-27
>
> Dear Reviewer,
>
> As the author-reviewer discussion period will end soon, we would appreciate it if you could check our response to your review comments. This way, if you have further questions and comments, we can still reply before the author-reviewer discussion period ends. Thank you very much for your time!

---

> > ### Comment · Reviewer_uL5G · 2025-11-28
> > **Thanks for the response**
> >
> > Thank you for the detailed response. I will maintain my positive score.

---

### Official Review · Reviewer_sNuJ · 2025-11-04

**Soundness:** 2
**Presentation:** 1
**Contribution:** 2
**Rating:** 4
**Confidence:** 3

**Summary:**

This paper addresses the problem of discussion inconsistency in Multi-Agent Discussion (MAD) systems, where multiple LLM instances fail to reach coherent solutions due to context misalignment. The authors propose M2CL, a two-stage method that (1) initializes diverse, approximately orthogonal instruction contexts, and (2) evolves these contexts across discussion rounds using learned generators with a self-adaptive balancing mechanism. Extensive experiments across 9 benchmarks (MMLU, MATH, GPQA, embodied tasks, GUI control) demonstrate 20-50% performance improvements over existing MAD methods with minimal computational overhead (~10%).

**Strengths:**

- Clear empirical demonstration (Fig 1-2) that fixed contexts cause discussion inconsistency, with concrete examples showing misalignment
- The dual gradient descent approach (Eq 16) for automatically tuning α is elegant and well-justified through Lagrangian duality (Appendix E)
- Testing across diverse domains (academic reasoning, embodied agents, mobile GUI) with multiple model sizes (7B to 72B parameters) demonstrates broad applicability
- Consistent 20-50% improvements are substantial, especially on complex tasks (e.g., +33% on GPQA with Qwen-72B)

**Weaknesses:**

Unclear notations and theoretical motivation.
- In its proof and many following lemmas, a constant term is usually ignored. In addition The constant term, however, grows exponentially with the quantity $\rho^2$. Can this constant term be ignored?
- Many concepts are introduced without context or formal introduction, making the paper extremely hard to read. For example,
  - Section 4:
      - the notation $P, X, I$ appeared in line 171-180 is understood generally as texts or sequences or tokens. But they can be magically applied in Eq (5) with weight matrices before any description. There is no clue from the authors about whether those $I, X, P$s are vectors or matrices and how they are calculated.  Furthermore, In the Theorem 4.1, the "activations of the correct answer" $a_c$ also comes from nowhere, readers still do now know whether this is a vector or a matrix. From Eq (5), a general guess is that $a(C^t_i)$ is a matrix of the same shape as $P$ because it is the output of attention layer, and it should be of the shape $[\text{number of layers (might be ignored), length of sequence, dimension of latent space}]$, if we guess $P$ is a matrix, and so as $a_c$. In the up coming Eq(8), the role of $a_c$ can be magically replaced by a sentence vector $v_P$ (this is first place the word vector appeared besides the weight vector $\omega$). Only now, people can guess $P, X, I$ are vectors.
  - Section 5.1
    - The notation $i$ in line 233 is confusing, before, it is about the i-th of N LLMs, but now it is for the predefined pool of M Initial contexts. This makes the Equation (7) very hard to understand. We do now know whether the authors wants to find a subset $S$ of what domain, of the pool? or of the LLM agents? or the tuple of (llm, context)?
    - The notation of the braces is confusing, line 184 $[I_i^t, X_i^{t-1}, P]$ uses $,$ and line 196/237 $[I_i^t; X_i^{t-1}; P]$ uses $;$, are they the same?
    - The sentence vector $v_P$ represents the vector of question P, is it the same as the activation of P?
    - Is $a([A;P])$ in Eq (9) computed from the Eq (5)? Are the parameters $W_V$, $W_K$, $W_Q$ shareable across all occurrence of $a$?
    - The loss function (10) is strange, why don't we just use the composition of functions $f$ and $a$ as $\mathcal{F}$?
    - Consider Eq (11), if functions are optimized by the loss functions in Eq (9-10), I didn't see why the entire formulation in Section 5.1 encourages orthogonality, because each $\mathcal{F}( ... )$ approximate $f( ... )$ and approximate $v_P$. And the critical input contains all information of $P$ , I think ideally, the function $F$ can collapse to a sentence embedding function to generate $v_P$. It is just a knowledge distillation of the sentence embedding network.
  - Section 5.2
    - Eq (12) is poorly written. In Eq (12), it tries to select the proper index j to maximize the objective. In its remark, the instruction $I$ magically appeared to be optimized. What is the optimization problem you actually want to define?
    - Eq (16) is strange, how do you choose $\beta$? If $\beta$ is sufficiently large, the $\alpha_i$ will just decrease to zero and another part of loss will only keep the first term, seems to only approximate $X_i^{t-1}$ using $\mathcal{G}_{\theta_i}$.

**Questions:**

My major concerns remains about the confusing methodology part. I would like to discuss with the authors about the notations above.

---

> ### Author Response · Authors · 2025-11-21
>
> **Q1. In its proof and many following lemmas, a constant term is usually ignored. In addition, the constant term, however, grows exponentially with the quantity $\rho^2$. Can this constant term be ignored?**
>
> To clarify, we have not ignored the constant term $\exp(\rho^2)$ in all the theoretical results, including Theorem 4.1 and lemmas to prove it. In fact, we omit this term in the training of the context generator as it is a constant and does not influence the optimization process. Further, the constant term arises from the matrix spectral exponential $\exp(\rho^2)$. Although exponential with $\rho$, it generally remains small in practice because the attention matrices ($(W_KX)^TW_QX$) are typically well-conditioned and $\rho\leq1$ holds [1]. To corroborate it, we test the maximum values of $\exp(\rho^2)$ across datasets as follows:
>
> |  Method  | MATH  | GPQA  | Code  |
> |  ----  | ----  |  ----  | ----  |
> | $\exp(\rho^2)$  | 2.84 | 2.17 | 2.99 |
>
> [1] Li, S., Song, Z., Xia, Y., Yu, T., & Zhou, T. (2024). The closeness of in-context learning and weight shifting for softmax regression. Advances in Neural Information Processing Systems, 37, 62584-62616.
>
> **Q2. Many concepts are introduced without context or formal introduction, making the paper extremely hard to read.**
>
> Thank you for pointing this out! To clarify, in our paper, $P,X,I\in\mathbb{R}^{d_{model}\times n}$ denote the embeddings of the question, other agents’ responses, and instructions respectively, rather than the raw text or token sequences. Each matrix consists of a number of token embeddings, each with dimension $d_{model}$. In addition, the output of the activation $a(C_i^t)\in\mathbb{R}^{d_{model}\times n}$ is a matrix with the same shape as the concatention $[P;X;I]$.
>
> We use $a_c\in\mathbb{R}^{d_{model}\times n}$ to represent the correct attention activation that guides the model toward the correct reasoning, and $v_P\in\mathbb{R}^{d\times n}$ represents the sentence embedding corresponding to question $P$ generated by the encoder of the context generator. Since we cannot obtain the correct activation $a_c$ during initialization, we utilize a function $f$ to map these activations into a sentence embedding space. In fact, we use a surrogate objective, defined as $\mathbf{I^b} = \\{I^b_1,\dots,I^b_N\\} = \arg\min\limits_{I^b}\\{\min_\omega\\|\sum_{i=1}^N \omega_i f(a([I^b_i; P])-f(a_c))\\|\\}$, to characterize the difference between the summation of activations and $a_c$ in Eq.(7).
>
> **Q3. The notation $i$ in line 233 is confusing, before, it is about the i-th of N LLMs, but now it is for the predefined pool of M Initial contexts. This makes the Equation (7) very hard to understand. We do now know whether the authors wants to find a subset $S$ of what domain, of the pool? or of the LLM agents? or the tuple of (llm, context)?**
>
> To circumvent the ambiguity, we have refined Eq.(7) to make its meaning explicit: $\mathbf{I^b} = \\{I^b_1,\dots,I^b_N\\} = \arg\min\limits_{\mathbf{I^b}}\\{\min_\omega\\|\\sum_{i=1}^N \omega_i a([I^b_i; P]) - a_c \\|\\}.$
>
> Here, $\mathbf{I^b}$ denotes the selected subset of initial contexts, and $I^b_i$ denotes the initial context of the i-th LLM.
>
> **Q4. The notation of the braces is confusing, line 184 $[I_i^t,X_i^{t-1},P]$ uses , and line 196/237 $[I_i^t,X_i^{t-1},P]$ uses ;, are they the same?**
>
> The notation in line 184 is a typo. Both expressions, $[I_i^t, X_i^{t-1}, P]$ and $[I_i^t; X_i^{t-1}; P]$, are intended to represent the same concatenation operation. We have fixed this inconsistency in the revised version.
>
> **Q5. The sentence vector $v_P$ represents the vector of question P, is it the same as the activation of P?**
>
> The sentence vector $v_P$ refers to the sentence embedding corresponding to question $P$, which is generated by the encoder of the context generator, and it is different from the activation of $P$ which is denoted as $a(P)$.
>
> **Q6. Is $a([A;P])$ in Eq (9) computed from the Eq (5)? Are the parameters $W_V$, $W_K$, $W_Q$ shareable across all occurrence of $a$?**
>
> The attention $a([A;P])\in\mathbb{R}^{d_{model}\times n}$ in Eq.(9) is computed using the attention mechanism in Eq.(5), and the parameters $W_V$, $W_K$, and $W_Q$ are shared across all occurrences of $a$.
>
> **Q7. The loss function (10) is strange, why don't we just use the composition of functions $f$ and $a$ as $\mathcal{F}$?**
>
> Using the composition of functions $f$ and $a$ directly is logically correct. However, in practice, computing the activation $a$ is time costly as it requires running a full forward pass of the LLM over the entire context pool. To reduce this computational cost, we distill the composition of $f$ and $a$ into $\mathcal{F}$. The results in Fig.23 demonstrate the efficiency of this initialization method.

---

> ### Author Response · Authors · 2025-11-21
>
> **Q8. Consider Eq (11), if functions are optimized by the loss functions in Eq (9-10), I didn't see why the entire formulation in Section 5.1 encourages orthogonality.**
>
> The dimension of the activation matrix $a([I^b_i;P])\in\mathbb{R}^{d_{model}\times n}$ is far larger than the number of selected contexts $N$. In such a high-dimensional space, a set of matrices that aims to best reconstruct the correct activation $a_c$ naturally tends toward forming a set of basis-like directions. Since $a_c$ is unavailable during inference, Eq.(8) replaces it with a projection function $f(\cdot)$ that maps activations into the problem embedding space. The objectives of Eq.(7) and Eq.(8) are designed to be equivalent, and the orthogonality among activations is preserved. Further, we introduces $\mathcal{F}$ Eq.(11) to distill the composition of $f$ and $a$ to reduce computational cost of the activation $a$, and this process also keeps the orthogonality property.
>
> Of note, the activation of LLMs and the correct activation $a_c$ are different from each other and the mapping function $f$ preserves this difference. This leads to the difference between $\mathcal{F}([I^b_i;P])$ and $v_P$, which means the optimal $\mathcal{F}$ is not a distilled embedding function. In fact, $\mathcal{F}$ bridge the connection between initial context with the problem space, which help to select contexts by Eq.(8).
>
> **Q9. What is the optimization problem you actually want to define?**
>
> Eq.(12) is not the optimization objective of the context generator; rather, it serves as a round-wise criterion to evaluate the individual contribution of each context $C_i^t$ during a discussion round. It measures how much each context improves collective reasoning quality at that step. The overall objective is defined in Eq.(6), which aims to minimize the difference between the correct activation and each agent's activation. To optimize this, we divide it into two steps including context initialization and evolution based on Theorem 4.1. For the context evolution, the objective is given in Eq.(14), which aims to maximize the cumulative contribution of generated contexts across rounds. In other words, Eq.(14) aggregates the per-round evaluations defined by Eq.(12) to update the context generator toward producing contexts that consistently enhance overall multi-agent discussion performance.
>
> **Q10. Eq (16) is strange, how do you choose $\beta$? If $\beta$ is sufficiently large, the $\alpha_i$ will just decrease to zero and another part of loss will only keep the first term, seems to only approximate $X_i^{t-1}$ using $\mathcal{G}_{\theta_i}$.**
>
> In Eq.(16), $\beta$ controls the regularization strength on context similarity—that is, how closely the generated context should remain to its initial form. If $\beta$ is set large, the regularization is weak, meaning the generated context does not need to stay close to the initial one. In this case, the model tends to favor agreement among agents, and the behavior may degenerate toward a Best-of-N style aggregation. Conversely, when $\beta$ is small, the regularization dominates, encouraging each agent to retain its original reasoning perspective, leading to debate where diverse roles are preserved.
>
> Further, we conducted experiments with varying $\beta$, and as shown in Fig.12–16, the performance first increases (benefiting from discussion consistency) and then decreases when $\beta$ becomes too large (due to loss of perspective diversity).

---

> ### Author Response · Authors · 2025-11-27
>
> Dear Reviewer,
>
> As the author-reviewer discussion period will end soon, we would appreciate it if you could check our response to your review comments. This way, if you have further questions and comments, we can still reply before the author-reviewer discussion period ends. Thank you very much for your time!

---

### Author Response · Authors · 2025-12-02

**Reviewer g8ch**

* Reviewer g8ch raised concerns about the clarity of Fig.2. Following the suggestion, we have verified the discrepancy difference between successful and failed cases, and clarified that with the help of the context generators, our method benefits from diverse initial perspectives and progressively guides agents toward consensus.
* Reviewer g8ch asked why cannot an LLM summarizer be used to generate contexts. We have clarified that the quality of the summary becomes a critical bottleneck and majority noise, where most agents generate the same wrong answer (see Section H), makes it difficult to generate an accurate and useful summary. We have also compared our approach with GPTSwarm, which uses an LLM summarizer, and show that our method outperforms it by a significant margin in all settings (see Fig.5–10).
* Reviewer g8ch questioned the derivation of Eq. 19. We have clarified that no term was omitted in the derivation and added detailed description in Section B.
* Reviewer g8ch questioned the orthogonality of the initial contexts. We have clarified that the dimension of the attention activation is far larger than the number of selected contexts, and thus the matrices that best reconstruct the activation naturally form basis-like directions, promoting orthogonality.

---

### Author Response · Authors · 2025-12-02

Dear AC, we summarize our responses and the updates made to the paper as follows:

**Reviewer sNuJ**

* Reviewer sNuJ questioned why the constant term is ignored. We have clarified that this term is **not** ignored in our theoretical analysis and is explicitly included in Theorem 4.1 and its accompanying lemmas. We have also verified that the matrix spectral radius $\rho$ is typically less than one, ensuring that this constant has minor impact on the tightness.
* Reviewer sNuJ raised concerns regarding ambiguous notations. We have meticulously refined the manuscript. Specifically, we have detailed the matrix dimensions in Lines 174–177 and 200; corrected typographical errors in Eq.3; revised the objectives in Eq.7–9 and Eq.12.
* Reviewer sNuJ questioned the purpose of using $\mathcal{F}$ in Eq.11. We have clarified that this lightweight $\mathcal{F}$ is a distilled version of the composition of $f$ and $a$, which reduces the computational cost of calculating the attention activation $a(\cdot)$.
* Reviewer sNuJ questioned the orthogonality of the initial contexts. We have clarified that the dimension of the attention activation is much larger than the number of selected contexts, and the set of matrices that best reconstructs the correct activation naturally tends toward forming basis-like directions, thereby promoting the orthogonality of the initial contexts.
* Reviewer sNuJ questioned the selection of the context constraint $\beta$. We have clarified that $\beta$ controls the regularization strength on context similarity, thereby influencing the diversity and consensus level of the multi-agent discussion. We have conducted experiments with varying $\beta$, as shown in Fig.12–15.

**Reviewer uL5G**

* Reviewer uL5G raised concerns about the gap between output activation alignment and semantic similarity as a proxy for discussion consistency. We have clarified that the difference between semantic tokens can be bounded by their activation difference due to the smoothness of neural representations, and that attention activations better capture deep representational similarity through the model’s internal reasoning process. Details have been added in Lines 202-205.
* Reviewer uL5G questioned whether the context pool may contain heavy semantic redundancy. We have clarified that the context pool consists of contexts from diverse domains, along with domain-specific sub-contexts within each domain. We have added a detailed description in Section F.3.
* Reviewer uL5G raised concerns about the instability of training the projection $f(\cdot)$ for initialization. We have clarified that this mapping only needs to project activations into a coarse problem-space representation, and the lightweight distilled model $\mathcal{F}$ makes the process robust to minor activation noise. The ablation study on initialization in Tab.13 further verifies that our projection-based initialization is both stable and beneficial.

**Reviewer xVos**

* Reviewer xVos raised concerns that the time cost of initialization grows combinatorially. We have clarified that preprocessing all context representations and running the selection process in parallel significantly reduces the computation cost. The experimental results in Fig.23 demonstrates the efficiency of the initialization process.
* Reviewer xVos questioned the construction and utilization of the context pool. We have clarified that the context pool is built from a large collection of high-quality initial contexts across diverse domains using GPT-4o, and we have added a detailed description in Section F.3.
* Reviewer xVos questioned the pattern of the learned contexts. We have provided a case study in Section H and clarified that the learned contexts explicitly guide agents to follow intermediate results from other agents.
* Reviewer xVos raised concerns regarding the experimental setup and the cross-dataset generalization of the context generators. We have clarified that the generators are trained per dataset. We have also added experiments on joint training and transferring generators to unseen domains to verify cross-dataset generalization.
* Reviewer xVos asked about open-sourcing the checkpoint and data. We plan to release all codes, checkpoints, and data necessary to reproduce the experiments upon acceptance.

---

### Meta-Review · Area_Chair_weUz · 2025-12-29

**Summary:**

The decision is grounded in the reviewers' acknowlegement on the significance of the discussion inconsistency problem in multi-agent systems. The primary reservations centered on the clarity of the theoretical exposition, specifically the mathematical notation regarding matrix dimensions and the justification for omitting constant terms in proofs.

**Reviewer Concerns:**

The rebuttal successfully resolved concerns regarding the computational scalability and generalization capability of the method through additional experiments and clarification of the pre-computation strategy.

**Reviewer Scores:**

Reviewers uL5G and xVos would maintained their strong acceptance ratings following the rebuttal, indicating their full satisfaction with the response.
Reviewer g8ch would likely have increased the score upon verifying the mathematical validity of the inequality derivations and the comparative analysis of discrepancy in success versus failure cases.
Reviewer sNuJ likely maintained the score but the clarifications regarding notations and definitions would have resolved their concerns about the paper's presentation.

---

### Decision · Program_Chairs · 2026-01-26

Accept (Poster)